# Towards Measuring Predictability: To which extent data-driven approaches can extract deterministic relations from data exemplified with time series prediction and classification

**Saleh Gholam Zadeh**                                        *salehgholamzadeh17@gmail.com*
*Autonomous Learning Robots,*
*Karlsruhe Institute of Technology*
*SAP SE*

**Vaisakh Shaj**                                                  *vaisakhs.shaj@gmail.com*
*University Of Edinburgh*

**Patrick Jahnke**                                                       *pj@turba.ai*
*Turba AI*

**Gerhard Neumann**                                           *gerhard.neumann@kit.edu*
*Autonomous Learning Robots,*
*Karlsruhe Institute of Technology*

**Tim Breitenbach**                                *tim.breitenbach@uni-wuerzburg.de*
*SAP SE*
*Julius-Maximilians-Universität Würzburg*

**Reviewed on OpenReview:** *https: // openreview. net/ forum? id= jZBAVFGUUo*

## Abstract

Minimizing loss functions is one important ingredient for machine learning to fit parameters such that the machine learning models extract relations hidden in the data. The smaller the loss function value on various splittings of a dataset, the better the machine learning model is assumed to perform. However, datasets are usually generated by dynamics consisting of deterministic components, where relations are clearly defined and consequently learnable, as well as stochastic parts where outcomes are random and thus not predictable. Depending on the amplitude of the deterministic and stochastic processes, the best achievable loss function value varies and is usually not known in real data science scenarios. In this research, a statistical framework is developed that provides measures to address the predictability of a target given the available input data and, after training a machine learning model, how much of the deterministic relations have been missed by the model. Consequently, the presented framework allows to differentiate model errors into unpredictable parts regarding the given input and a systematic miss of deterministic relations. The work extends the definition of model success or failure as well as the convergence of a training process. Moreover, it is demonstrated how such measures can enrich the procedure of model training. The framework is showcased with time series data on different synthetic and real-world datasets. The code is available at https://github.com/Saleh-Gholam-Zadeh/predictability_measure.

# 1 Introduction

Data analysis and the application of the corresponding insights work if there are reliable and stable relations between the measured quantities and their model. However, due to the fact that any measurement may not be error free or the dynamics that determine the values of the considered quantities may be inherently noisy to a certain extent, measurement values do not only purely reflect the relations to be investigated. Furthermore, the input data could miss relevant information, e.g., relevant features are not provided or even measured, to model the output data such that regarding the given input data some parts of the output are unpredictable due to the lack of information. One example is that past stock prices may not always provide a clear indication of future prices due to the complexity of market dynamics and the influence of external factors such as economic events and investor sentiments. Therefore, accurately predicting future stock prices might not be directly evident from historical observations alone. Without having a criterion for the amount of predictable relations between both the input and output or model deviations, respectively, in such scenarios, one could spend a huge amount of time on learning dynamics which either don't exist (such as a pure noise) or it is impossible to learn more than a certain level because the given history is sharing no or low information with the output. Usually, real-world datasets are not binary in terms of being predictable, meaning that there are some deterministic patterns plus patterns which can't be inferred from the provided historic data, as might be the case in the example from the financial markets above, and therefore there is no chance for any model to predict them totally accurately. Depending on the relative magnitude of these elements, the prediction error can vary even for a successful model. In such cases, it is beneficial to know the upper bound of the model's performance to avoid trying to train it at all where there is no deterministic relation or to improve it while further improvement is not possible. We extend the term for model success by its ability to extract or learn, respectively, all the available deterministic relations. Consequently, we argue that the magnitude of prediction errors alone does not unequivocally signify the model success or failure because a high inaccuracy might arise from the unpredictability of the data, rather than from deficiencies in the model or flaws in the learning process.

One key issue before any data analysis, machine learning (ML) model training or information extraction from a dataset is testing whether such reliable relations exist between the quantities in the dataset of interest, as outlined in the example above. Such tests deliver valuable insights to evaluate efforts for further analysis, and if reasonable at all. A second key aspect after an iteration of data analysis or information extraction, such as the training of an ML model, is testing if there is information left to extract or all the reliable information is extracted in terms of deterministic relations between input and output data. If all relations are extracted or learned, respectively, the deviations between the predictions given the input data and the target (ground truth) are supposed to be stochastically independent of the input data. Thus, given that input, additional analysis on this input-target relation might not reveal further insights or improve accuracy in terms of extracting more deterministic relations. Consequently, further training might not improve a model in terms of learning these deterministic relations.

Following the outline above, in the presented work, we provide a framework to address the following, which is illustrated in Figure 1:

- We utilize measures to evaluate the stochastic dependence and information content between random variables modeling the data to evaluate to what extent the data is interconnected and to quantify extractable information based on defined input and output (target) variables assembled from the dataset.

- After fitting a model, we use these measures to estimate if there is still information left to extract. This evaluation is done by testing the stochastic independence and information content between the input and the deviations of the model output regarding the ground truth. For this purpose, we investigate the relation of random variables that model corresponding quantities.

These bullet points provide the foundation of the presented framework for characterizing failure cases in prediction modeling. To provide an implementation of a practical framework and to give evidence with some showcases for its usability is the main purpose of this work. In our work, a failure case is defined as a

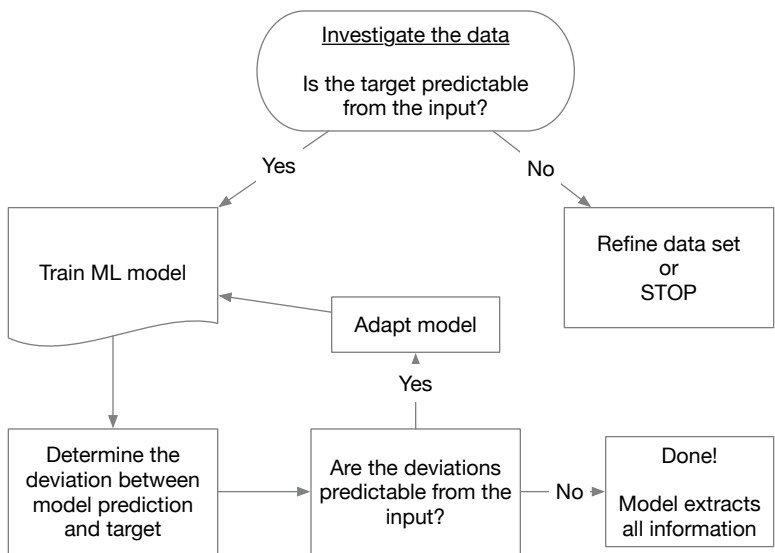

Figure 1: Graphical abstract representing the main concept of the presented work to analyze for information content and information extracted or learned, respectively, by a model.

scenario in which a prediction model fails to capture the essential, meaning deterministic, dynamics of the target variable(s), given the input information. This definition of failure allows us to decide on whether there is potential for improvement or the prediction model has already achieved the best possible scenario, where inaccuracies are not further predictable and thus not improvable, given the input data. Our mathematical framework provides an additional insight for model evaluation and extends the current performance measures of ML models considering not only the ground truth and the model output but also the input and thus to what degree the ground truth is predictable given the input. This extension aligns with the perception in the area of generative models such as GAN Yoon et al. (2019); Jeha et al. (2022) or diffusion models Kollovieh et al. (2024); Tashiro et al. (2021); Li et al. (2022); Yuan & Qiao (2024) where the discriminative score, the generative score, the Frechet Inception Distance (FID), the Maximum Mean Discrepency (MMD), the Negative Log Likelihood (NLL) or the Continuous Ranked Probability Score (CRPS) is also reported besides the standard evaluation framework of mean squared or absolute error (MSE/MAE) to have a more probabilistic view by taking distributions of predictions into account. One contribution of the presented work is to extend the term of model assessment beyond the model accuracy based on its output and ground truth to also taking the predictability of the deviation between model output and ground truth into account given the input. Moreover, we provide also numerical implementations of such measures and show its applicability on synthetic and real-world data sets. Through the numerical experiments, we exemplify determining if the deviation is improvable at all given the input data.

One benefit of our metrics, which are based on the stochastic independence of input and model deviations, is that they can provide an explanation for poor model accuracy. The explanation is given by analyzing the deviations if there is information left or deviations consist only of unpredictable parts from the perspective of the input data. The ability to differentiate what data points are predictable and what are not given the input data is not provided by accuracy measures or loss functions that only consider the model output and ground truth, for instance mean squared or absolute errors. Instead, we additionally consider the information the model deviations share with the input data. The framework is not limited to a specific model architecture and thus is model-agnostic since we only inspect the dependence of input and the model errors, which does not require the knowledge of the inner function of the model.

The implementation of this general framework provided with this work is the mutual information, the chi-square test of independence and the Pearson correlation to investigate the existence of relations between random variables. However, our framework is not limited to these tests. Consequently, other methods that test for information content or stochastic dependence can be used and easily included into our modularized

git repository. The choice of the measures and their implementations themselves are not the focus of this work.

Evaluating the deviations for dependence on the input can be applied to facilitate model training particularly as follows. The first three bullet points are elaborated in Sections 4.1, 4.2, 4.3 and 4.4, while the last two bullet points are addressed in Section 4.5:

- Success criterion: We provide a loss function agnostic framework to evaluate the performance of ML accuracy defined by the quantity of extracted deterministic relations, which can be used as a model selection criterion. Furthermore, we can decide if a loss function value is a corresponding lower bound for the given dataset since no further deterministic relations remain within the dataset.

- Further convergence criterion: Apart from utilizing the loss function for convergence, we can stop training whenever there is no information or dependence left between the input and the deviation of a model prediction from the ground truth.

- Hyperparameter tuning: We can stop a grid search whenever we have identified a hyperparameter set with which the corresponding model has extracted all the information on the training or validation set.

- Distribution shift detection: A data splitting into training and validation set or historical and present can result in different stochastic properties, therefore resulting in different learnable relations in the sets.

- Data efficiency: In case of a closed system where distribution shift can be excluded, e.g., the dynamics are known and fixed, the total amount of training data can be used for fitting model parameters since no data is required to be utilized as an additional set for early stopping.

In Section 4, we present examples demonstrating the utility of our framework for issues described in the bullet points above. Among different data modalities such as text, image, and time series, our insight is that in time series data, the existence of a clear relationship between input and output is often not directly noticeable. Therefore, in this work, we focus specifically on time series analysis. For instance, in natural language processing (NLP), we typically do not encounter a sequence of random words or tokens. Similarly, in image datasets, depending on the task (e.g., classification, segmentation), the relationship between input and output usually exists a priori, and one would not typically spend time proving such existence, instead would directly train a model to solve the task at hand. However, in time series analysis, the underlying dynamics of a process need to be learned from the measurements. For this purpose, we showcase our general framework with an application to time series forecasting. The basic procedure is the following. After quantifying the level of predictability by inspecting the input and ground truth target, we can perform an exactly similar test on the input and the residuals, which define the deviations as ground truth minus prediction, to see whether they are still predictable given the input. We use supervised ML techniques to predict future parts of the time series. Another use case is the application for time series classification with nominal target data. This use case differs from the ordinal output data above as model deviations cannot be defined by differences due to the lack of order. A corresponding concept is presented and its usability to quantify left information is showcased in this work. One further example of unpredictable time series might be the prediction of workloads, mainly network traffic in data centers, where time points of the triggering of the workload by users might be unpredictable. In Greenberg et al. (2009), we find: *"measurement studies found two key results with implications for the network design. First, the traffic patterns inside a data center are highly divergent..., and they change rapidly and unpredictably."* A more recent manifesto by Buyya et al. (2018) re-iterated the brittleness of existing *"demand estimation and workload prediction methods"*, leaving it as an open question if *"Machine Learning (ML) and Artificial Intelligence(AI) methods could fully address this shortcoming"*. With our present work, we can even provide a quantified answer in each case.

**Paper outline** The work is organized as follows: The main theoretical concepts used in the proposed framework are explained in Section 2. This includes a precise definition of the measure of mutual information and the chi-square test of independence, as well as an equinumeric discretization scheme for features with a continuous co-domain.

A section of related methods follows Section 2 and addresses among others how our work extends related works using mutual information, ensemble learning and time series prediction.

Applications of the information extraction evaluation of a model are showcased in Section 4 with different experiments in the area of time series prediction and classification. These experiments include a basic proof of concept, an analysis of the influence of the loss function on the information extraction depending on the structure of the noise, suggest additional convergence criteria based on the presented framework, as well as detect distribution shifts of the data.

In the Discussion, we provide assumptions and limitations of our current approach. Furthermore, we sketch further applications of our framework in the area of unstructured data. Moreover, efficient model size reduction is also discussed by ranking subparts of a model with stochastic measures since the presented framework is general and can be applied to any function generating output from input data.

## 2 Theoretical background and Methods

In this section, we provide the necessary background and mathematically establish our framework.

### 2.1 Basic concept for unpredictability in a nutshell

**Definition of unpredictability**
The random variable $Y$ is deemed unpredictable with respect to an information set given by the random variable $X$ if the conditional distribution $P(Y = y|X = x)$ aligns with the unconditional distribution $P(Y = y)$ according to

$$P(Y = y|X = x) = P(Y = y) \tag{1}$$

for all $x$ and $y$. Specifically, if $X$ comprises the past realization of $Y$, then (1) suggest that having knowledge about these past realizations does not enhance the predictive accuracy of $Y$. It is important to note that this form of unpredictability in $Y$ is an inherent attribute, unrelated to any prediction algorithm Bezbochina et al. (2023). One famous example of unpredictable time series is white noise where samples are identically but independently distributed (iid), where the independence of future samples from the past samples makes it essentially unpredictable and therefore training an ML model is pointless. In this case, the best predictor in terms of $L^2$-loss is a mean predictor, suggesting further investing on improving the prediction model is fruitless. Please note that while unpredictable data and noisy data are related, they are not equivalent. Noise represents a specific subset of unpredictability.

On the other hand, if (1) doesn't hold true, it suggests that given $X$, it is reasonable to train an ML model to predict $Y$. The more the distributions of the two sides of (1) deviate from each other, the more chance we have to train a model with a high accuracy to predict $Y$ based on $X$. In the scope of this work, we call $X$ the context/input variable and $Y$ the target/output that we want to predict. In this work, we use the terms input and context interchangeably, analogously the terms output and target. In the following part, we explain how to measure and quantify the concept we have introduced so far.

**Measuring predictability**
Assuming (1) is satisfied, we can derive the joint distribution by

$$P(Y = y, X = x) = P(Y = y)P(X = x) \tag{2}$$

for all $x$ and $y$ by utilizing the definition of conditional probability. However, in practical data science scenarios (2) barely holds entirely true even in case of independence of $X$ and $Y$ due to noise or numerical errors. Therefore, it is crucial to quantify the degree to which the independence assumption is met and to introduce statistical concepts to decide based on a level of significance if the hypothesis of independence cannot be rejected.

Our approach to quantifying the independence of these random variables is grounded in (2), where we measure the deviation between its left and right-hand side. Although in general any measure of deviation can be used, in this work, we mainly focus on Kullback–Leibler divergence and chi-square test of independence

as a measure of this deviation. In the former case, such deviation can be calculated based on the mutual information formula given by

$$I(X;Y) = D_{\mathrm{KL}}(P_{(X,Y)} \| P_X \otimes P_Y)$$

where $D_{\mathrm{KL}}$ is the Kullback–Leibler divergence, $P_X \otimes P_Y$ denotes the outer product distribution, and $P_{(X,Y)}$ is the joint distribution (see Murphy (2022) Section 2.2.4 Figure 2.3). A higher value of mutual information represents higher predictability of $Y$ based on $X$ which aligns with the definition of mutual information, measuring the information gained about $Y$ through observation of $X$.

**Quantifying measure of success**
We define a successful prediction when the deviations between model output and ground truth, e.g., the residual defined as the ground truth minus the prediction, are stochastically independent of the context/input information. In case of independence, the deviations contain no information shared with the context, rendering them effectively unpredictable based on the provided context information. Consequently, no further improvements are possible, marking the prediction as successful. It is crucial to note that as long as some mutual information remains, there is potential for enhancements, whether through selecting a different model, loss function, or adjusting various parameters. For the sake of assessment, lower values of mutual information suggest a better prediction quality.

We remark that the scope of this work does not include the development of new methods to calculate, estimate or approximate mutual information but the application of such methods to improve ML training. In the Related Work section, we provide references to such methods. For the demonstration of the application, we provide one implementation that worked for our experiments. Our framework is flexible and does not depend on a specific method to quantify stochastic independence, allowing for various measures such as mutual information. The following sections provide a rigorous mathematical formulation and details on the framework's implementation

## 2.2 Foundations and implementation details for the predictability framework

In this section, we explain our framework in detail.

**Measures for stochastic independence and information content**
We are given $n \in \mathbb{N}$ input features in the format $x \in \mathbb{R}^n$ and $m \in \mathbb{N}$ output features in the format $y \in \mathbb{R}^m$. Each feature is modeled as a random variable taking values upon measurement. Consequently, the set of input random variable is given by $X = (x_1, ..., x_n)$, $x_i : \mathbb{R} \to Z_{x_i}$, $t \mapsto x_i(t)$, $i \in \{1, ..., n\}$ where $t$ is a time point of measurement and $Z_{x_i} := \left\{ z_{x_i}^k \right\}_{k=1,...,l_{x_i}}$, $l_{x_i} \in \mathbb{N}$, is a set of discrete events. In our case, such an event is defined through the random variable $x_i$ taking a value at time point $t$ between two predefined boundaries (see Figure 9 and 10 in the Appendix). In this work, these boundaries are calculated by an adaptive scheme presented in Algorithm 1.

Analogously, the set of output random variables is defined by $Y = \{y_1, ..., y_m\}$, $y_j : \mathbb{R} \to Z_{y_j}$, $t \mapsto y_j(t)$, $j \in \{1, ..., m\}$ and $Z_{y_j} := \left\{ z_{y_j}^k \right\}_{k=1,...,l_{y_j}}$, $l_{y_j} \in \mathbb{N}$, is a set of discrete events. Our framework not only holds for random variables with a discrete co-domain but also for random variables with a continuous co-domain. In our case, we choose to discretize continuous domains and we will later explain Algorithm 1 that discretizes random variables with a continuous co-domain equinumerically. Furthermore, independent of the topology of the co-domain, we define an information or a dependence measure by $\Phi : X \times Y \to \mathbb{R}$, $(x_1, ..., x_n, y_1, ..., y_m) \mapsto \Phi(x_1, ..., x_n, y_1, ..., y_m)$ that describes how much information, resp., stochastic dependence exists between the input and the output variables. An example for such a measure can be the stochastic independence of multiple real-valued random variables, see, e.g., (Gallager, 2013, 1.3.4) or mutual information as outlined in Subsection 2.1.

In this work, we focus on a specific structure of $\Phi$, which is a pairwise test between input and output random variables providing corresponding information or stochastic dependence summing up each value of

the pairwise measure. The measure $\Phi$ is given by

$$\Phi\left(x_1, ..., x_n, y_1, ..., y_m\right) \coloneqq \sum_{i=1}^{n} \sum_{j=1}^{m} \phi\left(x_i, y_j\right)$$

where $\phi : X \times Y \to \mathbb{R}$, $(x_i, y_j) \mapsto \phi\left(x_i, y_j\right)$ for each $i \in \{1, ..., n\}$ and $j \in \{1, ..., m\}$.

We are aware that a pairwise test might be an approximation of the actual value of the measure for the deterministic relations, e.g., as in the case of the stochastic independence of multiple real-valued random variables. However, this approximation provides the advantage of much less computational costs, in particular for large $n$ and $m$ as provided in the discussion section of Breitenbach et al. (2022). It is one outcome of this work that this approximation is a useful measure to estimate the deterministic relations between input and output as well as model deviations between predictions and ground truth, moreover to estimate the learning success of a model which is the reduction of deterministic relations between the input dataset and the model deviations from the ground truth. A similar consideration holds for the mutual information where a precise calculation of the joint probability can be very costly in case of many input and output variables and where other mutual information estimators exist as well to circumvent this issue, please see the related work and the discussion section for references and further details about this issue.

In the present work, we focus on the mutual information and the chi-square test of independence between two random variables as measures, which are both explained later in detail. However, the presented work is generic and can also be executed with different measures for independence, like Pearson's correlation coefficient as defined in, e.g., Breitenbach et al. (2023) for random variables. We remark that in terms of testing a hypothesis if input and model deviations are independent of each other, it is beneficial to rely on several tests to be sure about the consistency of the results and not having a wrong decision because requirements of a test are not fulfilled, which is often challenging to verify.

In the following part, we explain the main ingredients of the present framework to analyze for deterministic relations. Although these concepts might be well-known, we repeat them here for the convenience of the reader since they are central for this work and a precise definition consistent with this work facilitates its understanding.

**Mutual information**

The probability $P\left(x_i = z_{x_i}^{k_1}\right)$ describes the likelihood that the outcome of $x_i$ equals the event $z_{x_i}^{k_1}$ for any $i \in \{1, ..., n\}$ and any $k_1 \in \{1, ..., l_{x_i}\}$. Analogously for $P\left(y_j = z_{y_j}^{k_2}\right)$ for any $j \in \{1, ..., m\}$ and $k_2 \in \{1, ..., l_{y_j}\}$. The probability $P\left(x_i = z_{x_i}^{k_1} \wedge y_j = z_{y_j}^{k_2}\right)$ describes the likelihood that the outcome of $x_i$ equals the event $z_{x_i}^{k_1}$ and the outcome of $y_j$ equals the event $z_{y_j}^{k_2}$ for any $i \in \{1, ..., n\}$, $j \in \{1, ..., m\}$, $k_1 \in \{1, ..., l_{x_i}\}$ and $k_2 \in \{1, ..., l_{y_j}\}$. Based on this definition, we can define the mutual information for a pair of random variables $x_i$ and $y_j$ as one example for $\phi$ as follows

$$I\left(x_i, y_j\right) \coloneqq \sum_{k_1=1}^{l_{x_i}} \sum_{k_2=1}^{l_{y_j}} P\left(x_i = z_{x_i}^{k_1} \wedge y_j = z_{y_j}^{k_2}\right) \log_a \left( \frac{P\left(x_i = z_{x_i}^{k_1} \wedge y_j = z_{y_j}^{k_2}\right)}{P\left(x_i = z_{x_i}^{k_1}\right) P\left(y_j = z_{y_j}^{k_2}\right)} \right) \tag{3}$$

for any $i \in \{1, ..., n\}$ and $j \in \{1, ..., m\}$ with the basis $a \in \mathbb{N} \setminus \{1\}$ of the logarithm. Mutual information describes how much information we gain about the outcome of $y_j$ given the outcome of $x_i$. If the outcome of $x_i$ is independent of $y_j$, namely

$$P\left(x_i = z_{x_i}^{k_1}\right) = P\left(x_i = z_{x_i}^{k_1} | y_j = z_j^{k_2}\right) \coloneqq \frac{P\left(x_i = z_{x_i}^{k_1} \wedge y_j = z_{y_j}^{k_2}\right)}{P\left(y_j = z_{y_j}^{k_2}\right)},$$

for all $k_1 \in \{1, ..., l_{x_i}\}$ and $k_2 \in \{1, ..., l_{y_j}\}$, where $P\left(x_i = z_{x_i}^{k_1} | y_j = z_j^{k_2}\right)$ is the conditional probability that $x_i = z_{x_i}^{k_1}$ under the condition that $y_j = z_j^{k_2}$, we expect zero mutual information since $\log_a 1 = 0$.

The mutual information is bounded from below by 0. The upper bound depends on the number of events of $y_j$. In order to normalize the mutual information such that it is bounded from above by 1 for any $y_j$, we define the corresponding log base $a = l_{y_j}$. We remark that in case where $y_j$ is replaced by the corresponding model deviation, the basis is defined accordingly to the number of different events of the model deviation from the ground truth. The normalization is in particular important to compare the information content between $x_i$ and $y_j$ with the left information content between $x_i$ and the corresponding deviation between model output and ground truth after the training of the ML model to estimate the information extraction of the model from the dataset.

As a next example, we introduce the chi-square test of independence of two random variables.

**Chi-square test of independence**
In case the chi-square test of independence is taken as the measure for stochastic independence, then we define in this work that $\phi$ returns 1 if the corresponding random variables of the pair are not independent of each other, else 0.

Let us have $N \in \mathbb{N}$ measurements where at each measurement the values of all random variables are determined. For any fixed $i \in \{1, ..., n\}$ and $j \in \{1, ..., m\}$, we define

$$P\left(x_i = z_{x_i}^{k_1} \wedge y_j = z_{y_j}^{k_2}\right) := \frac{O_{k_1 k_2}}{N}$$

where $O_{k_1 k_2} \in \mathbb{N}$ is the number of observed measurements where $x_i = z_{x_i}^{k_1}$ and $y_j = z_{y_j}^{k_2}$ for the corresponding $k_1 \in \{1, ..., l_{x_i}\}$ and $k_2 \in \{1, ..., l_{y_j}\}$. Then, we have

$$P\left(x_i = z_{x_i}^{k_1}\right) = \sum_{k_2=1}^{l_{y_j}} \frac{O_{k_1 k_2}}{N} \text{ and } P\left(y_j = z_{y_j}^{k_2}\right) = \sum_{k_1=1}^{l_{x_i}} \frac{O_{k_1 k_2}}{N}$$

with $N = \sum_{k_2=1}^{l_{y_j}} \sum_{k_1=1}^{l_{x_i}} O_{k_1 k_2}$. Under the hypothesis that the random variables $x_i$ and $y_j$, $i \in \{1, ..., l_{x_i}\}$ and $j \in \{1, ..., l_{y_j}\}$, are stochastically independent, the number of expected measurements $E_{k_1 k_2} \in \mathbb{R}$ where $x_i = z_{x_i}^{k_1}$ and $y_j = z_{y_j}^{k_2}$ is given by

$$E_{k_1 k_2} := P\left(x_i = z_{x_i}^{k_1}\right) P\left(y_j = z_{y_j}^{k_2}\right) N$$

for the corresponding $k_1 \in \{1, ..., l_{x_i}\}$ and $k_2 \in \{1, ..., l_{y_j}\}$ as we can take the corresponding proportions from $N$ independent of the outcome of the other random variable. Consequently, if $x_i$ and $y_j$ are independent, it is necessary that the observed and the expected number of measurements for all $k_1 \in \{1, ..., l_{x_i}\}$ and $k_2 \in \{1, ..., l_{y_j}\}$ are each equal. The chi-square statistic given by

$$\chi^2 := \sum_{k_1=1}^{l_{x_i}} \sum_{k_2=1}^{l_{y_j}} \frac{\left(O_{k_1 k_2} - E_{k_1 k_2}\right)^2}{E_{k_1 k_1}}, \tag{4}$$

equals zero if $O_{k_1 k_2}$ and $E_{k_1 k_2}$ equal each other for all $k_1 \in \{1, ..., l_{x_i}\}$ and $k_2 \in \{1, ..., l_{y_j}\}$ and quantifies the deviation from not being equal. However, due to the presence of noise, even under independence of $x_i$ and $y_j$, it might hold that $(O_{k_1 k_2} - E_{k_1 k_2})^2 > 0$ for some $k_1 \in \{1, ..., l_{x_i}\}$ and $k_2 \in \{1, ..., l_{y_j}\}$. Consequently, we need to estimate from a distribution how likely the observed chi-square value under the hypothesis of independence is. If the observed value is too unlikely, we rather assume that the opposite of our hypothesis of independence is the case, meaning the variables depend on each other and there exists a dependence between the random variables in taking their values. It can be proven that $\chi^2$ is chi-square distributed with $(l_{x_i} - 1)(l_{y_j} - 1)$ degrees of freedom, see, e.g., (Rao, 1973, 6d.2) or (Georgii, 2015, 11.3). One important assumption is that the term $\frac{O_{k_1 k_2} - E_{k_1 k_2}}{\sqrt{E_{k_1 k_1}}}$ is approximately normally distributed, which is usually sufficiently the case if $E_{k_1 k_2} \geq 5$ for all $k_1 \in \{1, ..., l_{x_i}\}$ and $k_2 \in \{1, ..., l_{y_j}\}$, see, e.g., McHugh (2013) or (Greenwood & Nikulin, 1996, page 21). Based on the distribution, we can calculate a p-value for the observed $\chi^2$ value. The p-value is the

probability to get a higher chi-square value than the observed one under the hypothesis of independence. If the p-value is too small, e.g., for this work we use lower than the level of significance of 0.01, we reject the hypothesis of independence and assume that the random variables take their values not independent of each other. Our measure of dependence is the number of chi-square tests that indicate dependence of the tested pairs for the input and output variables $(x_i, y_j)$ for each $i \in \{1, ..., n\}$ and $j \in \{1, ..., m\}$. However, since the number of chi-square tests is given by $nm$, which can scale to large numbers, we use an adapted p-value to lower the risk of wrongly rejected hypothesis, which would result in assuming too often dependence. We use the Bonferroni-correction dividing our level of significance 0.01 by the number of chi-square tests $mn$.

**Discretization scheme for co-domains of random variables**
In the next part, we describe how to discretize real-valued continuous random variables, meaning that the co-domain is continuous. We take Algorithm 4 from Breitenbach et al. (2022). We remark that our algorithm also works for real-valued discrete random variables without any change. Consequently, no separation between discretized and continuous random variables is necessary. The algorithm provides an adaptive discretization scheme for each random variable, meaning that boundaries of the bins, in which the co-domain of a random variable is divided and define the events denoted with $z_{x_i}^{k_1}$ or $z_{y_j}^{k_2}$, are not set equidistant but rather equinumeric. With equinumeric, we mean that each bin has - if possible - the same number of data points which balances the likelihood of each event.

---

**Algorithm 1** Discretize the co-domain of random variables

    1. Set $\rho \in \mathbb{N}$ number of minimum data points per bin.

    2. For any random variable $v$:

        (a) Determine the minimum value $m$ and the maximum value $M$ of all measured values of $v$.

        (b) If $M \leq m$: Skip $v$

        (c) If $M > m$:

            i. Sort the measured data points of $v$ in ascending order.

            ii. Go through the data points in ascending order. Determine the range of a bin such that there are at least $\rho$ data points within the current bin and that the value of the last data point of the current bin is smaller than the first one of the next bin.

            iii. If there are less than $\rho$ data points left: Join these data points with the last bin that has at least $\rho$ data points.

            iv. Output: Upper and lower bound of each bin defining the events $z_v^k$ for $k \in \{1, ..., l_v\}$.

---

Algorithm 1 works as follows. The parameter $\rho \in \mathbb{N}$ determines the minimum number of data points per bin in which the co-domain of a random variable, such as $x_i$ or $y_j$, is discretized. These upper and lower bounds of each bin define the event $z_{x_i}^k$ meaning that the random variable $x_i$ takes a value within the corresponding $k^{\text{th}}$ bin. Analogously, it holds for the events $z_{y_j}^k$ of the random variable $y_j$. Thus, the parameter $\rho$ influences the marginal probability $P\left(x_i = z_{x_i}^{k_1}\right)$ and $P\left(y_j = z_{y_j}^{k_2}\right)$ of the joint probability $P\left(x_i = z_{x_i}^{k_1} \wedge y_j = z_{y_j}^{k_2}\right)$ and consequently the corresponding expected frequency $E_{k_1 k_2}$ as well as the number of bins for each random variable. Increasing $\rho$ will increase the quantity $E_{k_1 k_1}$, which might be useful if $E_{k_1 k_1} < 5$ for one $k_1 \in \{1, ..., l_{x_i}\}$ and one $k_2 \in \{1, ..., l_{y_j}\}$. Consequently, this discretization scheme is beneficial for the implemented chi-square test and can be used without any restriction for the calculation of the mutual information as well. We need to keep in mind that a too big $\rho$ might lead to a too coarse discretization deleting information from the continuous random variable. In case of the chi-square test, one strategy might be to start with a small $\rho$ where $E_{k_1 k_1} < 5$ for one $k_1 \in \{1, ..., l_{x_i}\}$ and one $k_2 \in \{1, ..., l_{y_j}\}$ and increase $\rho$ until $E_{k_1 k_1} \geq 5$ for all $k_1 \in \{1, ..., l_{x_i}\}$ and all $k_2 \in \{1, ..., l_{y_j}\}$. A further advantage of that scheme is that by ensuring always a minimum number of data points in each bin of each marginal distribution, the denominator in the mutual information and chi-square formula is never zero which could happen with an equidistant discretization strategy.

In step 2 a), we determine the minimum and the maximum of the available data points of the corresponding random variable $v$ to filter out constant random variables in 2 b). Without any variation, constant random variables do not provide any information for predicting something for what at least two different kind of events are necessary. For non-constant random variables, we sort the data points of the random variable in ascending order. Once this is done, we can go through the points in ascending order and determine the boundaries of the bins such that at least $\rho$ data points are included in a bin. If there are several data points with equal values, we include all these points into the current bin such that the first data point in the next bin is larger than all data points in the bin before. If the remaining data points, which have not yet been associated to a bin, are less than $\rho$ data points, we include them into the bin with the largest upper bound.

The binning generated by Algorithm 1 can be used for the chi-square test and the mutual information to calculate corresponding marginal and joint probabilities as we do in our implementation provided with this work. The marginal and joint probabilities of (3) and (4) are calculated as follows. We take the frequency of each event $z_{x_i}^{k_1}$, $z_{y_j}^{k_2}$ and $z_{x_i}^{k_1} \wedge z_{y_j}^{k_2}$ and divide it by $N$ to obtain the corresponding probabilities $P\left(x_i = z_{x_i}^{k_1}\right)$, $P\left(y_j = z_{y_j}^{k_2}\right)$ and $P\left(x_i = z_{x_i}^{k_1} \wedge y_j = z_{y_j}^{k_2}\right)$ for all $k_1 \in \{1, ..., l_{x_i}\}$, $k_2 \in \{1, ..., l_{y_j}\}$, $i \in \{1, ..., n\}$ and $j \in \{1, ..., m\}$. In Appendix A, the process is illustrated with time series data.

**Stochastic independence as a hypothesis test**
We conclude this section with a remark about the distribution of test statistics. Even under the hypothesis of independent data/random variables, there are some variations by coincident that lead to a distribution of the test statistic. Consequently, we need a probability how likely it is under the assumption of independent random variables (no deterministic relation between input and output) to get a test statistic value bigger than the observed one, and thus exclude that the relation in the measured data is just by coincidence. Then, we can decide if a certain observed test statistic is too unlikely under the assumption of independent random variables, and we should rather assume that the opposite is true, meaning that there are some deterministic relations by whose effect random variables do not take their values independent of each other. Based on such a statistical view, all models where it cannot be rejected that their deviations from the ground truth are independent of the input are equally good in terms of extracting the deterministic relations.

The chi-square value $\chi^2$ is chi-square distributed with corresponding degrees of freedom under certain assumptions. Consequently, we can evaluate the likelihood for the observed chi-square value of being generated by two independent random variables. This likelihood can be calculated for each pair of random variables separately as in (4), which results in $mn$ decisions. However, when considering each term of (4) as a random variable where $(l_{x_i} - 1)(l_{y_j} - 1)$ of them take their value freely for each $i \in \{1, ..., n\}$ and $j \in \{1, ..., , m\}$, resulting in the corresponding degree of freedom under which we evaluate the likelihood of the observed $\chi^2$, we can add the chi-square value of all random variables together as follows

$$\chi^2 := \sum_{i=1}^{n} \sum_{j=1}^{m} \sum_{k_1=1}^{l_{x_i}} \sum_{k_2=1}^{l_{y_j}} \frac{(O_{k_1 k_2} - E_{k_1 k_2})^2}{E_{k_1 k_1}}. \tag{5}$$

The likelihood of $\chi^2$ defined in (5) under the hypothesis of independent input and output variables can be evaluated as sum of $\sum_{i=1}^{n} \sum_{j=1}^{m} (l_{x_i} - 1)(l_{y_j} - 1)$ freely varying normalized random variables, defining the degree of freedom for the corresponding chi-square distribution. The value of $\chi^2$ defined in (5) can be used as a measure for stochastic independence between input and output as an alternative to the number of stochastic independent input and output pairs as described in the context of (4). The difference is in defining the function $\phi$.

We are not aware of a theoretical distribution for the mutual information defined in (3) under the hypothesis of independent variables. However, to get a threshold for mutual information, such that we can decide based on a level of significance if an observed mutual information value is too unlikely under the assumption of independent input and output variables, we can numerically determine a distribution of mutual information with the following permutation procedure. The idea is to shuffle the association of value pairs between input and output based on the available data according to $(x_i(t), y_j(\pi(t)))$ for all $i \in \{1, ..., n\}$ and $j \in \{1, ...., m\}$ where $\pi : T \to T$, $t \mapsto \pi(t)$ is a bijective map, called permutation, and $T$ is the set of all time points of measurements of the data points. Intuitively, the permutation test destroys the relation between input and

output by shuffling them. The reason is the sampling of the data from the marginal distributions, instead of from the joint distribution, thus, enforcing independence by design. Ideally, the mutual information computed from samples drawn according to the marginal distributions is zero. However, in practice, due to possible bias in any finite sampling, it might not be exactly zero. Therefore, the mutual information computed from this test can serve as an experimental minimum or baseline. If the computed mutual information value based on the dataset samples, which are distributed according to the joint probability distribution, is unlikely according to the numerically determined distribution from the permutation test based on independent variables, we assume that the computed mutual information value is determined by deterministic mechanisms relating input and output. This applies to both model output and deviations from the ground truth. The observed mutual information value is considered unlikely under the hypothesis of independent input and output variables if less than a predefined percentage of mutual information values (level of significance; in this work 5%) generated with the randomly shuffled data is bigger than the observed mutual information value. The distribution is generated by calculating mutual information for several random permutations (100 times in this work). The same procedure holds to calculate a distribution for the chi-square value defined in (4) or (5) in case assumptions are violated to justify the application of the chi-square distribution for $\chi^2$. However, we remark that the theoretically available distributions for the test statistic defined in (4) and (5) are an advantage in terms of computation time as the above described permutation method requires repeated computations to generate the distribution under the assumption of independent input and output variables. Finally, with such a stochastic framework in place, we are able to make concrete decisions if a further improvement is possible given the data or if all the learnable relations are already extracted.

## 3 Related Work

**Relation to feature selection methods** In analogy to the idea of using information and stochastic dependence measures for feature selection, we apply the same methods to the input and the deviations between the model prediction and the ground truth. Our approach for measuring the predictability of output based on input variables is similar to feature selection methods, such as the minimize redundancy maximize relevance (mRMR) method Peng et al. (2005), which is based on mutual information.

**Mutual information estimation and applications**
Besides classical methods to estimate mutual information such as Darbellay & Vajda (1999); Kraskov et al. (2004); Gretton et al. (2005); Moon et al. (1995); Kozachenko & Leonenko (1987); Gao et al. (2017), a more recent neural network based estimation of mutual information Belghazi et al. (2018); Oord et al. (2018); Song & Ermon (2020); Franzese et al. (2024) is proposed. Such methods provide an alternative to Algorithm 1 for the calculation of mutual information. However, since it is the focus of this work to use such implementations to test model convergence in terms of independent input and output, we use a framework in which we can directly implement further stochastic measures such as the chi-square test, not being just specific for mutual information estimation. Our work may further motivate the development of such methods as we give another application of such measures and estimators for model evaluation. Furthermore, in different scenarios it may be that different requirements of certain estimation methods are more or less fulfilled, which justifies considering several stochastic measures and their implementations which is facilitated by our modularized approach. Moreover, by using different statistical tests, we can demonstrate how to include hypotheses testing to decide for independent input and output, like the target data, residuals or model deviations, which allows us to make a stochastic decision for model convergence based on the given data. Nevertheless, in Czyż et al. (2023) several mutual information estimators are benchmarked regarding their accuracy. This work provides evidence that in low-dimensional settings, such as the pairwise case, histogram based methods, e.g., Algorithm 1, can provide accurate results. Under model convergence with independent input and model deviations, it is required that the pairwise test confirms independent input and model deviations as well, which represents a necessary condition for convergence. On the other hand, if we have to calculate the full mutual information according to Czyż et al. (2023), neural estimators may better handle high dimensional settings, making it suitable to compute the joint mutual information.

The work Xu et al. (2019) introduces an information-theoretic measure called Determinant-based Mutual Information (DMI) to train deep networks that are robust to label noise. However, the key conceptual difference from our approach is that they do not account for the input in the same way we do. Specifically, once

the prediction and the ground truth are available, their method does not consider the input that generated the prediction anymore. In contrast, our approach consistently evaluates whether a given prediction can be improved based on the provided input, taking into account the entire triple of input, prediction, and ground truth, whereas Xu et al. (2019) focus solely on a tuple consisting of the prediction and ground truth.

DeepInfoMax (DIM) Hjelm et al. (2019) as well as contrastive predictive coding (CPC) Oord et al. (2018) maximizes mutual information between raw data and its compressed representation to find a better representation of raw images. Furthermore, Chen et al. (2016) employs mutual information to find a disentangled and interpretable representation of images. The work of Brakel & Bengio (2017) also focused on finding a disentangled/independent representation/features of images and uses mutual information as a measure of independence between these features and thus the degree of disentanglement. Our framework can be directly applied to the corresponding generated representations to test if the model accuracy is only caused by unpredictable parts like noise. For more details, please see the Discussion about the case of unstructured input data.

**Innovation method**
In time series analysis and signal processing, the Innovation method Reid & Term (2001); Houts et al. (2013) is a widely recognized metric. It is often used as an indirect measure of the consistency of the prediction of the model (typically Kalman Filter) when the true (ground truth) latent states of a dynamical system are inaccessible. The Innovation method works by comparing the observed data with the model's predictions with the goal that the differences —known as innovations or residuals— have a zero mean and a specific covariance where residuals from different time points are uncorrelated. For Kalman Filters, this covariance can be determined in closed form. A range of tests Reid & Term (2001) can be conducted to ensure these criteria are met, thereby confirming the filter is consistent and reliable. While the Innovation method uses stochastic means to investigate properties of the residuals to evaluate the model performance, our framework additionally includes the input data to evaluate the stochastic independence of input and residuals to evaluate the potential of a model to be improved due to remaining deterministic relations between input and residuals. Due to computational challenges the calculation of stochastic measures might have and thus due to the usage of approximations, the Innovation method and our presented framework can be used as necessary tests that a successful model has to pass. Furthermore, our framework also suggests an option how to extend to nominal ground truth data where it is not possible to define differences and thus residuals.

**Ensemble learning**
Our proposed framework can be considered as an extension of ensemble learning methods such as Wortsman et al. (2022), in particular any kind of boosting algorithms Schapire & Freund (2012). We refrain from the indiscriminate combination of multiple models. Our framework triggers a combination only if it confirms the presence of potential for further improvement in terms of further learnable/extractable deterministic relations. Thus, we can define a convergence criterion for the number of combined models.

**Information bottleneck**
Another application of our framework is to extend the information bottleneck concept Saxe et al. (2019); Kawaguchi et al. (2023). The basic framework of the information bottleneck is to maximize the mutual information between a data representation sought and the corresponding output data and at the same time to minimize the mutual information between this representation and the input data. There is a parameter to balance both contradicting requirements. With our framework, we can extend the information bottleneck method by providing a procedure how to choose this balance parameter to find a lossless compression of the input data. Starting with a configuration where the model extracts all the information between input and output data, we tune the balance parameter such that the compression is weighted higher until the model cannot extract all the information between input and output assessed by having dependent input and residuals or model deviations. With that procedure, we find the threshold of the balance parameter for a lossless compression.

**Time series prediction**
Besides the above related methods, the showcase of this work is in particular associated with time series prediction. In the past few years many of time series prediction models have been developed to improve the prediction performance, such as RNN, LSTM Memory (2010), GRU Cho et al. (2014), transformer and

its variants Vaswani et al. (2017); Kitaev et al. (2019); Zhou et al. (2021); Liu et al. (2022); Zhang & Yan (2022); Wu et al. (2021); Zhang et al. (2022) and state space models such as RKN Becker et al. (2019), S4 Gu et al. (2021), MTS3 Shaj et al. (2023) as well as simple yet effective linear models such as Zeng et al. (2023). The improvement is measured in terms of corresponding loss function values of the best weight configuration of a model. We would like to extend these evaluation by our framework where we introduce another quantity that evaluates how much information is left to extract from the time series given the input length of the historic data. Furthermore, our approach would like to establish a level before model training, that allows for evaluation if a time series is predictable given historic data. This is in particular important for challenging time series and to evaluate if already simple models are sufficient in a prediction scenario Zeng et al. (2023), Li et al. (2023). Nevertheless, there is a question in all scenarios: How can we make sure that the current method has already achieved the lowest possible bound of the error for each time series or can it still be improved by fitting a better model for each case?

## 4 Applications|Numerical experiments

In this section, we showcase our framework to analyze time series for their predictability, measure the dependence between input/context and output/target data for the purpose specified in the bullet points on page 4 and demonstrate how to measure learning success of models with our framework. Moreover, we show how our framework extends the performance evaluation of a model.

Apart from several real world datasets, we use a synthetic dataset to purely demonstrate the core concept of the paper described in the bullet points on page 2 where we can control properties of the noise, such as the ratio between information and noise, which we do not know in a real dataset. The synthetic dataset consists of the time series that is composed of the sinus function, $\sin : \mathbb{R} \to \mathbb{R}$, $t \mapsto \sin(t)$ and the random variable $\theta : \mathbb{R} \to \mathbb{R}$, $t \mapsto \theta(t)$, which generates noise by random values according to $y(t) := \sin(t) + a\theta(t)$ where $a > 0$ scales the amplitude of the noise. In order to evaluate performance in an already established metric, we use the normalized root mean squared error (RMSE) defined by $\frac{1}{\sigma}\sqrt{\sum_{i=1}^{N}(\tilde{y}_i - y_i)^2}$ where $N \in \mathbb{N}$ is the number of measurement points, $\tilde{y}_i$ is the model prediction at the discrete time point $i \in \mathbb{N}$, $y_i$ is the ground truth and $\sigma$ is the standard deviation $\sqrt{\sum_{i=1}^{N}(\bar{y} - y_i)^2}$ calculated on the training dataset where $\bar{y}$ is the mean of the data points of the training data.

The parameter $\rho$ of Algorithm 1 is set to 5% of the number of measurement points of the corresponding dataset on which the algorithm is performed, unless otherwise stated.

We remark that the term residuals is usually used instead of model deviations to name the difference between model output and the ground truth in case of ordered target data. In the nominal case, we further use the term model deviation, which is defined in Subsection 4.4. We show in both settings the stochastic dependence decays over training epochs on training and validation datasets.

We provide a short overview about the intention of each subsection in this section and what it is supposed to demonstrate:

- Subsection 4.1: The noise-independent evaluation of model performance shows the success criterion.

- Subsection 4.2: The loss-function agnostic evaluation of model convergence demonstrates the success criterion as well.

- Subsection 4.3: Differentiating model inaccuracies into systematic model failures or those due to unpredictability and thus show stochastic independence of input and residuals or model deviations as a stopping criterion. Consequently, it shows the success criterion, a convergence criterion and thus the usability for determining hyperparameters.

- Subsection 4.4: The application of our framework to nominal data. It demonstrates the first three bullet points of page 4 for the classification problem.

- Subsection 4.5: The detection of distribution shift in the sense that relations learned by a model do not hold on a data set different from the training data set. Consequently, it shows the distribution shift detection and data efficiency bullet point of page 4.

While these items demonstrate the numerical application of our proposed framework outlined in the bullet points of the Introduction, we have an additional section in the Appendix, Section E, where we stack models on top of each other to correct the output of the former model. The numerical experiments with the stacking architecture provide further evidence that any remaining stochastic dependence between input and model deviations, identified with our implementation, can be used to correct former predictions. These dependencies are not detectable with accuracy metrics that only consider model output and ground truth.

## 4.1 Splitting noise off from model inaccuracy

In practical scenarios where data is obscured by noise or cases that some components of the future samples are not predictable based on the given features, like the the history, evaluating prediction models becomes challenging. Traditional metrics such as $L^2$-loss on validation set may not suffice due to the uncertainty surrounding the ratio of unpredictable to predictable parts. This uncertainty complicates determining the lower bound of prediction error beforehand. Therefore, solely relying on loss metrics for model assessment is insufficient, as the source of error could be either model inadequacy/failure or inherent data unpredictability. For example, when the power of noise (as an unpredictable component) equals to half of the data power, the lowest possible normalized mean squared error (MSE) on validation is 0.5. However, by employing our framework that identifies when the model has learned the primary data component effectively, we can stop training and attribute the remaining residual to initial data noise, although the $L^2$-error is still high. Motivated by this discussion, we start our experiments with such a data where the first component is fully predictable such as the sinusoid function plus some independent, identical distributed Gaussian noise that is completely unpredictable, however, the portion of each is controllable for us in this synthetic case.

For the numerical implementation, we consider a window of size 50 and the first 49 samples are considered as context/input to the model and the $50^{\text{th}}$ sample as the target/output. To generate the dataset, the window is slid over the time series that was split into training and test/validation set before the experiment according to a ratio of 0.75/0.25.

For the experiments depicted in Table 1, the random variable $\theta$ adds Gaussian noise. According to the table, the chi-square test and the mutual information indicate that the MLP model, trained with $L^2$-loss, has extracted the sinus function. We see that all essential information is extracted because the chi-square test indicates that all residuals are independent of the input. In the case of mutual information where the p-values are greater than 0.01, it is indicating that based on that level of significance, we cannot reject the hypothesis that the input and the residual are independent of each other and thus the input is unlikely to carry information about the outcome of the residual. Therefore, in these cases, even a much more sophisticated model is not able to decrease the error further and might lead to overfitting to the noise. While the RMSE increases with the amplitude of the noise, indicating that the model performance would become worse, our stochastic measures indicate that the model has extracted the sinus function as the main deterministic driver of the time series. This can be seen since the $L^2$-norm between model prediction and the pure sinus value, shown as Actual Err in Table 1, is (almost) independent of the noise amplitude. Furthermore, the fact that the normalized test RSME is always close to the relative noise standard deviation indicates that the deviation between model and data results from the noise and not from a systematic deviation from the sinus function. We remark that the first and third column can only be presented because we exactly know the signal and the noise component separately. In real world data, these numbers usually cannot be computed, however, our method can provide the valuable insight if there is some relation left to extract or the deviation between model and data is rather due to errors coming from unpredictable relations given the input data.

## 4.2 Information extraction influenced by loss function and noise properties

In this part, we show that small values of stochastic measures coincide with a model that learned the underlying dynamics of the deterministic part of the data. Moreover, the choice of loss function is an important

| Rel. noise std | Exp. Err | | Chi-square Analysis | | Mutual Information Analysis | | | | |
|---|---|---|---|---|---|---|---|---|---|
| | RMSE | Actual Err | Init. dep. var | Res. dep. var | Init. MI | Init. Perm | Res. MI | Res. Perm | pv |
| 0 | $2.2 \times 10^{-5}$ | $2.2 \times 10^{-5}$ | 49 | 49* | 27.497 | $0.7385 \pm 0.0272$ | 27.2781 | $0.4039 \pm 0.236$ | 0 |
| 0.2715 | 0.2731 | 0.0616 | 49 | 0 | 8.3351 | $0.7076 \pm 0.0136$ | 0.7301 | $0.7096 \pm 0.0114$ | 0.05 |
| 0.4927 | 0.4914 | 0.1073 | 49 | 0 | 4.193 | $0.7337 \pm 0.0119$ | 0.7643 | $0.7332 \pm 0.0108$ | 0 |
| 0.6465 | 0.6568 | 0.1476 | 47 | 0 | 2.5647 | $0.7024 \pm 0.0104$ | 0.7472 | $0.7049 \pm 0.0095$ | 0 |
| 0.7482 | 0.7461 | 0.1682 | 38 | 0 | 1.6659 | $0.7325 \pm 0.0092$ | 0.7503 | $0.7054 \pm 0.0089$ | 0 |
| 0.8166 | 0.8184 | 0.1789 | 37 | 0 | 1.2591 | $0.7293 \pm 0.0095$ | 0.7862 | $0.7296 \pm 0.0085$ | 0 |
| 0.8611 | 0.8595 | 0.1831 | 30 | 0 | 1.053 | $0.711 \pm 0.0077$ | 0.7562 | $0.7102 \pm 0.0069$ | 0 |
| 0.8921 | 0.9068 | 0.1893 | 28 | 1 | 0.9725 | $0.7275 \pm 0.0086$ | 0.7931 | $0.7278 \pm 0.0075$ | 0 |
| 0.9148 | 0.9134 | 0.1846 | 19 | 0 | 0.8825 | $0.7109 \pm 0.0079$ | 0.7605 | $0.7384 \pm 0.0082$ | 0.01 |
| 0.9306 | 0.9415 | 0.1934 | 11 | 0 | 0.8751 | $0.7341 \pm 0.0075$ | 0.7521 | $0.7070 \pm 0.0073$ | 0 |
| 0.9425 | 0.9410 | 0.1952 | 5 | 0 | 0.8153 | $0.7053 \pm 0.007$ | 0.7652 | $0.7042 \pm 0.0085$ | 0 |

Table 1: Impact of varying amplitudes of white noise on a sinusoidal signal and assessing its influence on the performance of a simple MLP. The relative power of the noise component, which is the root of the variance of noise divided by the variance of the total time series, is shown in the first column, which is the theoretical lower bound of test RMSE. In the second column, the normalized RMSE on the test data is shown. The third columns shows the $L^2$-loss between the prediction and actual clean sinusoid. The fourth column illustrates the dependency of the target on the history evaluated by chi-square test, which is initially complete until almost half of the power is taken by noise and decreases to small numbers when noise becomes the dominant (94 percent) part of the time series. The fifth column evaluates the dependency of residual target on the corresponding context to show how successful the model is to reduce this dependencies. The last five columns show the same concept in terms of mutual information. Similar to chi-square, we have initial and residual values as well as two more columns to report their corresponding lower bound which is obtained by random permutation of input and the corresponding residual, see Subsection 2.2 for details. The last column shows the p-value of the observed mutual information under the hypothesis that residuals are independent of the input given the dataset calculated by the mentioned random permutation. *Note: * The model fits the data up to numerical errors occurring in the residuals around the peak of the sinus, which represents a predictable pattern due to the periodicity, see Figure 15. Such cases can be handled by introducing a threshold indicating a very close fit between model output and data in some norm. An alternative could be to impose a minimal bin width in Algorithm 1 which might impact the equinumeric property.

key factor to extract the deterministic relations from the data. To demonstrate it, we use our synthetic dataset where the random variable $\theta$ is defined by $\theta(t) := \frac{1}{10}\theta_1(t) + 10\theta_2(t)$, $\theta_1$ is Gaussian distributed with mean zero and standard deviation 1 and $\theta_2$ is a Poisson distributed random variable over the number of peaks (high values of the time series) within a time interval. If at a time point $t$ there is such a random peak, the variable $\theta_2(t) = 1$ and 0 otherwise. In this numerical experiment, the rate of arrivals of a peak is $\frac{1}{200}$ peaks per sample (or expected time between two peaks), meaning that we expect one peak after 200 data points on average.

Since peaks only increase the values of the time series randomly, the mean of the noise is greater than zero, indicating asymmetry. In this case, we see in Figure 2 that the initial dependence between input and output is totally reduced only by the MLP that is trained with the $L^1$-loss function while it does not extract the total deterministic relations when using an $L^2$-loss function. By "extracting deterministic relations", we refer to learn/approximate the sinus-function with the model. This extraction has taken place when only noise is left that is independent of the input, as shown in Figure 2 where we see that in the $L^2$-norm the model output trained with $L^1$-loss function is much closer to the corresponding sinus-function, evaluated on the validation set. In other words, the results of Figure 2 depict that the model trained with an $L^2$-loss function is prone to the properties of unpredictable patterns, specifically irregular peaks, in this experiment. For illustration, plots of predictions and residuals for $L^1$- and $L^2$-loss functions are given in Figure 16. We remark that also here the synthetic set is useful since we know the exact functional formula that generates the data apart from noise in order to confirm that our utilized metrics can better reveal which prediction is more in line with the underlying relations in the data, in particular where these metrics are solely applied on measured and possibly noisy data. Consequently, this example demonstrates that when a stochastic measure reports independence between model deviations and input, it indicates that the model has successfully extracted deterministic relationships, excluding noise or unpredictable patterns. This aligns with being close to the

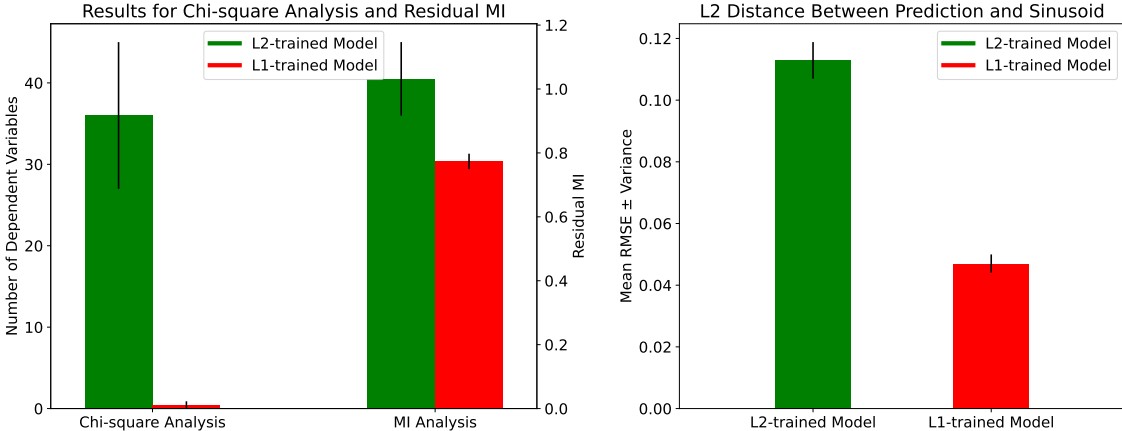

Figure 2: In the left sub-figure, the bars represent the stochastic measures of independence between the input and residuals, along with their corresponding standard deviations obtained from several training runs with different seeds. The initial number of dependent variables is 49. The value of initial MI is $13.342 \pm 0.071$. The p-values are shown in Figure 3. For the $L^2$ model, the p-value always remains zero. The right sub-figure shows the RMSE values between the model predictions and the clean (pure) sinusoid for models trained with $L^2$- and $L^1$-loss functions. The lower bound on the error is 0, as the predictions are compared to the pure sinusoidal function. This results is completely in line with the result of the left sub-figure as being closer to the clean data coincides with a lower value of the stochastic measures. Notably, these metrics require no prior knowledge of the clean data, underlying relationships, or model architecture, as they rely solely on the input, output, or residual data. All models were trained on noisy data. More information is provided in Figure 3. All values are reported on the validation set.

ground truth, not in terms of fitting the training data (which includes noise), but in capturing the underlying deterministic dynamics. In our case, this corresponds to the pure sinusoidal signal derived from the input data. A further example is given in the Appendix, Section C, showing that independence of input and model deviations indicates being close to the clean data.

An explanation for the observations is that the minimization of a loss function under the constraints of the model does not necessarily coincide with extracting the real dynamic that underlies a dataset or extracting the most information from the dataset, respectively. For example, the minimum of the $L^2$-loss function subject to the constraints of the model is more distracted from the real dynamics (sinus-function) by the specific noise than the $L^1$-correspondence. Another reason for the difference of the loss functions in taking optima might be that the $L^1$-loss weights any deviation between model and ground truth equally regardless if the corresponding data point is predictable given the input while mutual information, like any other measure based on stochastic independence, takes only predictable data points into account. Depending on the properties of the unpredictable data part, this difference might lead to different optima regarding the model's parameter values. The key point is that these stochastic dependence measures can determine which optimum is closer to the underlying dynamics, something loss functions cannot always achieve.

### 4.3 Stochastic measures as convergence criteria

Next, we show how the mutual information and the chi-square test evolve during the training after each epoch to demonstrate its capability to work as convergence criteria both for model training and hyperparameter search. The procedure is as follows. If input and output are not independent of each other, training of a model is started. If after some epoch, input and residuals or model deviations, respectively, are independent regarding the training or validation set, more detailed if we cannot reject the hypothesis that input and residuals, resp., model deviations are stochastically independent, then there is no information left to extract and we can stop the training. The importance of the stochastic measure is that we can evaluate if flattening of the loss function after some epochs is due to whether the training is done in terms of information extraction

or if the convergence speed meanwhile has just slowed down. In the latter case, it is worth further patience since further information may be extracted. Moreover, during later epochs, the convergence speed might increase again once the optimal weight updates are identified. The advantage of a stochastic measure taking the relation between input and model deviations into account is that there is a lower bound of dependence known in advance. This bound is theoretically always achievable by a model, namely when all the deterministic relations are extracted, which is by definition independent of the unpredictable parts given the input data. In contrast, for loss functions that do not take this relation into account, there is no lower bound known in advance that is achievable on the concrete dataset.

In this experiment, we use the data from Section 4.2 and plot relevant metrics in Figure 3. We see that when the loss function doesn't improve any more, the corresponding chi-square test is zero and the p-value of the mutual information becomes non-zero in a magnitude of order such that we cannot reject the independence of input and residuals. Although we only check pairwise instead of all possible combinations of variables (joint distribution), see the limitations in the Discussion for details, this experiment shows that our framework provides valuable convergence criteria as a stagnating loss function coinciding with stochastic independence can be rather interpreted that the model has extracted all the deterministic relations between input and output. The rationale is that if input and residuals or model deviations are independent of each other, it is necessary that also a pairwise test indicates independence. Furthermore, the turning point when stochastic measures increase marks the condition for early stopping without relying on a validation set. This turning point indicates that, while the loss function continues to decrease, the model may start overfitting to the training data by adapting to spurious correlations or noise. The overfitted relations do not seem to be relevant for the validation set as here the stochastic measures do not decrease any more after some epochs.

We provide a further example based on the dataset ETTh2 Zhou et al. (2021) with a similar result depicted in Figure 4. We see based on the chi-square test that, after some epochs, there is already a set of weights based on which the residuals are clearly stochastically independent of the input. Similarly, we see that the mutual information is close to the value generated by the permutation test, supporting the chi-square results. Furthermore, the increase of the chi-square and the mutual information on the validation set for later epochs might indicate an overfitting since the minimum based on the $L^1$-loss function, which further decreases over epochs, does not necessarily coincide with the maximum of extracted deterministic relations defined by a stochastic measure, see also Section 4.2.

## 4.4 Time series classification

In this section, we apply our framework on time series classification (TSC) problems. We remark that for ordered classes, we can apply the framework where the differences between the discrete output values and the ground truth model the residuals. The deviations from the ground truth in the nominal case are defined as follows. The random variable $\theta$ modeling the deviation of the model classification from the ground truth, takes the value of the correct class in case the output of the model is not correct and -1 otherwise. This definition allows us to measure the information content between the correction of the model (representing the model deviation) and the input. We apply our framework on two subsets of the well-known UCR dataset Dau et al. (2019). More specifically, the dataset DistalPhalanxOutlineCorrect shown in Figure 5 is a binary classification problem from time series data and the dataset ElectricDevices shown in Figure 6 consists of seven classes. We relabeled the classes always starting from 0 until all classes are labeled accordingly. The model architectures used are the Nonstationary Transformer (NSTs) Liu et al. (2022) and the MLP.

In this case, the parameter $\rho$ of Algorithm 1 is set to 10 to cope with the high imbalance of the classes and the fact that over epochs this imbalance increases since most of the cases are classified correctly and thus the size of the class labeled with -1 for $\theta$ increases. Results including cross-entropy loss, accuracy and the mutual information per epoch are shown in Figure 5 and Figure 6. In these figures, we observe that while the accuracy is increasing, the dependence of the variable $\theta$ with the input $x$ decreases over epochs as it is supposed to since less and less cases are not predicted correctly. In other words, the random variable $\theta$ tends to become a constant function where information gain becomes small or zero, regardless of any input time series. The difference in both cases is the following. From the results of Figure 5, we can conclude that over training epochs we approach a parameter configuration where we cannot reject the hypothesis, at a significance level of 1%, that the input and the model deviations are independent based on our mutual

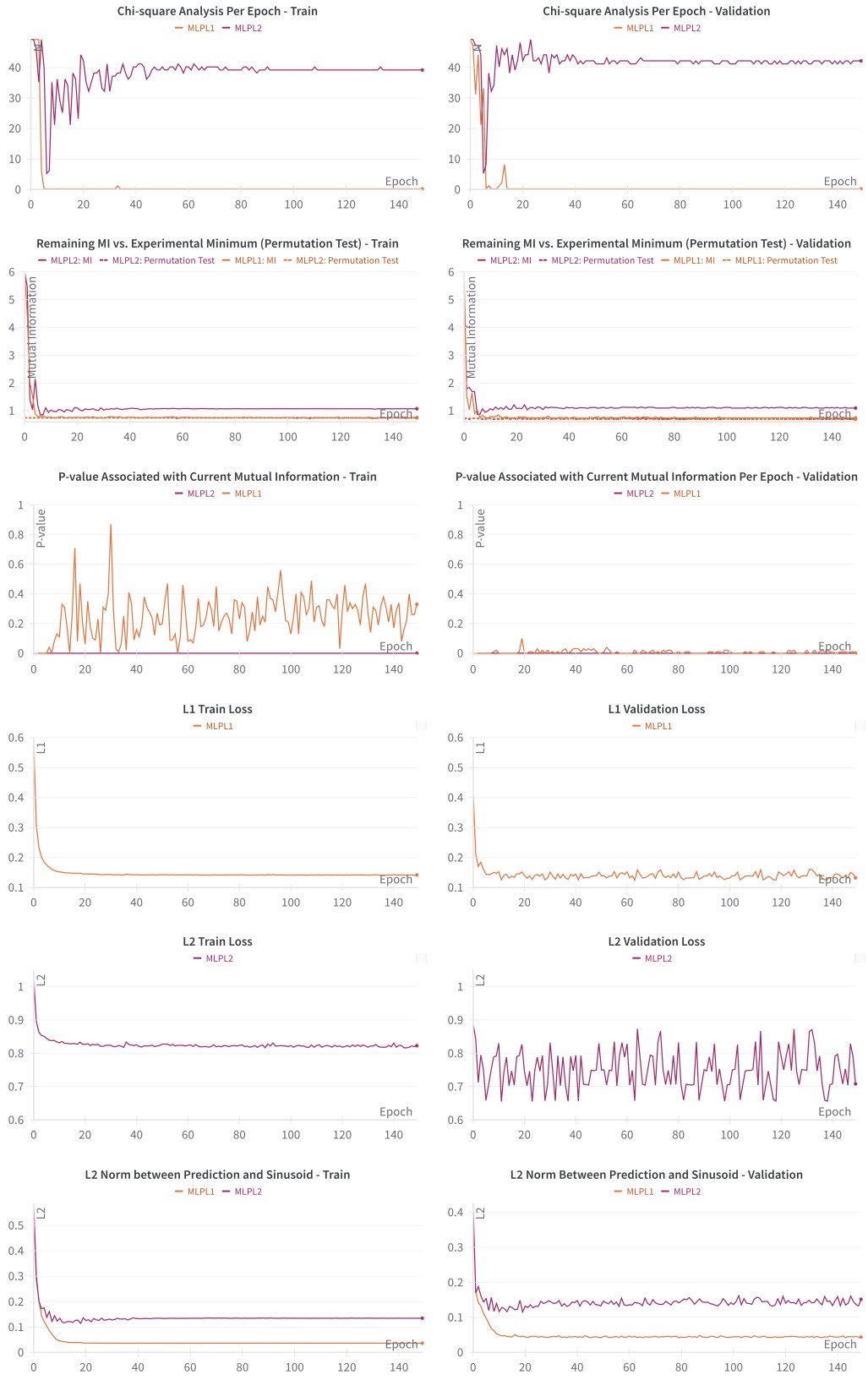

Figure 3: Curves showing stochastic measures of independence and loss function history.

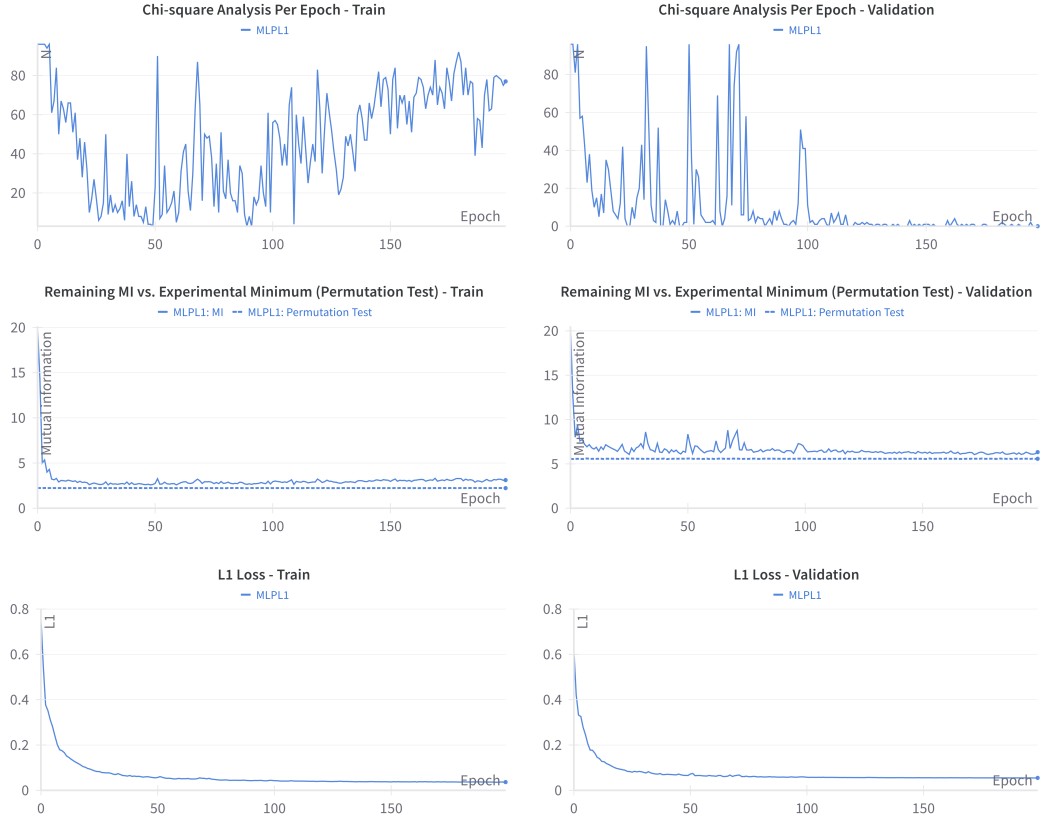

Figure 4: Curves showing values of stochastic measures of independence between input and residuals as well as the loss function history based on the ETTh2 dataset.

information measure. In this case, we do not expect a further improvement regarding the model performance since it potentially captures all deterministic relations. In contrast, in the experiment depicted in Figure 6, our mutual information test allows us to reject this hypothesis, indicating that there are still deterministic relationships to be extracted, which may improve the model's performance.

## 4.5 Detecting distribution shifts

One prevalent issue hindering the advancement of ML models towards higher accuracies is distribution shift, meaning that relations that hold within the training set do not hold on the validation set. A reason could be that the validation set contains newer data and that the underlying dynamics have changed or change over time. Especially several existing works such as (Zeng et al., 2023, Figure 5) and Kim et al. (2021), in particular (Kim et al., 2021, Figure 3) have confirmed this phenomenon, e.g., on ETT1 and ETT2 data sets. This section presents a novel insight into this phenomenon, illustrating how our proposed framework can detect and distinguish such cases from mere overfitting to noise. In a typical training scenario, after some epochs while training loss continues to decrease, validation loss may gradually start to increase. Without prior assurance of the absence of distribution shift, a pure loss function based approach without including stochastic measures struggles to differentiate between overfitting to the noise in training data and (partial) distribution shift due to different deterministic relations between the training and the validation set, as both can lead to similar observations of an increasing loss function on the validation set.

Our framework provides a concise solution. Instead of solely monitoring the loss function, tracking mutual information enables us to determine the types of relationships the model is learning. A decrease in mutual information across epochs indicates successful extraction of information, suggesting that the model is learning deterministic relationships within the training set, and not already overfitting to noise. If it does so as well

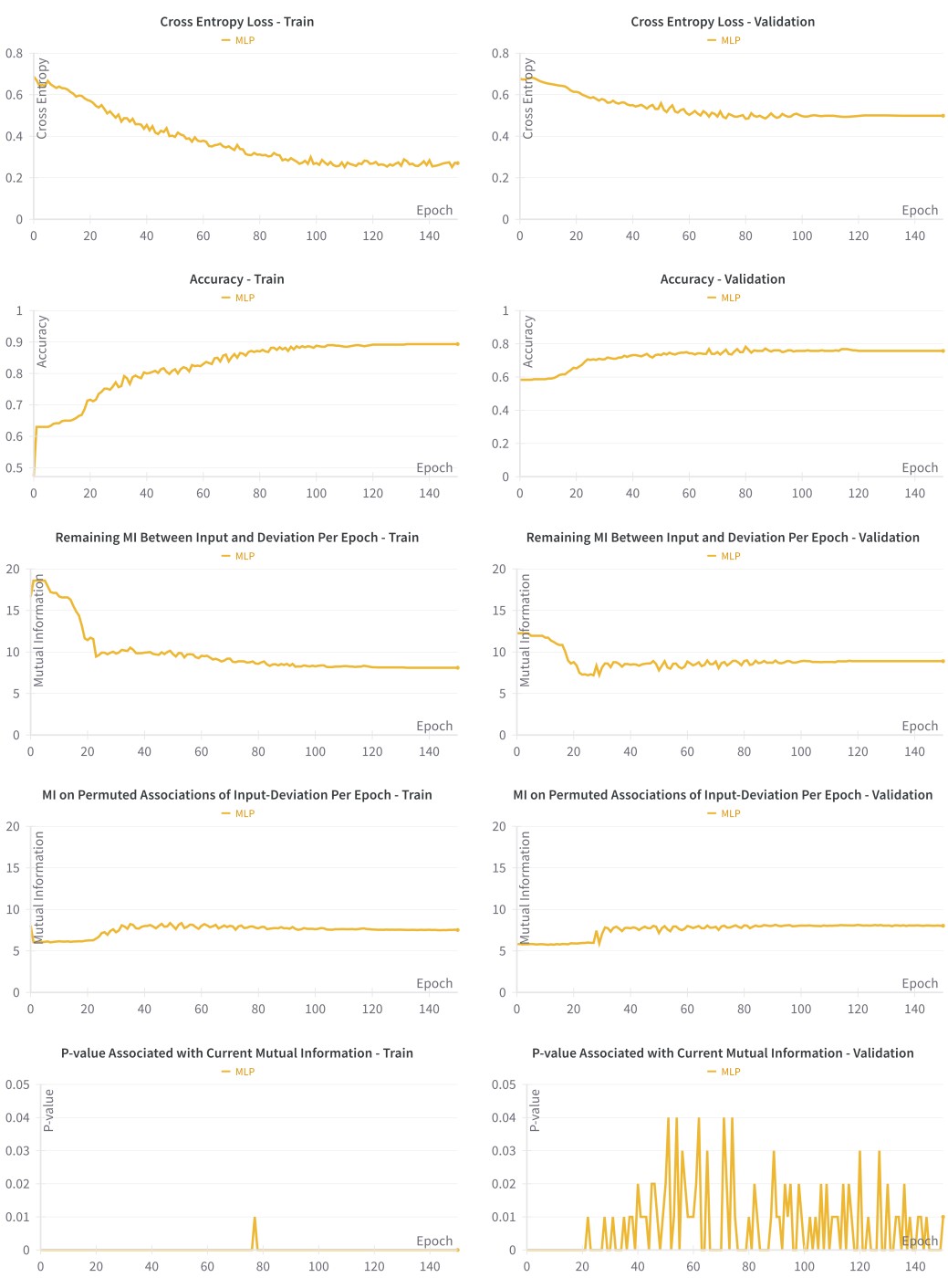

Figure 5: Results on a classification problem showcasing our framework, in particular the definition of the model deviation for nominal target data. Although validation accuracy remains below 0.8, mutual information analysis shows the model deviations are already independent of the input.

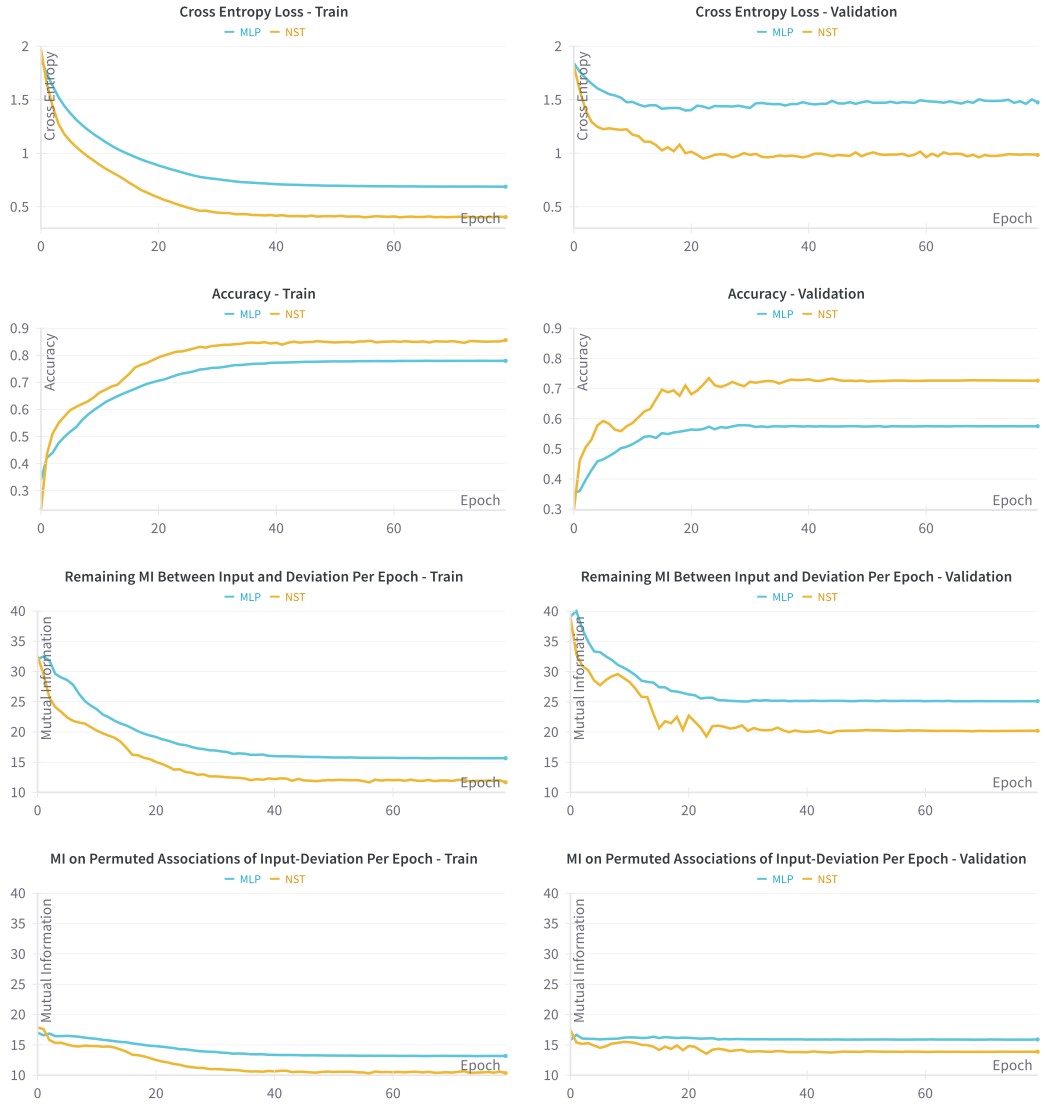

Figure 6: Results on a classification problem showcasing our framework for nominal target data. The p-value for mutual information always remains zero in this experiment for both the training and validation sets indicating that the achieved accuracy is still improvable.

on the validation set until input and deviations between model and ground truth are independent, then we can stop the training process since the model might have learned all deterministic relations on the training set that hold true for the validation set, too. In that case, further training might cause an overfitting to relations not present in the validation set. Similarly, if there is no significant reduction in mutual information despite decreasing training loss, it may indicate overfitting to the noise in the training data, as the model is fitting to the unpredictable elements of the target which share no mutual information with the input.

In our synthetic dataset, the deterministic relations are identical in training and test set by construction. In Figure 3, we observe that during the training based on the $L^2$-loss function that the mutual information increases on training and validation set after reaching their minimum within a few epochs simultaneously. It may mean that the model fits noise rather than the actual sinus function since the norm of the difference between model prediction and sinus function increases simultaneously while the loss on the training set further decreases.

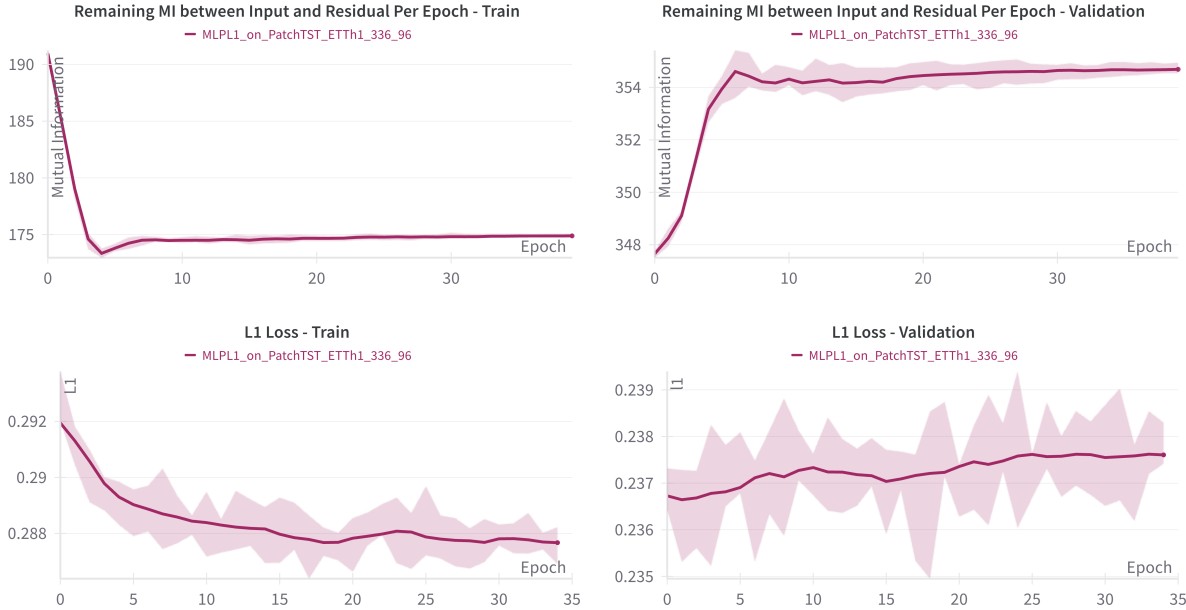

Figure 7: Distribution shift experiment on the ETTh1 dataset. The experiment is repeated for six different random seeds. The shaded area depicts the minimum and maximum over seeds and the solid line shows the average.

In case the deterministic relations in the training data may (partially) not hold true for the validation set, it may lead to an increase of the loss function value and potentially mutual information over epochs on the validation set, while mutual information on the training set decreases, as we see in Figure 7. In this figure, the deterministic relations learned on the training data do not cause a fitting output of the model on the validation set. In contrast, please see Figure 3 where mutual information between input and residuals on training and validation set demonstrate the same behavior. The results in Figure 7 show the learning curves when fitting an MLP with an $L^1$-loss function on the residuals of a PatchTST model Nie et al. (2022) with the best setting they proposed on the ETTh1 data set where the input length is 336 lags and we predict a target length of 96 lags. Details about the combination of both models can be found in Section E of the Appendix. We remark that this experiment also showcases the application of our proposed framework to multi-dimensional output.

## 5   Discussion

This section discusses the assumptions and limitations of our approach, followed by potential applications.

**Assumptions and Limitations:** The implemented approach tests the pairwise relationship between input and output features. We acknowledge that there is a difference between pairwise stochastic independence and mutual stochastic independence when dealing with more than two random variables (Gallager, 2013, Section 1.3.4). This implies that more information might be obtained by considering multiple input features simultaneously and testing their relationship with the output, rather than testing pairwise relations between individual input and output features. However, the full consideration, instead of a pairwise testing, scales exponentially in terms of the computational costs. Consequently, we are aware that the current pairwise approach, which is computationally cheap compared to the full approach, cannot provide in general a full statement that no information is left to learn. In this regard, the current approach can only be used as an additional metric to evaluate if training is complete and further iterations might not provide a further improvement. This could be the case if a low pairwise measure of the relation between input and output coincides with a little or no improvement in the loss function. Pairwise test is a specific case of a full consideration. Consequently, it holds: a pairwise test indicating deterministic relations between inputs and

residuals/model deviations implies that there is still information left that the model can learn. An analogous framework involves using gradient descent as a convergence criterion in optimization. Gradient descent is widely used in deep learning and has been successful despite in general lacking sufficient conditions for global minima convergence. The gradient provides necessary conditions for convergence, and only under certain circumstances, it can be a sufficient condition. Nonetheless, it produces useful optimization results and keeps computational costs manageable, similar to our pairwise stochastic measures. In this regard, if all deterministic relations are learned, it is necessary that the measures for independence based on a pairwise test indicate independence of the input and the residuals or the model deviations.

It is left to investigate under which conditions a pairwise consideration is sufficient to test for total stochastic independence of input and output. Furthermore, we remark that our framework is not limited to a specific choice of stochastic dependence/information measures and also our git repository is designed that new measures can be quickly included in a modularized manner.

Due to the fact that any approximation or estimation of a measure, such as mutual information, may not be sufficiently accurate Czyż et al. (2023), it is prudent not to rely only on a single estimator but rather to use multiple ones. We achieve this by considering both an approximation for mutual information and the chi-square test. The generality of our framework allows for the use of different measures for independence where we can also consider other features like computational runtime apart from accuracy. For example, the test statistic of the chi-square test of independence provides a theoretically known distribution enabling quick hypotheses testing to determine if based on the data corresponding variables are independent.

For time series, we would like to remark, even if a method shows (full) mutual independence between the input (history of the time series) and the output (prediction target), that this result does not imply that the time series is not predictable. It just indicates that with the given input/provided information, the time series is not predictable. Maybe with other features that are related to the quantity measured as a time series, there is a deterministic relation that can be used for predicting the time series.

**Alternative of model deviation for nominal data:** One further option to model deviations of a model from the ground truth in the case of nominal target data could be a multi-dimensional random variable that models the difference between the actual probability distribution (e.g., 1 for the correct class, 0 otherwise) and the predicted distribution from the model over the classes to be predicted. If the input variables and the difference of the distributions, which serves as the model deviation, are deterministically related, then the model deviation could be represented as a multi-dimensional variable (one variable for each class) to measure the predictability. In other words, the classification problem could be seen as a multi-dimensional ordinal data scenario.

**Extension to unstructured data:** For models that extract information from an input like videos, images, sound, or text, the input data needs to be transformed into a numerical representation that shares information with the ground truth. An example are the pixels in a figure where a small object of interest moves within a big blank picture or tokenized text, where the position of a word can vary while not changing the meaning. We could optimize first layer(s) to have the highest mutual information with the ground truth that is to be predicted analogously to Chen et al. (2016) or Brakel & Bengio (2017) to optimize the encoding of the unstructured input into numerical vector representations.

It is the task of an encoder to find a numerical representation (vector embedding) from an input signal that maximizes information content, e.g., in the form of mutual information or stochastic dependence, with the ground truth to be predicted. Thus, the encoder is supposed to extract the relevant information from the (unstructured) input regarding the prediction target. This extraction can be seen as a definition for "relevant information". In that sense, training a model can be two-staged. First, we could train an encoder part to maximize the mutual information or stochastic dependence between the vector embedding (output of an encoder) and the ground truth. Using a gradient-based method for this purpose requires differentiable estimators of mutual information (e.g., Franzese et al. (2024)) or any other differentiable definition of stochastic independence. To consider the example above, the role of the encoder may be to extract the moving object from the figure, independent of its position. In the second stage, we train the decoder part, which is supposed to make the prediction based on the vector embedding, with some loss function to minimize errors between model output and ground truth. Another option is to optimize

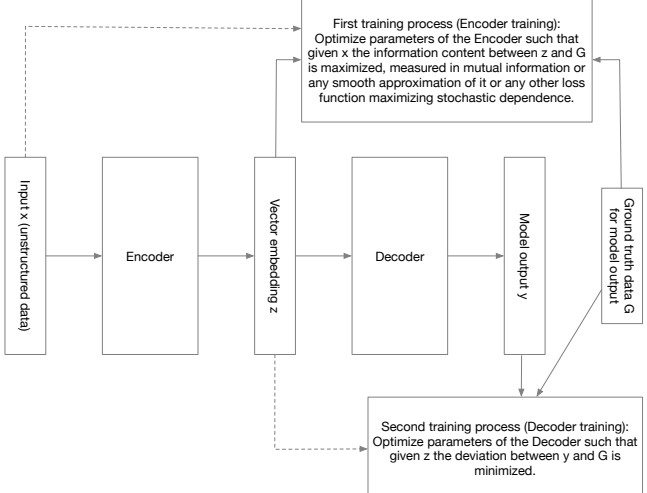

Figure 8: Encoder-Decoder architecture for a separated training of the encoder and the decoder based on stochastic dependence.

the decoder for minimizing mutual information or the stochastic dependence between the model deviation (decoder's output and ground truth) and the vector embedding (encoder's output). However, this approach also requires a differentiable function for defining stochastic independence or mutual information. We remark that separating the training of the encoder from the decoder's training might be important in case we use a loss function based on reducing stochastic dependence between the encoder's output and the decoder's deviation from the ground truth because if the encoder's representation does not contain much information about the ground truth, the decoder's deviation from the ground truth is not predictable given the encoder's representation and would thus indicate training success. Given the fixed dimension of the vector embedding, we can compare different encoder architectures and define the best according to, e.g., the highest (normalized) mutual information between vector embedding and prediction target. Analogously, the best decoder can be defined as the one having the lowest (normalized) mutual information between the encoder's vector embedding and the model deviation of the decoder's output and the ground truth. The mutual information measure can be replaced by any measure for stochastic dependence. The corresponding encoder-decoder architecture is illustrated in Figure 8.

With the definition of model deviation, see Subsection 4.4, subparts of the model, such as layers, in particular those associated with representing the unstructured input as a numerical vector representation, can be analyzed regarding their contribution to find the next token in the context of large language models. Moreover, investigating the dependence of the model's layers/substructures with the model deviation can help to answer the question how small a model can be and where most information is learned/extracted to make large language or even multimodal models more efficient. For more details, see the next paragraph about cutting down models. Since distributions over classes represent the probability for corresponding outputs, it might also be an option to work on distributions directly to investigate the dependence between the layers and the model deviation as described above under "Alternative of model deviation for nominal data", although the dimensionality of the problem increases accordingly.

Another advantage of calculating stochastic measures between representations within a (pre-)trained model and the ground truth belonging to a specific prediction task such as classifying input or predicting numbers, is to find out which representations have the highest dependence (information content) on the corresponding task, which might make the inference more efficient by only utilizing the most contributing parts.

**Cutting down models to structures extracting most information:** If we train a model with usual loss functions without any additional optimization for mutual information in specific layers, we could identify the best structures within a trained model that extract most information and which subparts only contribute in a minor way to, e.g., an embedding. For this purpose, we zoom more in a model's architecture to analyze

the model itself regarding its subunits (e.g. layers, embeddings, attention mechanisms, etc.). The approach is to apply mutual information similar to the scenario of multi-dimensional input and output. The output can either be filled with the data to predict or the output of some other subunit, such as an embedding, upstream towards the output of the total model where the output of several subunits is combined. If there is no contribution or only a minor one, we could cut the corresponding subunit out.

By ranking subunits with such measures, we have a clear procedure what subunits to prune to make a model more efficient, instead of randomly selecting some for cutting out before retraining the model. The training could start from the current model parameters to just fine-tune the remaining layers. Training from the scratch is also possible but could delete a lot of parameter values that are still valid. The procedure can be iteratively repeated and even to that point where model size is balanced against a (small) drop of accuracy. Our framework can additionally help to find a lossless pruning similar to the Related Work section about the information bottleneck by testing if the stochastic measures between input or any representation of it and the model deviations get worse after a pruning. Such investigations might facilitate an understanding about the problem-specific relevant structures and keeping model sizes efficient without lowering their accuracy. As an example, we could test how large, e.g., large language or multimodal models need to be and which structures extract the most information, similar to Liu et al. (2024). Further, we remark that with a measure for self-similarity, like the Pearson correlation, we could probably identify identities, resp., structures that behave similarly, e.g., in a sequential arrangement of layers that have the same output behavior, where all the layers except the last one should be considered as less relevant since they only direct information through. A similar work is done in Gromov et al. (2024) to identify layers with a similar behavior that can be pruned. However, with mutual information, besides considering similarly behaving units, we can also investigate the single subunits' contribution within a branch structure to the part where the branches come together.

If a training fails, we could also use such methods to test which structures fail. An example is a cascade of layers with one layer where input and output do not share any information anymore. The disruption of routing information could mean that one layer might not be well trained and is like a constant function, which may delete important information.

## 6 Conclusion and Future Work

In this work, a framework for measuring predictability of input-output relation was developed. Furthermore, it was shown how the information extraction of an ML model from this input-output-relation can be measured. Moreover, it was demonstrated how the corresponding stochastic measures for predictability can be used to extend the current definition of model convergence and training success. The total framework was showcased with time series prediction and classification on synthetic and real world datasets.

The presented framework provides measures to evaluate the existence of deterministic relations that a model can extract and how successful a model has been in extracting them. All these measures consider the triple of input, model output and ground truth to determine whether the model deviations from the ground truth are independent of the input. Promising further research could involve developing a differentiable loss function based on such stochastic measures to fit the model to the deterministic relations directly, sorting out unpredictable parts like noise. In contrast, there are loss functions that only consider a tuple of ground truth and model output, such as $L^1$, $L^2$, cross entropy or KL-Divergence that minimize the deviation between the model's prediction and the ground truth without differentiating if a data point is mainly influenced by, e.g., noise or the deterministic relations.

According to the presented framework the mutual information between input and the model deviations might serve as such a loss function sought. However, for the implementation, there are some challenges left that need further research to overcome them. One challenge might be that the current implementation is not differentiable since the discretization procedure of Algorithm 1 is not differentiable when a boundary of a bin is crossed by a data point. To overcome the challenge of the non-smoothness issue, smooth approximations of mutual information, such as Belghazi et al. (2018) and Franzese et al. (2024) could be utilized. However, these approximations can become computationally too costly since the mutual information estimator model might have to be retrained after every epoch of the prediction model to approximate the mutual information between input and model deviations. Another challenge is the potential instability in estimating mutual information,

as reported in Choi & Lee (2022). Such an inaccuracy in the estimation of the mutual information utilized as a loss function could cause divergence of the optimization procedure and hinder the model's capability to extract more deterministic relations differentiating them from unpredictable parts like noise given the input data. Further research for a smooth and computational cheap approximation of mutual information is promising to focus the model training on the deterministic relations encoded in the data to the degree the corresponding architecture is able to extract.

Such a framework can also benefit applications outside machine learning where parameters of function are optimized to map input to ground truth data as closely as possible. One example is in the field of optimization and data-driven modeling, such as best parameter fitting Raue et al. (2015); Crouch et al. (2024) since such a loss function might improve the parameter finding with regard to making their value choice more robust against noise. In this regard, the input variable is the time $t$ in case the modeling is done with ordinary differential equations (ODEs) or the space $x$ and optionally the time $t$ if the modeling is done with partial differential equations (PDEs). Additionally, we could have some external stimuli $u$ as further input variables, such as the effect of drugs in biology or the heating of a work piece. The corresponding quantities $y$ that are to be described by the model are the model output (ground truth). Thus, we ideally have that, e.g., $f(t, x, u)$ approximates $y$ as well as possible where $f$ solves the underlying system of differential equations. With such an approach, we can test if $y$ is predictable given the input data at all. After that, we can see if the underlying model (consisting of, e.g., ODEs or PDEs) fits the data, meaning the model is able to capture all the deterministic relations within the data or if the model should be extended, e.g., with further interactions between the system states, to capture the information hidden in the data. The successful extraction is the case if we cannot reject the hypothesis that the input variables and the residuals are independent.

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

# A   Further explanation on discretization and empirical probability estimation

This section aims at providing further explanation on the discretization schema and the details of the calculation of the empirical marginal and joint probability functions. We exemplify the procedure how we compute these entities based on Algorithm 1 with our time series data.

We first consider a sufficient number of input-output windows as depicted in Figure 9a, each having the same total length, which is the sum of the input length and the output length. In each window, the $i^{th}$ point in the input is considered as the random variable $x_i$, while the $j^{th}$ point in the output is treated as the random variable $y_j$. This is illustrated in Figure 9b. For each window, we get one realization of each random variable $x_i$ and $y_j$.

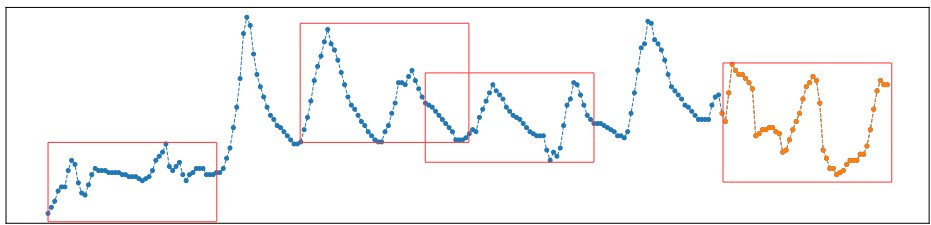

(a) Several input-output windows randomly selected from a dataset.

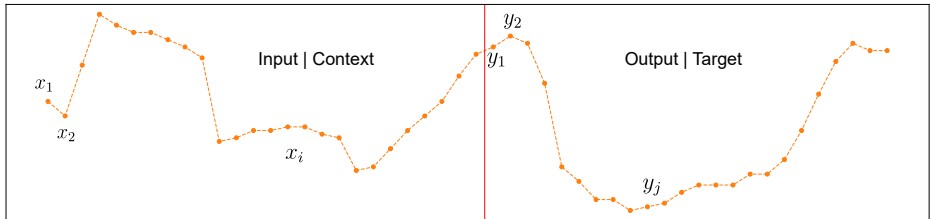

(b) Details of a selected window including input and output modeled as random variables.

Figure 9: Preparation of the data for the statistical tests.

Once we have a sufficient number of realizations for each of the random variables, we apply Algorithm 1 to discretize the co-domain of each random variable. Each bin of the discretized co-domain of a random variable defines the event that the value of this random variable is between the boundaries of that bin. Algorithm 1 can be viewed as performing adaptive binning and histogram construction for each of the random variables. In this context, each event $z_{v_i}^k$ represents the random variable $v_i$ taking a value within the interval defined by the upper and lower bounds of the $k^{th}$ bin after discretization. Consequently, the output of this algorithm includes the upper and lower bounds of the bins as well as the frequency with which the random variable $v_i$ falls into each bin.

To compute the joint probability distribution of random variables, modeling the input of a model, the output of a model, the residuals or the model deviations, we calculate the frequency of the joint events where each random variable is within a bin. For this purpose, each joint realization is placed into the corresponding 2D cell (or cuboids in higher dimensions where more than pairwise combinations of random variables are considered), whose bin boundaries are predetermined by the outer product of the marginal bins. This process is illustrated in Figure 10. The probability for marginal and joint distributions is obtain by dividing the corresponding frequency by the number of realizations (in this case the number of windows).

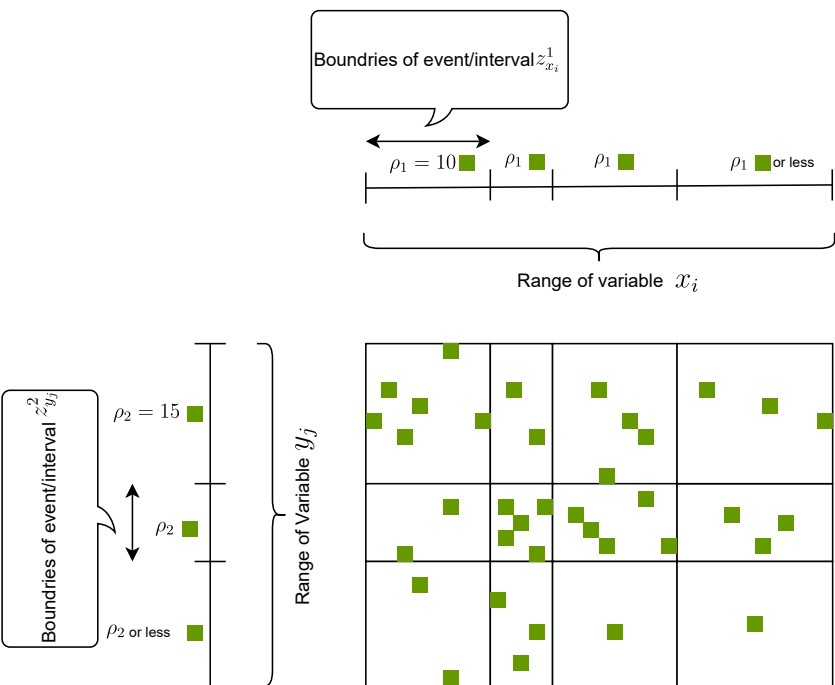

Figure 10: Computation of the joint histogram for pairs of random variables. Each joint realization of $x_i$ and $y_j$ observed across all input-output windows is represented by a green square and is placed in the corresponding 2D bin. The boundaries of the 2D bins are determined by the outer product of the boundaries for each random variable. Discretization of each random variable follows the procedure outlined in Algorithm 1. Note that the marginal distributions are discretized to have equal numbers of data points per bin (where there might be an exception for the last bin according to our realization or if there are many points with the same value), the joint distribution can deviate arbitrarily much from a uniform distribution. In this figure, $\rho_1$ and $\rho_2$ are illustrated with different values ($\rho_1 = 10$, $\rho_2 = 15$) for the sake of generality. Consequently, each bin of the marginals here contains 10 or 15 green squares, respectively. However, in our experiments, these values are set to be equal.

## B    Convergence and bounds on estimation of mutual information

In this section, we discuss the convergence property of our approximation of mutual information based on Algorithm 1. For this purpose, we focus on Darbellay & Vajda (1999) which describes a convergence result for mutual information based on an adaptive discretization scheme of the co-domains of random variables. We show that our method fulfills the same conditions as used in Darbellay & Vajda (1999). These conditions are that the discretization scheme decouples the process of decreasing bin width and approximation of the true density with the empirical one by increasing the samples in each bin. Once we have shown that our approach fulfills the same conditions, we can follow the reasoning in Darbellay & Vajda (1999) to prove convergence of the mutual information based on our framework to the true mutual information between random variables. For this rigorous analysis, we have to equip our algorithm by a further threshold keeping a bin at a minimal width. In this case of reaching the minimum bin width, the condition of the minimum number of points per bin is suspended for this bin. These two further items are used for the theoretical consideration to apply it several times for the convergence process, however, do not come into action for its application.

First, we show how we can ensure the decreasing bin width by including more samples. The data, prediction and therefore residuals/model deviations from the data are finite. The reason for the boundedness is that the

model is a continuous function and thus bounded input is mapped to a bounded output. Consequently, as these entities constitute our random variables, the marginal and joint distributions are of bounded support. Therefore, adding more data points would lead to an arbitrarily smaller bin width when a fixed amount of data points per bin is considered, unless the marginal probability is exactly zero. However, in this case the joint probability is also zero resulting in the fact that these events do not contribute to the total mutual information. Consequently, we can discard these bins and without loss of generality, we require non-zero probability density within the domain. By iteratively including more data points, our discretization scheme provides decreasing bin width until the minimum bin width is reached.

The next requirement for applying the reasoning of Darbellay & Vajda (1999) is to show that putting more samples increases the data points per bin to approximate the true probabilities by the empirical ones to any arbitrarily small error. Due to the action of the minimum bin width, putting more data points solely increases the number of data points per bin since we have a non-zero probability. Consequently, the minimum bin width ensures a decoupling of the limit process of getting a finer discretization of the domains of the probability functions and more data points per bin to make empirical distributions converge to the true ones.

We remark that the main focus of our work is to show that measures quantifying the stochastic independence between input and output, and thus residuals or model deviations, are useful to monitor model convergence. Examples are the chi-square test of independence or the mutual information. In particular, for the mutual information there are many different implementations that have their pros and cons. For some overview, please have a look into (Czyż et al., 2023, Appendix), including convergence results for the estimators.

## C   Extra notes on Subsection 4.2

During the training of models in experiment 4.2, we observed an intriguing phenomenon. Specifically, there exists a model trained with an $L^2$-loss function that not only performs better in the $L^2$-metric (MSE) but also achieves comparable performance in the $L^1$-metric (MAE) to a model trained with an $L^1$-loss function on the validation set. Under normal circumstances, one would naturally prefer the $L^2$-loss-trained model in such a scenario.

However, our analysis using the proposed framework reveals that, even in this situation, the $L^1$-loss-trained model is superior. To validate this finding, we examined the distance to the true underlying relationship, which is sinusoidal function in this experiment. As shown in Figure 11, our framework effectively highlights the importance of model selection and demonstrates the advantages of our proposed metric compared to traditional loss functions.

## D   Data description

**Datasets**   Here is a description of the datasets used in our experiments:

(1) *ETT* Zhou et al. (2021) contains seven features including the oil temperature and six power load feature. ETTh indicates the ETT data with a granularity of 1-hour-level and ETTm indicates the ETT data with a granularity of 15-minutes-level.

(2) *Weather*[1] is recorded every 10 minutes for the whole year 2020, and contains 21 meteorological indicators such as humidity and air temperature.

(3) *Nasdaq* dataset consists of 82 variables, including important indices of markets around the world, the price of major companies in the U.S. market, treasury bill rates, etc. It is measured daily, having a total of 1984 data samples for each variable. We set the corresponding input length as 60 similar to Kim et al. (2021). In this work, we conduct our experiments on DE1 variable.

(4) *ElectricDevice* is a subset of the UCR time series classification dataset. It consists of seven different classes, each representing a specific type of electric device which is to be predicted from the time series.

---

[1] https://www.bgc-jena.mpg.de/wetter/

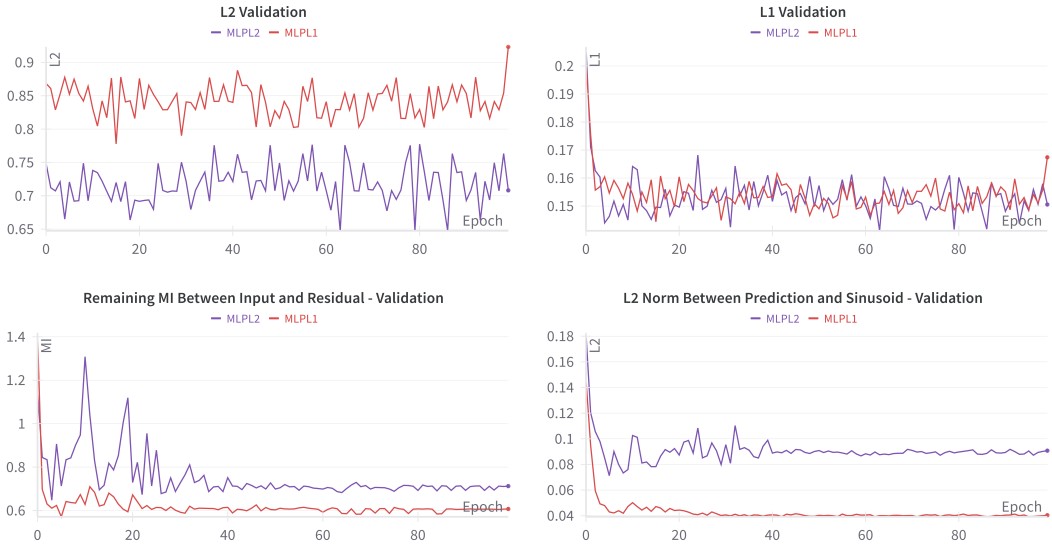

Figure 11: Curves showing mutual information as a stochastic measures of independence between input and residuals, along with the loss function history. Traditional metrics such as $L^1$- and $L^2$-distance, when measured on noisy validation data, are not as indicative as mutual information (MI) in identifying which model is closer to the underlying clean relationships in the data (e.g., the sinusoidal pattern).

(5) *DistalPhalanxOutlineCorrect* is a subset of the UCR time series classification dataset. This dataset focuses on the classification of outlines of distal phalanx bones from time series.

# E  Model stacking architecture and numerical experiments

With the stacking of models, we provide further evidence that information left between input and model deviations can be used to correct the corresponding predictions.

## E.1  Stacking architecture

Different properties of a model may influence its capability of extracting deterministic relations. Consequently, in this part, we provide an architecture to combine different models with different properties to systematically extract deterministic relations. Roughly, models are stacked together where all models get the same input, however, try to learn only what the models in the stack of models so far have not extracted regarding deterministic relations.

We need to differentiate two cases. The first case is where the target is ordinal data. Here the random variables modeling the deviations of the model output from the ground truth are defined by the difference between the model output and the ground truth. In this case the model deviations are termed as residuals. In case where the target is nominal data, the random variables that model the deviations of the model are defined as follows. In case the prediction of the model is not correct, the random variable modeling the deviation of the model from the ground truth takes the value of the ground truth class label. In case the model is right, the corresponding random variable takes a value that does not represent a model class, e.g., a negative integer.

Next, we explain the stacking procedure in detail for both cases. We are given the data $\{(x_i, y_i), i \in I\}$ where $x_i$ is the vector-valued input, $y_i$ the corresponding vector-valued output and $I$ is a finite subset of the natural numbers $\mathbb{N}$. With $x$ and $y$, we denote the corresponding vectors of random variables that take the corresponding values for a given $i$. First, we check if $x$ and $y$ are stochastically dependent on each other. If

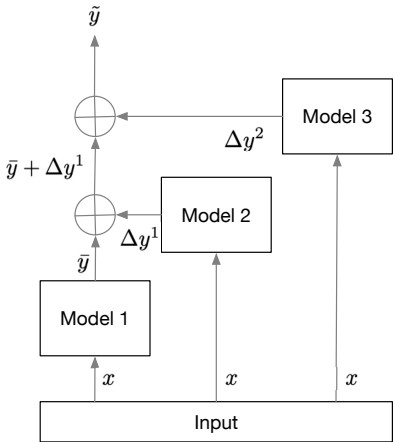

Figure 12: Stacking architecture for problems with ordinal target data.

yes, we train the first model/part of the model stack to best fit the prediction $\bar{y}$ to $y$ given $x$. With $\bar{y}$, we denote a (vector-valued) random variable taking the values $\bar{y}_i$ for the corresponding $i \in I$.

For ordinal target data, the procedure looks as follows. If $\Delta y^1 := y - \bar{y}$ is not independent of $x$, there is still some information left that can be extracted. Consequently, we fit a model that may have properties different to the current model to learn these relations between $x$ and the residuals $\Delta y^1$. In other words, the purpose of the second model on top of first model is to correct the prediction of the first model. The output of the two models is then the prediction of the first model $\bar{y}$ plus the correction $\Delta y^1$. In the next iteration, we test the residuals $\Delta y^2 := y - \bar{y} - \Delta y^1$ for stochastic dependence on $x$. This procedure can be repeated until the corresponding residuals are stochastically independent of $x$. We call this procedure stacking of models and can be generalized as follows such that a new model on top of a stack learns to correct the prediction of the previous stack. The procedure is illustrated in Figure 12.

To generalize, we define

$$\Delta y^j := y - \sum_{k=0}^{j-1} \Delta y^k$$

where $\Delta y^0 := \bar{y}$ for any $j \in \mathbb{N}$. In a purely deterministic scenario, we would extend the stacking until $\Delta y^j = 0$. In a real scenario where there are unpredictable parts in the data, our definition of no further improvement possible is that $\Delta y^j$ is stochastically independent of $x$ based on a statistical test or measure. The random variable representing the output of the total stack is denoted with

$$\tilde{y} := \sum_{k=0}^{j-1} \Delta y^k,$$

taking the corresponding values $\tilde{y}_i$ upon the input $x_i$ for $i \in I$, where $j$ is the smallest number such that $\Delta y^j$ is stochastically independent of $x$. Consequently, the stacking stops if $y - \tilde{y}$ is independent of $x$. This definition also works for discrete ordinal data, where the differences, resp., residuals take only discrete values. We remark that this architecture does not require more inference time since each model of the stack does not depend on the output of other layers but all perform their inference on the same input data and thus can be run in parallel.

For nominal target data (the difference between class labels has no meaning), the stacking architecture works analogously except the definition of the deviation of the model from the ground truth is different, please compare with Figure 13. The deviation of the model output from the ground truth is defined by a random variable $\theta^j$, $j \in \mathbb{N}$ that take the value of the correct class in case of a wrong prediction from the model/stack below and a value that does not represent a discrete class of the ground truth in case the prediction of the model/stack below is correct. If $\theta^j$ is independent of the input $x$ for one $j$, we can stop the stacking of more

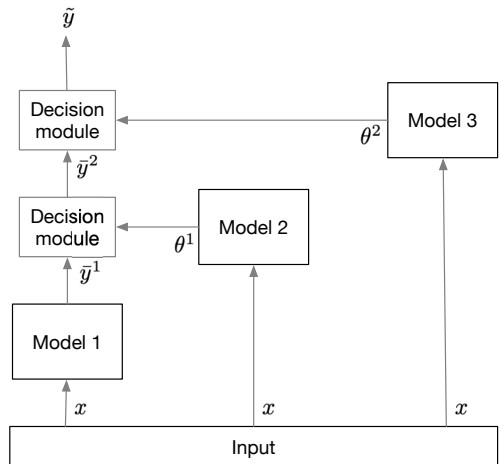

Figure 13: Stacking architecture for problems with nominal target data.

models. As long as $\theta^j$ is dependent on the input $x$, where $\theta^1$ is based on the predictions of the first model, there is a deterministic relation that another model can learn to predict $\theta^j$, which models a correction to the prediction of the model/stack below. In such a case, the stack is extended by another model predicting $\theta^j$. Based on the prediction $\bar{y}^j$ of the model/stack below and the prediction for the corresponding $\theta^j$, we can decide during inference which prediction to take. In case of $\theta^j$ equaling a class label, we take the value of $\theta^j$ predicted by the corresponding model subsequent upstream in the stacking architecture as the value for $\bar{y}^{j+1}$. Otherwise, i.e. $\theta^j$ equals a number not representing a class label, the value of $\bar{y}^{j+1}$ equals the one of $\bar{y}^j$. The final prediction of the stack is denoted with $\bar{y}$, which represents a the vector-valued case as well. We remark that the co-domain of the random variable $\theta^j$ is not one-hot encoded but the models output $\bar{y}^j$ should be due to the nominal character. However, in terms of the random variable $\theta$, since there is a bijection between the co-domain of this random variable and the corresponding one-hot encoding, the information content or statistical independence between the random variable $\theta^j$ with its one-dimensional co-domain consisting of integer numbers and the input also exists and is supposed to be equal to the one-hot encoded version of $\theta^j$.

One greedy implementation of both stacking concepts above is to combine models randomly and check that after each new model $\Delta y^j$ or $\theta^j$ has less information with the input than $\Delta y^{j-1}$ or $\theta^j$ has with the input to ensure benefits of the new model on top of the stack. This is a test that the new model successfully extracts remaining information and doesn't do guessing rather than really filling in what is missing in terms of deterministic relations and thus provides a well-working correction.

### E.2 Stacking of models systematically extracts information and improves prediction

We have seen in this work that the loss function has an influence on the information extraction depending on the noise, see in particular Section 4.2. We investigate in this section how different model properties, like the architecture or hyperparameters, contribute to capturing different kind of information and how such differences could be systematically combined to extract all available deterministic relations in a dataset. We present a general architecture that is supposed to combine the capabilities of different models as described in the Section E.1. Further, this section showcasing the function of the stacking architecture provides evidence that dependent input and model deviations indeed contain information to correct the output of the former model.

We showcase the efficacy of our framework with a real-world time-series datasets. The dataset is a Nasdaq datasets taken from the UCI repository and the M4 competition dataset Makridakis et al. (2020). More specifically, we take the variable DE1. We train models according to MLP and Nonstationary Transformer (NSTs) Liu et al. (2022) architecture on the dataset that is split according to the ratio of 0.7/0.3 into a training and test set. The input of the models are the past 60 time lags and the output is the next value in the time series (singlestep prediction). To this end, an MLP model trained with $L^1$-loss is chosen as the

| Metrics | Init MI | Init Perm | **Init diff** | Res MI | Res Perm | pv | **Res diff** | Init Chi-square | Res Chi-sauare |
|---|---|---|---|---|---|---|---|---|---|
| MLP L1-0.5 | 23.038 | 4.167 | 18.871 | 5.256 | 4.617 | 0 | 0.6392 | 60 | 2 |
| MLP L2-0.5 | 23.038 | 4.167 | 18.871 | 5.347 | 4.579 | 0 | 0.7679 | 60 | 8 |
| stacked-0.5 | 23.038 | 4.167 | 18.871 | 4.843 | 4.564 | 0.06 | 0.2798 | 60 | 0 |
| 2nd stacked model-0.5 | 5.256 | 4.617 | 0.6392 | 4.483 | 4.564 | 0.06 | 0.2798 | 2 | 0 |
| MLP L1 | 23.038 | 4.167 | 18.871 | 5.262 | 4.586 | 0 | 0.6758 | 60 | 4 |
| MLP L2 | 23.038 | 4.167 | 18.871 | 5.414 | 4.533 | 0 | 0.8814 | 60 | 5 |
| Stacked MLP | 23.038 | 4.167 | 18.871 | 4.955 | 4.579 | 0.01 | 0.3765 | 60 | 0 |
| 2nd stacked model | 5.262 | 4.543 | 0.7182 | 4.955 | 4.579 | 0.01 | 0.3765 | 4 | 0 |
| NST L1 | 23.038 | 4.167 | 18.871 | 5.267 | 4.577 | 0 | 0.6901 | 60 | 1 |
| NST L2 | 23.038 | 4.167 | 18.871 | 5.468 | 4.564 | 0 | 0.9038 | 60 | 8 |

Table 2: Comparison of performance metrics for various standalone and stacked models. Initial mutual information (MI), permutation analysis values, and their differences are presented, providing insights into the starting states. Residual metrics, including mutual information, permutation values, and the corresponding p-values in the mutual information framework assessing the independence of input and residual output, are also reported. Additionally, chi-square values for both initial and residual states are included indicating the number of correlated input lags of the time series.

first model and another MLP trained with $L^2$-loss is chosen as a second model in the stack. The weights of the last layer of the models in the stack, except the first model, are initially set such that the output of each model is close the zero. The rationale is that if the prediction from the stack below is correct, only minor corrections are necessary building on the previous predictions. Furthermore, the last layers in models are chosen for weight rescaling since the first layers are usually intended for feature extraction. Additionally, the layer norm operation (dividing by standard deviation) would cancel the scaling to small values, for details about this part please see the Appendix, Section I.4.

As illustrated in Table 2, none of the individual models successfully render the residuals independent of the input. Remarkably, it was only through the combination in stacked models that an increase of p-values was observed, enhancing the overall performance. To provide a more comprehensive comparison, results for NST are also included. We see that in this case, the MLP stack not only extracts more information than the NST but is computationally even cheaper. In order to exclude that the effect is a result of more free parameters, we have included MLPs with about the half of free parameters each indicated by the "0.5". Moreover, we see that only the stacked models provide a non-zero p-value such that only in this case, we cannot reject the hypothesis that input and residuals are independent based on a level of significance of 1%. Beyond mutual information and chi-square test, considerations such as $L^2$-loss and learning curves in Figure 14 further support the empirical evidence that stacking models outperforms their individual counterparts by having a smaller $L^2$-loss function (comparison only valid if the last layer of the stack is trained with the same loss functions, which is in this case $L^2$-loss function, as the corresponding single model). This evidence showcases the capacity of multiple models to learn diverse aspects. However, we remark that a comparison in terms of loss functions and information extraction is tricky as shown in Section 4.2.

We remark that a linear combination of loss functions according to $\lambda_1\|\cdot\|_{L^2}^2 + \lambda_2\|\cdot\|_{L^1}$ with the hyperparameters $\lambda_1, \lambda_2 > 0$ could be an alternative approach to stacking in terms of loss functions. However, the success of that formulation might depend on the right choice of the hyperparameters $\lambda_1$ and $\lambda_2$. Our framework provides an option to choose the hyperparameters accordingly such that most information is extracted from the data, e.g., the mutual information is minimized between the input and the residuals. Please note that finding the best combination or any hyperparameter optimization is not the focus of this work. In the present work, the focus is on showcasing our framework in terms of its potential for applications in various ML use cases.

Next, we show that our stacking framework also works for classification problems. For architectural details, please see Section E.1. In Table 3, we provide numbers based on the ElectricDevices dataset. To further improve the effect of stacking, we see potential when including a corresponding loss function into the training process for optimizing parameters such that the corresponding ML architecture extracts most deterministic

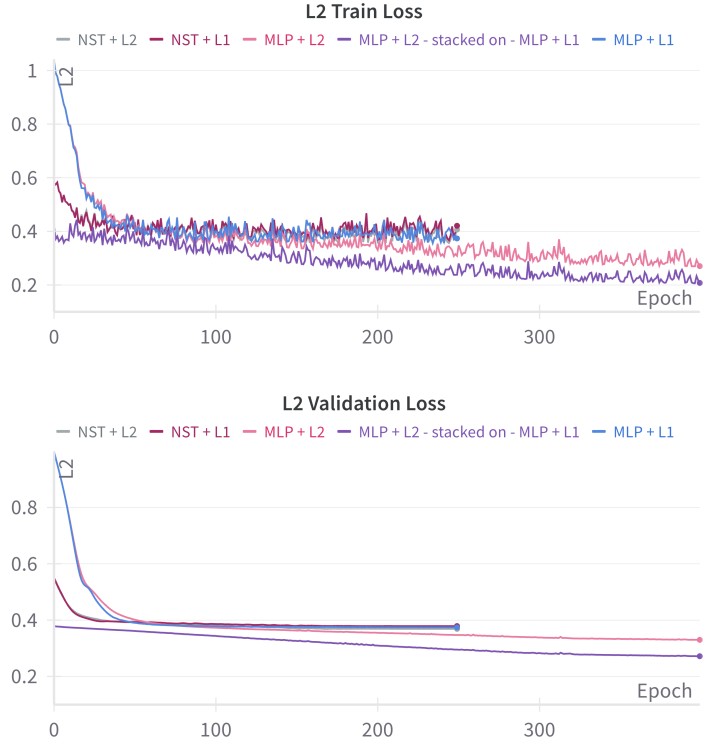

Figure 14: The $L^2$-loss comparison for stacked models and single models.

relations possible. Regarding this topic, please see the Conclusion and Future Work section in combination with the Discussion part "Alternative of model deviation for nominal data".

| Metrics | Init MI | Init Perm | **Init diff** | Res MI | Res Perm | **Res diff** |
|---|---|---|---|---|---|---|
| SVM-rbf kernel | 41.65 | 16.00 | 25.65 | 23.72 | 15.50 | 8.22 |
| SVM-sigmoid kernel | 41.65 | 16.00 | 25.65 | 34.13 | 16.62 | 17.51 |
| SVM-sigmoid + SVM-rbf | 41.65 (23.72) | 16.00 (15.50) | 25.65 (8.22) | 23.23 | 14.41 | 7.82 |

Table 3: Numbers in parentheses show the starting point of the last stacked model. The abbreviation SVM refers to the standard Python sklearn implementation of a support vector machine. The p-values assessing the independence of the input and the model deviation from ground truth alsways remains zero in all experiments. For a description of the meaning of the columns, please refer to Table 4.

The results of the next experiment demonstrating 1-step ahead prediction with the stacking procedure is provided in Table 4 for the Weather dataset, see Subsection D. The split setting from Nie et al. (2022) is used here. Except the prediction length, we use the same training parameters and architecture for PatchTST as used in Nie et al. (2022).

| Metrics | Init MI | Init Perm | **Init diff** | Res MI | Res Perm | **Res diff** |
|---|---|---|---|---|---|---|
| PatchTSTL2 (0.41M) | 3213 | 153 | 3060 | 659 | 177 | 482 |
| MLPL1 (0.59M) | 3213 | 153 | 3060 | 514 | 174 | 340 |
| NSTL1 (3.88M ) | 3213 | 153 | 3060 | 608 | 182 | 426 |
| NSTL1(1.05M ) | 3213 | 153 | 3060 | 641 | 183 | 458 |
| NSTL2(0.426M) | 3213 | 153 | 3060 | 700 | 182 | 518 |
| NSTL1(0.86M)+PatchTSTL2(0.41M) | 3213(659) | 153(177) | 3060(482) | 506 | 181 | 325 |
| NSTL1(1.13M)+PatchTSTL2(0.41M) | 3213(659) | 153(177) | 3060(482) | 516 | 183 | 333 |
| NSTL2(0.53M)+MLPL1(0.59M) | 3213(514) | 153(171) | 3060(343) | 400 | 181 | 219 |
| NSTL1(0.86M)+MLPL1(0.59M) | 3213(514) | 153(171) | 3060(343) | 441 | 182 | 259 |
| NSTL1(0.53M)+MLPL1(0.59M) | 3213(514) | 153(171) | 3060(343) | 442 | 182 | 260 |
| NSTL1(2.05M)+MLPL1(0.59M) | 3213(514) | 153(171) | 3060(343) | 448 | 182 | 266 |
| MLPL1(0.59)+NSTL1(1.13M) + PatchTSTL2 (0.41M) | 3213(516) | 183(153) | 3060(363) | 499 | 182 | 317 |

Table 4: Comparison of performance metrics for various standalone and stacked models on weather dataset. Initial mutual information (MI), permutation analysis values, and their differences are presented, providing insights into the starting states. Residual metrics, including mutual information, values from the permutation test of mutual information are also reported. The corresponding p-values assessing the independence of input and residual output is always 0 in all experiments. Additionally, in the first column, number of parameters for the models is shown in parentheses in millions. In the other columns, the values of the metrics only for the last model of the stack is depicted in the parentheses. In the first column, the model on the left is the first one and the one on the right is the last model in the stack.

In the next experiment, we demonstrate our framework for a multistep prediction. We take the Nasdaq dataset from Kim et al. (2021). We choose the input of length 60 to predict a target length of 30. The result is provided in Table 5 and shows that also in this case, with stacking of models, we can systematically extract the information and gradually make the input independent of the residuals in contrast to the single models. The evidence is provided by the fact that a stack of MLP and NST models provides the smallest mutual information between input and residuals when subtracting the mutual information generated by randomly shuffling the data (column "Res diff" of Table 5).

| Metrics | Init MI | Init Perm | **Init diff** | Res MI | Res Perm | **Res diff** | Init Chi-square | Res Chi-sauare |
|---|---|---|---|---|---|---|---|---|
| MLP L1(2.09M) | 617.48 | 123.29 | 494.19 | 169.45 | 135.11 | 34.34 | 1800 | 332 |
| MLP L2 (3.77M) | 617.48 | 123.29 | 494.19 | 163.53 | 135.36 | 28.17 | 1800 | 182 |
| MLP L2 (4.61M) | 617.48 | 123.29 | 494.19 | 162.82 | 135.47 | 27.35 | 1800 | 177 |
| NST L1(2.67M) | 617.48 | 123.29 | 494.19 | 183.15 | 135.15 | 48.08 | 1800 | 811 |
| NST L2(2.67M) | 617.48 | 123.29 | 494.19 | 179.47 | 135.28 | 44.19 | 1800 | 729 |
| MLP L2(0.67M) + MLP L1(2.09M) | 617.48(169.45) | 123.287(135.11) | 494.19(34.35) | 157.91 | 135.28 | 22.63 | 1800(332) | 55 |
| MLPL2(0.64M)+NSTL1(2.67M) | 617.48(183.23) | 123.287(135.15) | 494.19(48.08) | 156.84 | 135.22 | 21.62 | 1800(811) | 35 |
| MLP L2(0.64M)+NSTL2(2.67M) | 617.48(179.47) | 123.287(135.28) | 494.19(44.19) | 153.15 | 135.25 | 17.90 | 1800(729) | 30 |
| MLPL2(0.09M)+MLPL2(0.64M)+NSTL2(2.67M) | 617.48(153.17) | 123.287(135.28) | 494.19(17.89) | 151.80 | 135.27 | 16.53 | 1800(30) | 20 |
| Avg Ensemble (3.4M) | 617.48 | 123.29 | 494.19 | 170.95 | 135.36 | 35.59 | 1800 | 387 |

Table 5: Comparison of performance metrics for various standalone and stacked models. Initial mutual information (MI), permutation analysis values, and their differences are presented, providing insights into the starting states. Residual metrics, including mutual information, values from the permutation test of mutual information. Additionally, the number of dependent input lags tested by the chi-square test of independence for both initial and residual states are included. In the first column, the number of parameters for the models is shown in parentheses in millions. In the other columns, the values of the metrics only for the last model of the stack is depicted in the parentheses. In the first column, the model on the left is the first one and the one on the right is the last model in the stack. In the last row, we take the average prediction of the three models of the penultimate row when each of those models is separately trained to predict the original ground truth. The p-values assessing the independence of input and residual output based on mutual information remains always zero in all experiments in this table.

**Relation to ensemble learning:** The stacking approach is characterized by progressiveness: We don't assign each model the task of learning the ground truth Wortsman et al. (2022) but rather focus on capturing

what remains unlearned by the stack of previous models. This progressiveness not only contributes to the efficacy of our stacking strategy but also enables each model to focus on specific tasks that are not covered by other models, similar to the concept of gradient boosting. We stack models in a way that the input is given to all models in a stack, and each model attempts to correct the errors left by the preceding models in the stack. In order to enable models to extract different information that the preceding models could not so far, it could be helpful that the models vary in their properties, like the model parameters, the loss function they are trained with or the architecture itself. The similarity to gradient boosting, where the models are trained to predict residuals defined based on the gradient of the loss function with respect to the model output, is that the proposed stacking framework directly uses the model residuals between model stack and ground truth as a reference to train a new model. While for ordinal data, the stacking procedure is very related to gradient boosting with $L^2$-loss, the difference for nominal data is that the stacking procedure runs on the correction of the classification while gradient boosting uses the gradient of the corresponding loss function, like cross-entropy loss. The presented stacking architecture is supposed to showcase our proposed framework to provide further evidence that a model output can be further corrected visualized by measuring stochastic dependence. By considering the stochastic independence of the input and the model deviation, we can set up a further convergence criterion to stop adding further models. This is the case if input and model deviations are independent, as adding new models to the ensemble would not extract more deterministic relations as the input does not contain any information about model deviation.

**Stacking software pipeline:** Apart from finding relevant properties of a model to vary, like size, depth, loss function etc., another aspect is that we assume the stack to be constant once trained and parameters are kept constant while only the new model on top of the stack is trained. It is to investigate in future research if, e.g., training all the parameters of the whole stack after training the new model on top of the stack might benefit the accuracy before testing if another model for the top of the stack is needed. One important application of a pipeline is to facilitate a precise time series prediction related to a concrete single time series and provide this capabilities to a broad audience even outside the ML community that apply the predictions of time series, like weather forecasts. Another use case is improving therapies where their effect depends on time-varying patient-specific parameters. Thus, by taking, e.g., the daily rhythm of gene expression of humans into account, as argued in the concept of chronotherapy Zhang et al. (2014), a precise prediction of the expression levels may improve the effect of therapies and can be one brick for personalized medicine.

# F A remark on the sensitivity of loss functions regarding distracting the model from the ground truth

In Section 4.2, we see the $L^1$-loss function is less prone to the asymmetric noise than the $L^2$-loss function. Since $L^0$-loss function weights all deviations from the real data with the same penalty, we formulate the hypothesis that $L^0$-loss function might perform even better than $L^1$-loss function. However, since the $L^0$-loss function is discontinuous and thus not differentiable at all, a lack of a numerical efficient optimization algorithm capable to deal with the discontinuity of the loss function might hinder its broader application. Consequently, a starting point for further research might be to analyze the performance of loss functions with regard to information extraction that consist of parts cutting off bigger deviations with, e.g., $\min(\max(\tilde{y} - y, -\tau), \tau), \tau > 0$, which can be solved with semi-smooth methods, see, e.g., Ulbrich (2011) or as shown in Breitenbach (2022) by transforming the corresponding optimization problem into a higher dimensional one to resolve the min- and max-function to differentiable functions. Such a loss function, e.g., taking the absolute value or the square of the projection $\min(\max(\tilde{y} - y, -\tau), \tau)$, could be a tradeoff between numerical efficiency and robustness against asymmetric noise or outliers parameterized by the parameter $\tau$. Starting the learning with a big $\tau$ and restarting the optimization with a smaller $\tau$ with the result from the last optimization procedure or decreasing $\tau$ within one optimization run could also accelerate the convergence speed. This procedure could make the prediction more precise with regard to extracting the deterministic relations assuming that bigger determinations come (mostly) from noise given the input data. Our framework can monitor the effect of $\tau$ with regard to extracting the deterministic relations.

# G    Models' settings for numerical experiments

This section of the appendix is allocated to the architecture of the utilized neural networks (NNs) in the experiments. Through this appendix we show the architecture of the MLPs with the number of nodes per each layer inside a list. The number of layers is the same as the length of the list. Unless specified differently, all activation functions are Relu and initial learning rates are 1e-4.

**Section 4.1**:

All NNs are MLPs with Relu activation functions.
MLP Layers: [49,490,980,1]
Activation functions: Relu
Initial learning rate: 1e-4
0.506M parameters

**Section 4.2**:

All NNs are MLPs with Relu activation functions. Number of nodes in each layer is written in the list.
MLP Layers: [49,490,700,490,1]
Activation function: Relu
Initial learning rate: 1e-4
0.712M parameters

Figure 5 : Timesreise Classification
MLP CrossEntropy (0.84M)
MLP Layers: [80, 720, 720, 360, 2]

Figure 6 : Timesreise Classification
NST CrossEntropy (0.69M)
Number of encoder layers: 2
Number of decoder layers: 1
Number of heads: 8
d_model: 128
Dropout: 0.1

MLP CrossEntropy (0.85M)
MLP Layers: [96, 720, 720, 360, 7]

**Table 2**:
MLPL1-0.5 (0.025M) & MLPL2-0.5 (0.025M):
Layers ob both models: [60,80,120,80,1]
Activation function: Relu
InitiaL learning rate: 1e-4

MLPL1(0.05M) & MLPL2(0.05M) :
Layers for both models: [120,80,180,1]

NSTL1 (0.05M) & NSTL2 (0.05M)
Number of encoder layers: 1
Number of decoder layers: 1
Number of heads: 2

d_model: 40
Dropout: 0.1

**Table 4**:
Here is the details of the pool of the used models -MLPs and Transformers- in one step ahead prediction experiment on weather dataset. For all non-stationary transformers (NSTs) dropout is set to 0.1.
The initial learning_rate for all models as the first stack is 1e-4 and for the second and the third stack is 1e-5.
The dropout for NSTs is 0.1 and for PatchTsT is 0.2, and there is no dropout for MLPs.
The size of subsequent hidden layer after the attention head (d_ff) in transformers are set to the provided default numbers, i.e. 4*d_model for NSTs and for 2*d_model for PatchTST.
Please note that PatchTST uses the vanilla Transformer encoder as its core architecture Nie et al. (2022) and therefore the number of decoder layer is zero.

PatchTST L2(0.41M)
PatchTST L2 architecture
Number of encoder layers: 3
Number of decoder layers: 0
Number of heads: 16
d_model: 128
Dropout: 0.2

MLP L1(0.59M)
MLP Layers: [96, 720, 720, 1]

NST L1(3.88M)
NST L1 architecture
Number of encoder layers: 2
Number of decoder layers: 2
Number of heads: 8
d_model: 256
Dropout: 0.1

NST L1(1.05M)
NST architecture
Number of encoder layers: 2
Number of decoder layers: 2
Number of heads: 8
d_model: 128
Dropout: 0.1

NST L1(0.86M) on PatchTST L2(0.41M)
NST architecture
Number of encoder layers: 2
Number of decoder layers: 1
Number of heads: 8
d_model: 128
Dropout: 0.1
PatchTST architecture
Number of encoder layers: 3
Number of decoder layers: 0
Number of heads: 16
d_model: 128

Dropout: 0.2

NST L1(1.13M) on PatchTST L2(0.41M)
NST architecture
Number of encoder layers: 2
Number of decoder layers: 2
Number of heads: 8
d_model: 128
Dropout: 0.1
PatchTST architecture
Number of encoder layers: 3
Number of decoder layers: 0
Number of heads: 16
d_model: 128
Dropout: 0.2

NST L2(0.53M) on MLP L1(0.59M)
MLP Layers: [96, 720, 720, 1]
NST architecture:
Number of encoder layers: 2
Number of decoder layers: 1
Number of heads: 6
d_model: 96
Dropout: 0.1

NST L1(0.86M) on MLP L1(0.59M)
MLP Layers: [96, 720, 720, 1]
NST architecture:
Number of encoder layers: 2
Number of decoder layers: 1
Number of heads: 8
d_model: 128
Dropout: 0.1

NST L1(2.05M) on MLP L1(0.59M)
MLP Layers: [96, 720, 720, 1]
NST architecture:
Number of encoder layers: 4
Number of decoder layers: 4
Number of heads: 8
d_model: 128
Dropout: 0.1

MLP L1(0.59M) on NST L1(1.13M) on PatchTST L2(0.41M)
MLP Layers: [96, 720, 720, 1]
NST architecture
Number of encoder layers: 2
Number of decoder layers: 2
Number of heads: 8
d_model: 128
Dropout: 0.1

PatchTST architecture
Number of encoder layers: 3
Number of decoder layers: 0
Number of heads: 16
d_model: 128
Dropout: 0.2

**Table 3**:
In this experiment, two simple SVM models are used:
SVM with rbf kernel
SVM with with sigmoid kernel

**Table 5**:

Here is the details of the pool of the used models -MLPs and Transformers- in multistep ahead prediction experiment on Nasdaq-DE1 dataset.

MLP L1 (2.09M):
MLP Layers: [60,360,3440,240,30]

MLP L2 (3.77M):
MLP Layers: [60,720,3440,360,30]

MLP L2 (4.61M):
MLP Layers: [60,900,3440,420,30]

NST L1 & NST L2 (2.67M):
Number of encoder layers: 2
Number of decoder layers: 1
Number of heads: 8
d_model: 256
Dropout: 0.1

MLP L2 (0.67M) on MLPL1 (2.09M)
MLP L2 Layers: [60,360,1080,240,30])
MLP L1 Layers: [60,360,3440,240,30]

MLP L2 (0.64M) on NSTL1 (2.67M):
MLP L2 layers: [60,360,1020,240,30]
NST L1 architecture:
Number of encoder layers: 2
Number of decoder layers: 1
Number of heads: 8
d_model: 256
Dropout: 0.1

MLP L2 (0.09M) on MLP L2 (0.64M) on NST L2 (2.67M)
NST L2 architecture:
Number of encoder layers: 2
Number of decoder layers: 1

Number of heads: 8
d_model: 256
Dropout: 0.1
MLP L2 (0.64M) layers: [60,360,1020,240,30]
MLP L2 (0.09M) layers: [60,720,60,30]

**Section 4.5**:

MLP L1 (0.6581M) on PatchTST
MLP L1 layers: [96 ,720 , 720, 96]
details of PatchTsT can be found in Nie et al. (2022).

## H    Further tables and figures

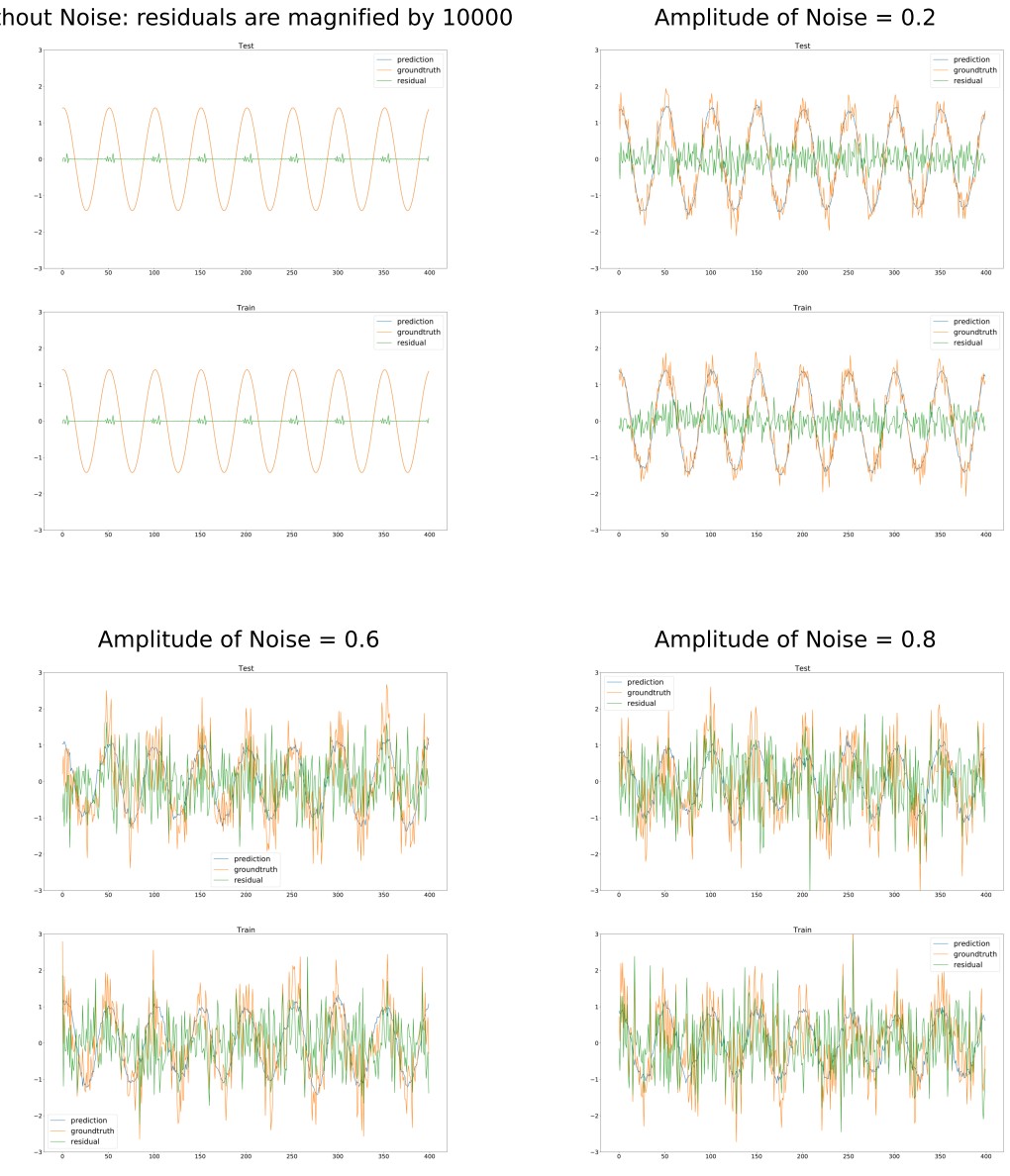

Figure 15: Plots of model outputs (prediction), residuals and the data (ground truth) of the experiments of Section 4.1.

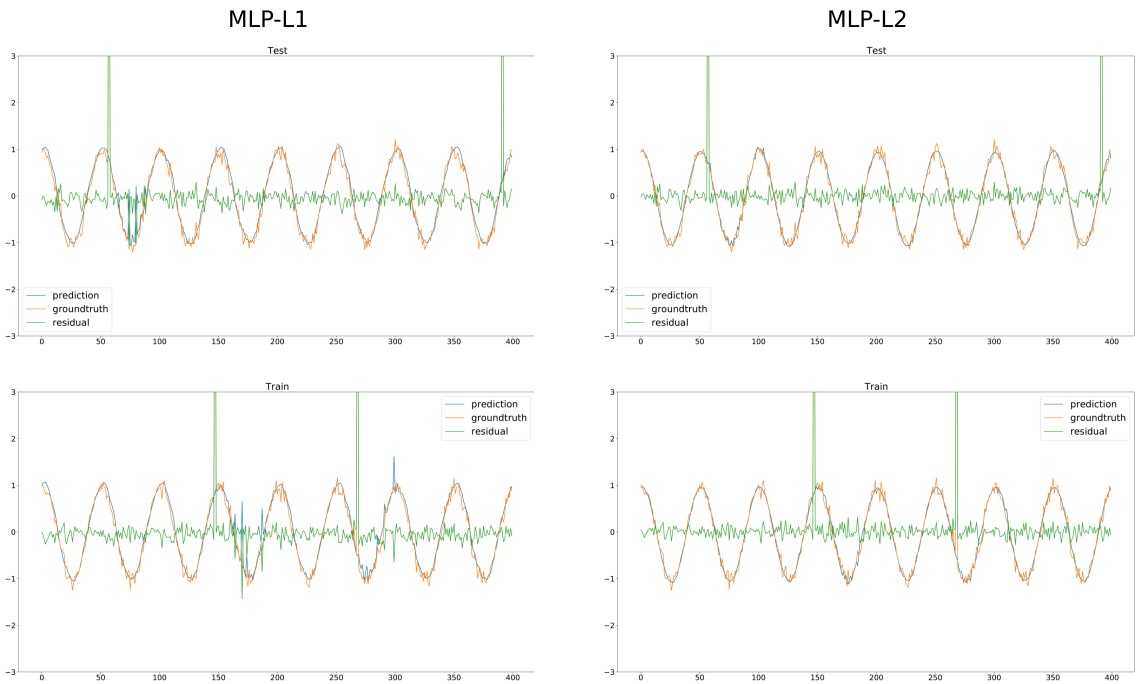

Figure 16: Plots of model outputs (prediction), residuals and the data (ground truth) of the experiments of Section 4.2.

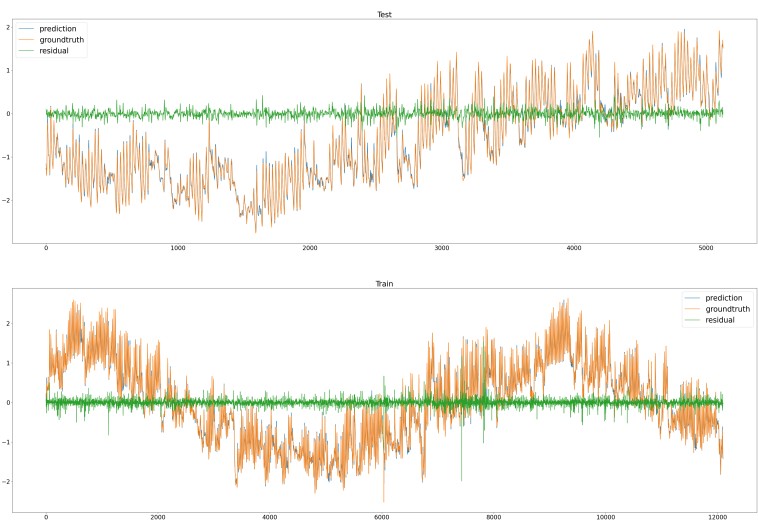

Figure 17: Plots of model outputs (prediction), residuals and the data (ground truth) of the ETTh2 dataset.

| Theoretical bound | Experimental Error | Pearsonr Analysis | |
|---|---|---|---|
| Relative noise std | Normalized Test RMSE | Initial R | Residual R |
| 0 | 0.000023 | 30.764 | 0 |
| 0.2715 | 0.2737 | 28.538 | 0 |
| 0.4919 | 0.5043 | 23.129 | 0 |
| 0.6465 | 0.6687 | 17.494 | 0.9446 |
| 0.7499 | 0.7683 | 13.197 | 0 |
| 0.8161 | 0.8287 | 10.455 | 0 |
| 0.8614 | 0.8718 | 7.868 | 0 |
| 0.8923 | 0.901 | 5.561 | 1 |
| 0.9149 | 0.9206 | 4.664 | 0 |
| 0.9308 | 0.9427 | 3.723 | 0 |
| 0.9429 | 0.9373 | 2.882 | 0 |

Table 6: Pearson correlation results for Section 4.1. The test consists of the sum of the absolute value of the Pearson correlation between each input and output features. However, the correlation measure is only considered if the p-value is smaller than 0.01. Otherwise, the corresponding input and output pair is considered as uncorrelated. In the first column, there is the relative noise, which is the root of the variance of the noise divided by the total signal. The second column provides the normalized RMSE as defined in Section 4. The third column is the sum of the absolute value of the Pearson correlation between input and the single step forecast as output and the forth column is analogous to the third column where the single step forecast is replaced by the corresponding residual (model prediction minus data). Since the residuals are less correlated or even decorrelated with the input, the model extracted the deterministic relations measured in the Pearson correlation test.

| - | Initial pearsonr | Residual Pearsonr + L2 | Residual Pearsonr + L1 |
|---|---|---|---|
| Trial 0 | 15.33 | 0 | 0 |
| Trial 1 | 16.925 | 0 | 0 |
| Trial 2 | 15.25 | 0 | 0 |
| Trial 3 | 17.104 | 0 | 0 |
| Trial 4 | 15.449 | 0 | 0 |

Table 7: The effect of the choice of the loss function in mitigating asymmetric noise effect in terms of distracting a model from extracting/learning the deterministic relations. The experiments are conducted in Section 4.2. The Pearson test is described in Table 6. While the chi-square test and the mutual information test depicted in Figure 2 reflect the better fitting of the model trained with $L^1$-loss function to the real data, see Figure 2, the Pearson test does not, which could be traced back to the limitation of testing only linear correlations where chi-square and mutual information are generalizations in terms of measuring stochastic independence.

# I Transformer architecture explanation

In this section, we explain the transformer architecture in more detail. Specifically, we explain the building blocks as depicted in (Vaswani et al., 2017, Figure 1).

## I.1 Token embedding

In the transformer model, each token of the input sequence is first represented as a dense vector called a token embedding. This embedding captures the semantic meaning of the token in the context of the task being performed.

Each token is encoded by a vector $z \in \mathbb{R}^{d_z}$ one-hot encoding the corresponding token where $d_z \in \mathbb{N}$ is the number of different tokens generated from the vocabulary. The token embedding is obtained by applying a linear transformation $A \in \mathbb{R}^{C \times d_z}$ to the one-hot encoded representation of tokens $z$ where $C \in \mathbb{N}$ is the number of dimensions of the transformer's internal representation of the embeddings ($C$ sometimes also denoted with *embedding_size*). The entries of $A$ are learnable weights and optimized during the training process. The mapping $A : \mathbb{R}^{d_z} \to \mathbb{R}^C$, $z \mapsto Az$ can be implemented as:

$$\text{token\_embedding} = \text{nn.Linear}(\text{config.vocab\_size}, \text{config.n\_embed})$$

where $\text{config.vocab\_size} = d_z$ is the size of the vocabulary (number of unique tokens), and $\text{config.n\_embed}$ is the desired embedding dimension $C$. Consequently, this mapping turns the input shape $(B, T, d_z)$ to the output shape of the token embedding $(B, T, C)$, where $B$ represents the batch size and $T$ represents the sequence length of the input (number of tokens). The transformation $A$ is applied to each token and element of a batch by $z(b, :, :) \mapsto z(b, :, :)A$ for each $b \in \{1, ..., B\}$ where $z \in \mathbb{R}^{B \times T \times d_z}$, $z(b, :, :) := (z_{bik})_{i \in \{1,...,T\}, k \in \{1,...,d_z\}}$ provides the one-hot encoded representation for each token and each element of the batch. By using batches, several inputs can be considered at once.

For the specific case of time series modeling, the token embedding is replaced by the following. A convolutional neural network (CNN) is used for generating the embedding. The input channels of the CNN equal 1 in an autoregressive scenario (only historic parts of the time series itself are use to predict future parts of the time series) but can be set to any feature number $F \in \mathbb{N}$, which is measured at each time point, in case, e.g., an output is predicted from several input time series. Each time window for each channel, which cuts out each part of the $F$ time series of length $T$ and which is used as the input for the prediction, is represented by a vector of size $T$. These vectors from the sliding window are transformed by a one-dimensional CNN into the space $\mathbb{R}^C$. The padding is set such that the input length $T$ equals the output length of the CNN. We remark that the shift one by one time points is not necessary and can be increased such that the token number in the embedding is smaller than the number of time points used as the input for the transformer. In our case, each filter (convolution) of the CNN is applied to each output, controlled by the parameter groups=1 and a common choice for the kernel size of 3. To achieve this transformation, we employ a one-dimensional convolutional layer in our implementation. Specifically, a 1D convolutional layer with an input channel size of $F$ and an output channel size of $C$ (the embedding dimension of the transformer) can be utilized:

$$\text{nn.Conv1d}(In\_channels = F, out\_channels = C)$$

This convolutional layer applies a set of learnable filters across the temporal dimension $T$ of the input data, extracting relevant patterns and features. It's important to note that the kernel size, padding, and stride parameters of the convolutional layer can be adjusted to ensure that the output length matches the input length $T$. For more details, see the PyTorch documentation, e.g., https://pytorch.org/docs/stable/generated/torch.nn.Conv1d.html.

In general, the transformation (token embedding by a CNN) is applied as follows $CNN : \mathbb{R}^{T \times F} \to \mathbb{R}^{T \times C}$, $z(b, :, :) \mapsto CNN(z(b, :, :))$ to each each element of a batch $z(b, :, :) \in \mathbb{R}^{B \times T \times F}$, $z(b, :, :) := (z_{bik})_{i \in \{1,...,T\}, k \in \{1,...,F\}}$, numerated by $b \in \{1, ..., B\}$ with the batch size $B \in \mathbb{N}$, (in parallel) generating a tensor of dimension $(B, T, C)$.

For more details about the general framework of token embedding, see, e.g., Zhou et al. (2021).

### I.2 Positional embedding

The purpose of a positional embedding is to include information about the position of a token relative to other tokens from the input into the total embedding of each token. The positional embedding is a vector of dimension $C$ and is added to the token embeddings to provide information about the relative positions of tokens.

There are several methods available to code for positional information, see, e.g., Vaswani et al. (2017).

### I.3 Attention head operation

The attention head is the essential building block of a transformer. Each attention head forms one layer where the output of one layer is processed by a subsequent layer until the output of the final layer is processed by a linear layer to obtain a corresponding output, see, e.g., (Vaswani et al., 2017, Figure 1).

Let's consider a single attention head operation in a transformer model. In this operation, we have the input tensor $x \in \mathbb{R}^{B \times T \times C}$. The tensor $x$ can be the one after the token embedding (inclusive positional encoding) or the output of a previous layer. We remark that each layer in this presentation preserves the format $(B, T, C)$.

The input tensor $x$ is next transformed into different representations via linear transformation matrices. These matrices contain the learnable weights and are given by $W^K \in \mathbb{R}^{C \times C}$, $W^Q \in \mathbb{R}^{C \times C}$ and $W^V \in \mathbb{R}^{C \times C}$. We define a tensor for key $(K)$, query $(Q)$, and value $(V)$ by the linear mappings

$$K(b, \cdot, \cdot) := x(b, \cdot, \cdot)W^K \in \mathbb{R}^{T \times C}, \quad Q(b, \cdot, \cdot) := x(b, \cdot, \cdot)W^Q \in \mathbb{R}^{T \times C}, \quad V(b, \cdot, \cdot) := x(b, \cdot, \cdot)W^V \in \mathbb{R}^{T \times C}$$

for each $b \in \{1, ..., B\}$ where $x(b, \cdot, \cdot) \in \mathbb{R}^{T \times C}$ such that $x(b, i, k) = x_{bik}$ for all $b \in \{1, ..., B\}$, $i \in \{1, ..., T\}$ and $k \in \{1, ..., C\}$. Each of the matrices $W^K$, $W^Q$ and $W^V$ is an instance of a linear layer and can be implemented with PyTorch as follows:

$$nn.Linear(C, C, bias = False).$$

We remark that $C$ may be called *embedding_size*.

In order to quantify the attention of token $i \in \{1, ..., T\}$ represented in its key representation $(K)$ with regard to token $j \in \{1, ..., T\}$ of the input represented in its query representation $(Q)$, the dot product is calculated for each $b \in \{1, ..., B\}$ between the query $(Q)$ and key $(K)$ tensors over the vector embedding for each token pair $i, j \in \{1, ..., T\}$. This operation can be represented as

$$\Theta : \{1, ..., B\} \times \{1, ..., T\} \times \{1, ..., T\} \to \mathbb{R}, \quad (b, i, j) \mapsto \Theta(b, i, j) := \frac{1}{\sqrt{C}} \sum_{k=1}^{C} Q_{b,i,k} \cdot K_{b,j,k} \tag{6}$$

where $b$ represents the batch index, $i$ and $j$ represent the positions in the sequence (input), and $k$ represents the embedding dimension. The sum is scaled by $C^{-0.5}$. One reason behind dividing the sum by the square root of the embedding dimension is given in the section about the Softmax function below, see Section I.3.1.

We implement the mapping $\Theta$ using the key $(K)$ and query $(Q)$ tensors and the Einstein summation convention as follows:

$$\Theta(b, i, j) = \frac{1}{\sqrt{C}} torch.einsum(bij, bkj-> bik, Q, K)$$

### I.3.1 Softmax

After calculating the similarities between keys and queries, the purpose of the Softmax function is to normalize the scores of similarity. A high similarity between the key of token $i$ and the query of token $j$ is a proxy for a high association or attention the token $i$ has to token $j$, meaning the connection is important for predicting the corresponding output. Due to the monotonicity of the Softmax function, a bigger similarity score between the corresponding key and query will result in a bigger value, called attention between the corresponding tokens, compared to smaller ones.

The Softmax function in our case is defined by

$$Softmax : \{1, ..., B\} \times \{1, ..., T\} \times \{1, ..., T\} \to \mathbb{R}, \quad (b, i, j) \mapsto Softmax(b, i, j) := \frac{e^{\Theta(b,i,j)}}{\sum_{l=1}^{T} e^{\Theta(b,i,l)}}.$$

Here, the Softmax function is applied along the last dimension, ensuring that the attention weights sum up to 1 along this dimension. This normalization means, fixing a batch number $b$ and a token number $i$ of the input provides us a normalized attention score about all the other token numbers $j \in \{1, ..., T\}$. The implementation is done by applying the corresponding Softmax function along dim$=-1$ to the tensor $\Theta_{b,i,j} := \Theta(b, i, j)$ for all $b \in \{1, ..., B\}$ and $i, j \in \{1, ..., T\}$.

Next, we explain the normalization by $C^{-0.5}$ of (6). For large numbers, the Softmax function is approximately a constant function and changes in the weights of the transformer model do not result in a significant change of the attention. Depending on the embedding dimension $C$, the corresponding sum scales. For an illustration, see, e.g., (Vaswani et al., 2017, footnote 4). The scaling of the sum by $C^{-0.5}$ ensures to stay in a range where the Softmax is in an area of larger steepness and changes in the weights of the transformer result in significant changes of the attention values. Similarly, see (Vaswani et al., 2017, Section 3.2.1).

### I.3.2 Weighted aggregation

We apply the attention scores to the value tensor $(V)$ to obtain a weighted sum. The attention tensor is defined by

$$\mathcal{A}_{b,i,j} := \text{Softmax}(b, i, j)$$

for all $b \in \{1, ..., B\}$ and $i, j \in \{1, ..., T\}$ such that $\mathcal{A} \in \mathbb{R}^{B \times T \times T}$. The weighted sum of attention scores is given by

$$\mathcal{A}(b, \cdot, \cdot)V(b, \cdot, \cdot) \in \mathbb{R}^{T \times C}$$

for each $b \in \{1, ..., B\}$ and turns the output of an attention head into the tensor format $(B, T, C)$. The output tensor of attention head $H \in \mathbb{R}^{B \times T \times C}$ is defined by

$$H_{b,i,k} := \sum_{l=1}^{T} \mathcal{A}_{b,i,l} V_{b,l,k}$$

for all $b \in \{1, ..., B\}$, $i \in \{1, ..., T\}$ and $k \in \{1, ..., C\}$.

### I.3.3 Multi-head attention

Optionally, in order to calculate attention on different subspaces of keys and queries for the same input in each layer, there is a multi-head attention taking only projected parts of keys and queries.

In the multi-head attention formalism, the output of each head is concatenated along the last dimension, which is the embedding dimension

$$Concat((B, T, C/n\_heads), ..., (B, T, C/n\_heads)) = (B, T, C)$$

where $n\_heads$ is the number of attention heads. Specifically, the output of each head is calculated with the following weight matrices $W_h^K \in \mathbb{R}^{C \times C/n\_heads}$, $W_h^Q \in \mathbb{R}^{C \times C/n\_heads}$, $W_h^V \in \mathbb{R}^{C \times C/n\_heads}$ and $W^O \in \mathbb{R}^{C \times C}$ where $h \in \{1, ..., n\_heads\}$. Furthermore, $n\_heads$ and $C$ are chosen such that the quotient $C/n\_heads$ is an integer. The matrices calculate the corresponding projections of keys, queries and values as follows $K^h(b, \cdot, \cdot) := K(b, \cdot, \cdot)W_h^K \in \mathbb{R}^{T \times C/n\_heads}$, $Q^h(b, \cdot, \cdot) := Q(b, \cdot, \cdot)W_h^Q \in \mathbb{R}^{T \times C/n\_heads}$, $V^h(b, \cdot, \cdot) := V(b, \cdot, \cdot)W_h^V \in \mathbb{R}^{T \times C/n\_heads}$ for all $b \in \{1, ..., B\}$.

The weight matrices $W_h^K$, $W_h^Q$ and $W_h^V$ are each implemented with a linear layer according to

$$nn.Linear(C, head\_size, bias = False)$$

and $head\_size$ is calculated as

$$head\_size = \left\lfloor \frac{embedding\_size}{n\_heads} \right\rfloor$$

where for $W^O$ the number $head\_size$ is replaced by $C$.

Applying the procedure for the single-head attention for each $h \in \{1, ..., n\_heads\}$ by replacing each $K$ by the corresponding $K^h$, each $Q$ by the corresponding $Q^h$ and each $V$ by the corresponding $V^h$ provides us the tensor of each attention head $H^h \in \mathbb{R}^{B,T,C/n\_heads}$ where in (6) the index in the sum is only over 1 to $C/n\_heads$ each.

After processing all attention heads, the outputs are concatenated along the last dimension, resulting in a tensor of shape $(B, T, C)$. The output of the multi-head attention is given by

$$H(b, \cdot, \cdot) \coloneqq \text{Concat}(H^1(b, \cdot, \cdot), ..., H^{n\_heads}(b, \cdot, \cdot))W^O \in \mathbb{R}^{T \times C}$$

for each $b \in \{1, ..., B\}$.

### I.4    Adding and layer normalization

After the attention head operation, residual connections and layer normalization are applied.

The adding of the input and the output of a layer (residual learning He et al. (2016)) is crucial for maintaining an information flow and is easing the training of deep networks. The residual connection involves adding the input tensor $x$ to the output of the multi-head attention operation $H$ according to $x + H$ where $+$ denotes an element-wise addition. The addition helps to loop through the original information from the input to all the layers in a sequential layer architecture while also incorporating the information learned by the attention mechanism. A prerequisite is that the attention head preserves the input format.

Following the residual learning operation, layer normalization is applied to stabilize the learning process Ba et al. (2016) according to

$$\text{LayerNorm} \coloneqq \frac{z - E(z)}{\sqrt{Var(z) + \epsilon}}$$

where $z$ is the output of the previous layer, $E(z)$ is the mean value of all the values of the output of the previous layer (mean over the elements of $z$) and $Var(z)$ is the corresponding variance. Since the square root is not differentiable at 0, a small constant $\epsilon > 0$ keeps the numerical implementation stable in case of a small variance of $z$. Moreover, the constant $\epsilon$ avoids division by zero errors. More details about the implementation can be found under `https://pytorch.org/docs/stable/generated/torch.nn.LayerNorm.html`. Due to its construction, the normalization is done for each element of the batch.

We remark that for our implementation, the normalization is applied over the embedding dimension ($C$) separately for each token of the input ($T$ dimension), meaning that $z \in \mathbb{R}^C$. This is reasonable since the feed forward networks (explained in Subsection I.5) are applied over the d_model ($C$) on each element of the batch ($B$) and input length ($T$).

### I.5    Feed forward network (FFN)

After the normalization step of the attention head's residual learning, each token's representation is passed through a feed-forward neural network (FFN). Such an FFN consists of two linear transformations separated by a non-linear activation function $g : \mathbb{R}^{m \times n} \to \mathbb{R}^{m \times n}$, $m, n \in \mathbb{N}$, $z \mapsto g(z)$, such as ReLU (Rectified Linear Unit; $g = \max$) according to

$$\text{FFN}(x(b, t, \cdot)) = g(x(b, t, \cdot)W_1 + b_1)W_2 + b_2 \in \mathbb{R}^C$$

for each $b \in \{1, ..., B\}$ and $t \in \{1, ..., T\}$ where $x \in \mathbb{R}^{B \times T \times C}$ is the output from the previous operation in the architecture, $W_1 \in \mathbb{R}^{C \times d_{ff}}$, $d_{ff} \in \mathbb{N}$, in our implementation $d_{ff} = 4C$, $W_2 \in \mathbb{R}^{d_{ff} \times C}$ and $b_1 \in \mathbb{R}^{d_{ff}}$. Furthermore, $x(b, t, \cdot)W_1 \in \mathbb{R}^{d_{ff}}$, which is the same bias for each $t$ in contrast to $b_1 \in \mathbb{R}^{T \times d_{ff}}$ where

for each $t$ there is another bias, and $b_2 \in \mathbb{R}^C$ are learnable parameters. These operations are applied by Pytorch's linear layer `https://pytorch.org/docs/stable/generated/torch.nn.Linear.html`. The activation function $g$ introduces non-linearity to the model, enabling it to learn non-linear patterns from the data. In the transformer architecture, an FFN is also followed by residual learning and layer normalization as described above in Subsection I.4.

### I.6   Masking

Looking at (Vaswani et al., 2017, Figure 1), the masking is one of the essential building blocks located within the decoder (explained in Subsection I.7).

The masking of values of the attention weights $\Theta$ has the purposes of forcing attention between tokens to zero, meaning not allowing them to interact or to extract information from the interaction. Due to the iterative application of the transformer for text generation, we would like to force the attention mechanism that a token only considers tokens backwards in time (that come earlier in a sentence). This backward orientation helps to generate representations of tokens that collect information from tokens that are already there and prevents generating representations that make use of tokens that come after that token in a sentence. By the procedure of masking, the representations of tokens are more unified independent of the input length. As an example: "I am hungry and thus I go to a restaurant." Although probably "hungry" should get a lot of attention with "restaurant" without masking, in an iterative application of the transformer, the word "hungry" in "I am hungry and thus I go" would not be useful since its most attention was on restaurant that is not there yet. However, with masking we force the transformer to find embeddings and representations such that the word "hungry" gets a useful representation to predict the next token during learning, independent if the input is "I am hungry and thus I go to a restaurant." or "I am hungry and thus I go".

A different interpretation of the masking can be causality in a use case where the sequence of events is of importance forcing attention only to historic events.

We implement the masking effecting only backwards interaction by a lower triangular mask $M \in \mathbb{R}^{T \times T}$. This matrix is applied to the attention weights $\Theta$ generating masked attention weights. The tokens of the input are counted in the second dimension of the tensor $\Theta$ where the third dimension accounts for the dimension of the current vector representation ($T$ or $C$ depending on the current representation). Consequently, the lower triangular matrix (1 on the diagonal and below and 0 above), allows token 1 to have a non-zero attention only with token 1, token 2 can interact with token 2 and token 1 and so on until token $T$.

Subsequently, for each $b \in \{1, ..., B\}$, the entries of the attention weight tensor $\Theta(b, :, :)$ are replaced by $-\infty$ where $M$ equals zero. The attention tensor $\Theta$ is thus transformed into the masked attention weight tensor $\Theta^M$. The $-\infty$ forces the corresponding Softmax calculation to zero in the corresponding positions, meaning that the corresponding token does not pay attention to the corresponding other tokens.

### I.7   Encoder and decoder

In a transformer architecture, there are typically two main components. This is the encoder and the decoder. Next, we explain both components according to (Vaswani et al., 2017, Figure 1).

Usually each layer in the encoder consists of an attention head followed by a residual learning and normalization, which is the input into a two linear transformation separated by a non-linear activation function, also followed by a residual learning and normalization. For the decoder, a layer consists of a masked attention unit (self-attention), followed by residual learning and layer normalization, followed by an attention unit where key and values are calculated from the corresponding encoder layer (cross-attention). The rest of the layer is according to an encoder layer. The repetition of layers of encoder and decoder is the main building block for the transformer.

The encoder processes the input sequence. These are the text input or for time series prediction the historic time series (the time series itself or other time series of features) generating a vector representation that

captures the contextual information of each token, which is meant also for the time series case as discussed in Subsection I.1.

The decoder, on the other hand, takes the encoded representations and generates an output sequence. It also consists of multiple layers, each containing self-attention mechanisms and cross-attention mechanisms. The self-attention mechanisms help the decoder focus on different parts of its input sequence, while the cross-attention mechanisms allow it to incorporate information from the encoder's output.

For the case of generating iteratively the next token for text generation, the prediction target is typically the next token in a sequence. Since for the text generation, several predicted tokens are required, the input of the decoder grows by the predicted token after each iteration. For inference the next token is predicted. Also during the training, the model is trained to predict the next token given the previous tokens in the input sequence. The number of input tokens for the decoder is given by $L \in \mathbb{N}$. The input to the encoder can be passed to the input of the decoder. If tokens are iteratively generated, the number $L$ is supposed to be bigger than $T$ in such cases, where $T$ is the input length of the encoder. If the iterative output of the decoder becomes longer than a maximum size $\tilde{L} \in \mathbb{N}$ for the decoder's input, which can exist due to limitation on the hardware to calculate attention for such an input length between each token, then only the latest $\tilde{L}$ tokens are used as an input for the decoder. If the sequence is shorter, corresponding positions are masked out as explained in Subsection I.6. For translating, input language of the encoder's input can be different to the language of the decoder's output/input. In such a case, the encoder's input may not be passed to the decoder's input and the input of the decoder is iteratively generated by several applications of the transformer.

In any case, the output of the decoder $x \in \mathbb{R}^{B \times L \times C}$ is transformed by a linear map $\bar{W} : C \to \mathbb{R}^{d_z}$ with bias $\bar{b} \in \mathbb{R}^{d_z}$ such that the last dimension fits the number of available tokens from a dictionary according to

$$\bar{x}(b,i,:) := x(b,i,:)\bar{W} + \bar{b} \in \mathbb{R}^{d_z}$$

for each $b \in \{1, ..., B\}$ and $i \in \{1, ..., L\}$. Then, the Softmax function is applied to the last slice $L$ of the tensor $\bar{x}$ according to

$$\text{Softmax} : \{1, ..., B\} \times \{1, ..., d_z\} \to \mathbb{R}, \quad (b,s) \mapsto \text{Softmax}(b,s) := \frac{e^{\bar{x}_{b,L,s}}}{\sum_{l=1}^{d_z} e^{\bar{x}_{b,L,l}}}$$

to obtain a probability over all possible tokens to choose the most likely token as the following token for each $b \in \{1, ..., B\}$. To include some variety on choosing the next token, we can disturb this distribution for the next token (e.g., introducing a temperature parameter) a bit such that also tokens become the most likely one that are close to the most likely token according to the undisturbed distribution over the tokens.

For time series forecasting tasks, the prediction target may vary depending on the application. It could be the next value in the time series sequence, multiple future values, or even a binary classification indicating whether certain conditions will be met in the future. The basic concept is that a linear transformation $\tilde{W} : C \to E, E \in \mathbb{N}$, with a bias $\tilde{b} \in \mathbb{R}^E$ transforms the output of the decoder to the output format that corresponds to what is to predict, like the number of features or the numbers of classes that is then turned into a probability over classes by a corresponding Softmax function.

In this work, we focus on time series prediction. As discussed in Zhou et al. (2021), it is advantageous to generate a multistep prediction (which includes a singlestep prediction) not by an iterative application of the transformer, like explained above, but provide the prediction at once, meaning to provide the prediction of length $L \in \mathbb{N}$ with a single application of the transformer. As a consequence, the training is done with a direct multistep loss. The rationale behind generating the prediction at once is to avoid error accumulation within the multistep ahead time series prediction task. Considering (Liu et al., 2022, Algorithm 4), we define the input for the decoder by $x \in \mathbb{R}^{B \times \frac{T}{2} + L \times F}$ where $L \in \mathbb{N}$ is the number of steps within the multistep prediction or prediction length, respectively, and $F \in \mathbb{N}$ the number of features, analogously to the token embedding for time series prediction tasks described in Subsection I.1. While for the encoder the initialization is the input sequence, the initialization values for the decoder are as follows. The first $\frac{T}{2}$ slices of the decoder input $x$ are filled with the last $\frac{T}{2}$ slices of the input of the encoder $\tilde{x} \in \mathbb{R}^{B \times T \times F}$ according to $x_{b,i,f} = \tilde{x}_{b,\frac{T}{2}+i,f}$ for all

$b \in \{1, ..., B\}$, $i \in \{1, ..., \frac{T}{2}\}$, $f \in \{1, ..., F\}$. The last slices of the decoder are initialized with zeros according to $x_{b,i,f} = 0$ for all $b \in \{1, ..., B\}$, $i \in \{\frac{T}{2} + 1, ..., \frac{T}{2} + L\}$, $f \in \{1, ..., F\}$. This representation is embedded, see Subsection I.1, and processed as shown in (Vaswani et al., 2017, Figure 1) by a number of layers within the transformer. The output of the decoder, again denoted with $x \in \mathbb{R}^{B \times \frac{T}{2} + L \times C}$, is transformed by a linear mapping according to

$$P(b, i, :) := x(b, i, :)\tilde{W} + \tilde{b} \in \mathbb{R}^E$$

for each $b \in \{1, ..., B\}$ and $i \in \{1, ..., \frac{T}{2} + L\}$ where $\tilde{W} \in \mathbb{R}^{C \times E}$, $\tilde{b} \in \mathbb{R}^E$ and $P \in \mathbb{R}^{B \times \frac{T}{2} + L \times E}$. In our application, where we predict the time series from its history, $F = E = 1$. The output after the linear transformation represents the $L$-step prediction and is given by

$$P(b, i, e) \text{ for all } i \in \{\frac{T}{2} + 1, ..., \frac{T}{2} + L\}$$

for each element of the batch $b \in \{1, ..., B\}$ and dimension $e \in \{1, ..., E\}$. Based on the output, loss functions are calculated with respect to the corresponding ground truth.

## J  Architecture for multilayer perceptrons for time series prediction

In this section, we describe the multilayer perceptron (MLP) architecture that we use for the time series prediction in this work. There is evidence that also MLPs are a very powerful model to predict time series Zeng et al. (2023).

Iteratively, an input tensor $x \in \mathbb{R}^{B \times F \times T}$ with $B \in \mathbb{N}$ as the batch size, $T \in \mathbb{N}$ as the length of the historic input of the time series for the prediction and $F \in \mathbb{N}$ as the number of features is transformed to the output tensor $y \in \mathbb{R}^{B \times F \times L}$ where $L \in \mathbb{N}$ is the length of the multistep prediction, which includes singlestep prediction where $L = 1$. In between there can be several hidden layers. All layers have the following structure taking an input tensor $z_{d-1} \in \mathbb{R}^{B \times F \times N_d}$ with a certain number of nodes ("neurons") $N_d \in \mathbb{N}$ where $d \in \{1, ..., n\}$, $n \in \mathbb{N}$ is the number of layers, $N_1 = T$, $N_{n+1} = L$ and $z_0 := x$. The layer $d$ is defined by the function given as follows

$$M_d : \mathbb{R}^{N_d} \to \mathbb{R}^{N_{d+1}}, \quad z_{d-1}(b, f, :) \mapsto M_d(z_{d-1}(b, f, :)) := g_d(z_{d-1}(b, f, :)W_d + b_d)$$

for each $b \in \{1, ..., B\}$ and $f \in \{1, ..., F\}$ where $g_d : \mathbb{R}^{F \times N_{d+1}}$ is a pointwise applied non-linear activation function for each $d \in \{1, ..., n\}$, like the ReLu function where $g_d = \max$, $z_{d-1} \in \mathbb{R}^{B \times F \times N_d}$ is the output from layer $d - 1$ and the input for layer $d$, $W_d \in \mathbb{R}^{N_d \times N_{d+1}}$ and $b_d \in \mathbb{R}^{N_{d+1}}$. The operation $z_{d-1}(b, f, :)W_d$ is the common matrix-vector multiplication for any $d \in \{1, ..., n\}$, $b \in \{1, ..., B\}$, $f \in \{1, ..., F\}$. We remark that $g_d$ can be but does not have to be a different function for each layer. For each $b \in \{1, ..., B\}$ and $f \in \{1, ..., F\}$, we have that $y(b, f, :) := M_n(z_{n-1}(b, f, :)$. In this formulation, all weights in each layer $d$ are the same for all features. This is the implementation we provide and is used in Zeng et al. (2023). However, in the examples within the present work, we have $F = 1$.

To implement a version that has different weights for each feature in each layer $d$, we just need to reformulate the input of the layers by $z_{d-1,f}(b, :) = z_{d-1}(b, f, :) \in \mathbb{R}^{N_d}$ for all $f \in \{1, ..., F\}$. Accordingly, the definition of the layers looks like

$$M_{d,f} : \mathbb{R}^{N_d} \to \mathbb{R}^{N_{d+1}}, \quad z_{d-1,f}(b, :) \mapsto M_{d,f}(z_{d-1,f}(b, :)) := g(z_{d-1,f}(b, :)W_{d,f} + b_{d,f})$$

where applying the definitions separately to each feature leads to $F$ different MLPs where the weight matrices and bias can differ per feature.

In order to introduce cross learning where information from one feature can influence the prediction of other features, we need to reshape the three dimensional tensor $(B, F, N_d)$ to $(B, FN_d)$ for some layers where a corresponding weight matrix $W_d^* : FN_d \to FN_{d+1}$ can mix information from different features.

In a multi layer architecture, we can combine cross learning and learning per feature in different layers assembling them in one model by reshaping outputs in the corresponding formats from $(B, F, N_d)$ to $(B, FN_d)$ or $(B, FN_d)$ to $(B, F, N_d)$ after a layer before the next one depending on the learning type to change.

With Pytorch such layers are implemented with

$$nn.Linear(n, m, bias = True)$$

where $n \in \mathbb{N}$ is the dimension of the input and $m \in \mathbb{N}$ is the dimension of the output. The *bias* parameter *True* adds a bias with non-zero values and the parameter *False* fixes the values of the bias to zero.

