# OpenReview forum: "Towards Measuring Predictability: To which extent data-driven approaches can extract deterministic relations from data exemplified with time series prediction and classification"
_TMLR — Accepted by TMLR_

### Review · Reviewer_Zume · 2024-06-18

**Summary Of Contributions:**

The work presents a statistical framework to measure how well a machine learning model captures deterministic relationships in data, distinguishing these from unpredictable components. It also proposes a stacking architecture to systematically extract information where individual models fall short and demonstrates its effectiveness on application to time series forecasting. This approach paves the way for developing new loss functions that focus on fitting models to deterministic relations, potentially improving model accuracy by accounting for noise.

**Audience:**

Yes

**Broader Impact Concerns:**

No Broader Impact Statement provided, and I do not think one is required.

**Claims And Evidence:**

Yes

**Requested Changes:**

The requested changes are detailed in the Weaknesses section above.

**Strengths And Weaknesses:**

Strengths:
- The list of related works seems to be quite comprehensive, and the paper is generally well written, however, there are some points that remain unclear (see the Weaknesses section).
- The work presents the general statistical framework for improving ML training with an application to time series forecasting, demonstrating its practical relevance and real-world applicability. This general framework is not reliant on a specific implementation of mutual information, allowing for the use of any method to calculate mutual information.

Weaknesses:
- The parts concerning the calculation of margin probabilities and joint distributions for estimating mutual information lack detail. For example, the section "Discretization scheme for co-domains or random variables" can be revised to include exact formulas for implementing the chi-square test and calculating mutual information.

---

> ### Author Response · Authors · 2024-12-02
> **Answer to Zume's review report**
>
> We thank the reviewer for the dedicated time to provide us feedback and highlighting the value of our research. We have added our extensions of the manuscript in blue. Below we explain in detail how we approached the requested changes.
>
> ### The parts concerning the calculation of margin probabilities and joint distributions for estimating mutual information lack detail. For example, the section "Discretization scheme for co-domains or random variables" can be revised to include exact formulas for implementing the chi-square test and calculating mutual information.
>
> Thanks for your suggestion, to address your request we have added the followings to the paper:
>
> 1. We have added Appendix A entitled "Further explanation on discretization and empirical probability estimation" which includes further explanation and visualization of the discretization process.
> 2. We have added the following to Section 2.2:
>     - A paragraph starting with "In our case, such an event is defined through..." and "These upper and lower bounds of each bin define..." to provide more details how our distributions are calculated from random variables that model input and output.
>     - The output of Algorithm 1 is now clarified inside the pseudo code.
>     - In particular, we added "The marginal and joint probabilities of (3) and (4) are calculated as follows. ..." to explain how the distributions are calculated after discretization.
>
> We hope these new explanations clarify our method. We remain at your disposal for further questions.

---

### Review · Reviewer_oj2b · 2024-07-11

**Summary Of Contributions:**

This paper presents a novel framework that aims to measure predictability in machine learning models by using mutual information. The proposed method assesses the predictability of a target given the available input data and utilizes this information to refine the training process. By differentiating model errors into unpredictable components and systematic misses of deterministic relations, the framework provides a new perspective on model success and convergence. The primary contribution lies in the stacking architecture, which enhances model training by combining different models to extract remaining information. The framework's application is demonstrated on both synthetic and real-world time series datasets. Overall, I find the ideas presented in this paper very promising. With some additional theoretical analysis, comparative studies, and a bit of polishing in writing, this work could make a significant impact in the field.

**Audience:**

Yes

**Broader Impact Concerns:**

The paper currently doesn’t address the broader ethical implications of the proposed framework. Given the increasing focus on transparency and fairness in machine learning, it would be beneficial to include a Broader Impact Statement. Specifically,
1. How does the framework handle potential biases in the input data? Ensuring fair model training and predictions is crucial.
2. How transparent and interpretable are the predictability measures for stakeholders who may not be machine learning experts?
3. Are there considerations for data privacy, especially when dealing with sensitive datasets?

**Claims And Evidence:**

Yes

**Requested Changes:**

1. Provide a more detailed theoretical analysis of the mutual information measure and its approximations. Discuss the potential impact of the measure's discrepancy on the framework's effectiveness.

2. Elaborate on why mutual information was selected and compare it with other potential measures, such as those used in signal processing for time series modeling. Discuss the advantages and limitations of each.

3. Include additional experimental details and results that explore how the discrepancy in the measure affects the training paradigm. This could involve sensitivity analysis or robustness checks.

4. Writing and presentation improvements:
   - Reduce excessive cross-referencing and repetitive statements to improve readability.
   - Correct typos and improve the overall organization and formatting of the manuscript.
   - Provide clearer, more concise explanations of key concepts and findings.

**Strengths And Weaknesses:**

**Strengths:**
1. The framework introduces a unique approach to measuring predictability and using this measure to guide the training process of machine learning models.
2. The paper provides concrete examples and applications, particularly in time series prediction and classification, showcasing the framework's effectiveness.
3. The framework is designed to be independent of specific model architectures, making it versatile and broadly applicable.
4. The differentiation of model errors into predictable and unpredictable parts offers valuable insights into model performance and potential for improvement.

**Weaknesses:**
1. The approximation used for measuring mutual information, due to computational complexity considerations, may affect the framework's effectiveness. A thorough theoretical analysis is missing, raising concerns about the reliability of the measures.
2. There is a need for more in-depth theoretical analysis to support the framework. Experimental details are provided, but a rigorous theoretical foundation is lacking.
3. The paper could benefit from exploring and comparing other measures of predictability, such as innovations in signal processing, to strengthen its claims and provide a more comprehensive evaluation.
4. The manuscript suffers from organizational issues, including excessive cross-referencing of sections, repetitive statements, and numerous typos, which can distract readers from the main contributions.

---

> ### Author Response · Authors · 2024-12-02
> **Answer to reviewer oj2b**
>
> We thank the reviewer for the dedicated time to provide us feedback with which we can further improve our work. We would like to mention that the main focus of our work is not how to estimate information measures. There is already plenty of work such as **Belghazi et al. (2018)**, **Franzese et al. (2023)**, **Kraskov et al. (2004)**, **Moon et al. (1995)**, **Pizer et al. (1987)**, **Darbellay & Vajda (1999)**, **Song & Ermon (2020)**, **van den Oord et al. (2018)**, and **Gao et al. (2017)** on measuring mutual information. Additionally, **Czyż et al. (2023)** provides a thorough benchmarking of various methods across a wide range of scenarios and discusses the advantages and disadvantages of different mutual information estimators.
>
> Our framework is intended to apply such methods for introducing a measure for the predictability of an output given the input data, as well as to measure how much of this information is extracted by a model. Therefore, we believe designing and benchmarking a new estimator is another line of research, as we have written in the Introduction:
>
> > *“The choice of the measures and their implementations themselves are not the focus of this work.”*
>
> Furthermore, the bullet points in the Introduction clearly explain this focus. Our framework and Git repository are modularized and thus do not depend on a specific implementation of such measures. They can be interchanged or replaced with whatever the user thinks is suitable. Please see the related work section **"Mutual information estimation and applications"**, which we have extended with further literature.
>
>
> Furthermore, we would like to mention that we do not rely solely on mutual information as a measure for predictability, as we show by utilizing the chi-square test of independence. The intention is to test a statistical hypothesis about the independence of input and output, or model deviations. Due to the limitations mentioned in the Discussion, we think it is advantageous to have several statistical tests. For instance, we write:
>
> > *"... we used a framework in which we could directly implement further stochastic measures like the chi-square test, not being just specific for mutual information estimation.”*
>
> and
>
> > *"Moreover, by using different statistical tests, we can demonstrate ..."*
>
> in the Related Work section. We have chosen a discretization framework (Algorithm 1) that can be applied for both MI estimation and the chi-square test of independence to keep the focus on our above-defined scope of the work.
>
> In addition, to provide even a third option for creating a statistical test for the independence of input and model deviations, we added:
>
> > *"This likelihood can be calculated for each pair of random variables separately...”*
>
> Below, we answer the requested changes in detail. In the manuscript, the changes are highlighted in blue.
>
> ---
>
> ### Continued on the next page

---

> ### Author Response · Authors · 2024-12-02
> **Answer to reviewer oj2b (2nd part)**
>
> ## Requested Changes
>
> ### Provide a more detailed theoretical analysis of the mutual information (MI) measure and its approximations. Discuss the potential impact of the measure's discrepancy on the framework's effectiveness.
>
> We follow the related work **Darbellay et al. (1999)**, which utilizes an adaptive discretization scheme to compute mutual information. Therefore, based on that, we have added appendix B, containing more explanation regarding the convergence of our estimation to the ground truth MI in case of having sufficiently large amounts of data. We showed that the same assumptions hold in our case, and therefore we can have the same theoretical results.
>
> Besides the theoretical results, **Czyz et al. (2023)** showed for low-dimensional settings that histogram-based methods provide a good estimate of MI.
>
> Additionally, we have pointed out in the Discussion that our current implementation of pairwise independence is a necessary condition but computationally cheaper than the full consideration. We wrote:
> > *"Consequently, we are aware that the current pairwise approach, which is computationally cheap compared to the full approach ..."* and *"Under model convergence with independent input and model deviations, it is required that the pairwise test confirms independent input and model deviations as well, which represents a necessary condition for convergence,"* analogous to the utilization of the gradient to define global optima.
>
> It is important to note that pairwise independence is a necessary condition of joint independence. This means that if the pairwise dependency remains, there may still be underlying relationships that the machine learning model needs to learn. And this is exactly what we check with our statistical tests. We wrote:
> > *"Consequently, the opposite direction is true that a pairwise test indicating deterministic relations between input and residuals/model deviations implies that there is information left a model can learn"*
>
> We began with this simpler/relaxed goal since, as noted in **Czyz et al. (2023)**, histogram-based methods can provide a good approximation in low-dimensional settings. However, as said earlier, neither the focus of this work is on estimating MI nor our framework relies on a specific way of computing MI, allowing the most recent and powerful estimation methods such as diffusion-based ones **Franzese et al. (2023)** to replace our estimator to handle the estimation of high-dimensional densities, especially in low data regime areas, thanks to the recent advances in diffusion literature.
>
> The main downside with the current estimator is that it can't accurately estimate the joint MI between all inputs and outputs in high-dimensional settings because the number of bins would increase exponentially, as would the number of necessary data points. Especially in the low data regime (low-density areas), the situation is worse. However, this problem is already addressed in the diffusion literature. The work **Franzese et al. (2023)** is an existing example of such methods, which can be plugged into our framework later.
>
> As we write in the Conclusion:
> >*"Such an inaccuracy in the estimation of the mutual information for a loss function could cause divergence of the optimization procedure and thus not improve the model's capability to extract more deterministic relations differentiating them from unpredictable parts like noise given the input data,"* we are aware that a well-working estimation of stochastic measures is central. However, there is literature about it, and our main focus is to show another use case/application for all this research: *"Our work may further motivate the development of such methods as we give another application of such measures and estimators for model evaluation."*
>
> -----
>
> ### Continued on the next page

---

> ### Author Response · Authors · 2024-12-02
> **Answer to reviewer oj2b (3rd part)**
>
> ### Elaborate on why mutual information was selected and compare it with other potential measures, such as those used in signal processing for time series modeling. Discuss the advantages and limitations of each.
>
> We thank the reviewer for the suggestion and believe the reviewer is referring to the Innovation ([wikipedia](https://en.wikipedia.org/wiki/Innovation_(signal_processing)), **reid2001**, **houts2013**,**bode1950**, **mitter2005**) metric, that is popular in signal processing community. We start by briefly introducing the innovation metric to make sure we mean the same concept.
>
> **Innovation metric** is extensively used in applications that use Kalman filters (or variants) and acts as an indirect measure of the consistency of the filter parameters when the true (ground truth) latent states of a dynamical system are typically inaccessible. It compares the noisy "true sensory observations” with the predicted observations, computed using the observation model of the filter, i.e.,
> $$ \nu_t = o_t - \hat{o}_t = o_t - H_t \hat{z}_t $$
>
> Here:
>
> - $ t $ is the time index.
> - $ o_t $ and $ \hat{o}_t $ are the actual and predicted observations respectively.
> - $ \hat{z_t} $ is the predicted latent state, computed using the prediction model (transition matrix) as $ \hat{z_t} = F_{t-1} \hat{z_{t-1}} $.
> - $ H_t $ and $ F_{t} $ are the observation and the prediction/transition model of the filter respectively. These models can be either **handcrafted** based on expert knowledge or **learned** using data-driven methods.
> - $ \hat{z}_{t-1} $ is the estimated state from the previous time step.
>
> **Innovation sequence and model consistency** Consider an innovation sequence over a prediction window  $t:t+H$, is defined as $\{\nu_t, \nu_{t+1}, \nu_{t+2}, \dots, \nu_{t+H}\}$. A filter is consistent if $\boldsymbol{\nu}_t$ is standard normally distributed with zero mean and a covariance $\mathbf{S}_t$ indicating successive innovations are uncorrelated with each other, i.e., constitute a white noise time series. Thus, for filter consistency, the residuals satisfy the following conditions,
>
> $$
> \mathbf{v}_t \sim \mathcal{N}\left(\mathbf{0}, \mathbf{S}_t\right),
> $$
>
> $$
> \mathrm{Cov} \left( \mathbf{v_t}, \mathbf{v_{t-k}} \right) =
> \begin{cases}
> \mathrm{Cov}\left(\mathbf{v}_t\right) & \text{if } k = 0, \\
> 0 & \text{otherwise}.
> \end{cases}
> $$
>
> Here, the innovation covariance $ \mathbf{S_t} $ is given by $ \mathbf{S_t} = H_t \mathbf{P_{t|t-1}} H_t^T + R_t $, where $ \mathbf{P_{t|t-1}} $ is the predicted state covariance and $ R_t $ is the observation noise covariance.
>
> If the above conditions for residuals are not satisfied, it indicates a model mismatch or suboptimal Kalman filter. A variety of tests including visual inspection, innovation magnitude bound test, Normalized innovations $\chi^2$ test, autocorrelation test etc. can be performed to check these conditions. Please see page 19, Section 2.2.3 from *reid2001estimation* for a detailed description of these.
>
> **Differences to our proposed predictability metrics**
>
> 1. **Model Agnostic**
>    Our main aim is to calculate the “predictability” directly by investigating data (see the first step in Figure 1 of the manuscript) even before a model is given/trained. In this regard, the innovation metric and related tests are not appropriate since they are always calculated in reference to a given model, i.e., these methods depend on the transition model $F_t$, observation model $H_t$, or the noise assumptions in the filter. However, our metrics (MI and Chi-Square tests) are agnostic to any specific models and allow for the measurement of the predictability even before the model training/designing phase to save time and costs.
>
> 2. **Generality**
>    Our metrics, based on stochastic independence and MI, are more general and do not rely on modeling assumptions made for the filter parameters like the transition/observation matrix and corresponding distributional assumptions on noises. Our metrics are suited both as a test of predictability and as a convergence criterion, irrespective of any assumptions about the models or the residuals themselves.
>
> 2. **Input-Output Relation**
> We do not only consider the investigation of residuals themselves but also whether, given the input data, the residuals could be optimized in the sense of extracting more deterministic relations between input and output (target) that a model could learn. We wrote,
> >*"All these measures are supposed to consider the triple of input, model output, and ground truth to calculate ..."*
>
>       and please see the corresponding paragraph in the Conclusion.
>
> **We have added a description of the Innovation method to the Related Work section**.
>
> Further, we remark our work is not only applicable to residuals (where we can subtract data and model output) but also to nominal data.
>
> -----
>
> ### Continued on the next page

---

> ### Author Response · Authors · 2024-12-02
> **Answer to reviewer oj2b (4th part)**
>
> ### Include additional experimental details and results that explore how the discrepancy in the measure affects the training paradigm. This could involve sensitivity analysis or robustness checks
>
> In this work, we have  used MI as a necessary condition and additional criteria.
> Please note that the current version of our work does not affect the training process directly, as training process is conducted using gradient descent with traditional losses such as L2, L1 or Cross Entropy in classification settings.
> We extend the existing training schemes with additional criteria that defines a good convergence of a model. Particularly with the bullet points in the introduction, we highlight where the consideration of stochastic measures brings further insights into the current procedure.
>
> In summary, our framework checks to what extent the reliable information is learned by the prediction model, where solely tracking of the traditional loss functions (learning curves) might not reveal those optimal points. Please see how in the experiment 4.2 where rare positive peaks, modeled by random variable $\theta_2$, arriving at random time (and thus unpredictable), affects the traditional training scheme. L2 optimization simply results in predicting higher values at all points (to compensate for the unpredictable positive peaks) and therefore would lead to a wrong prediction for all values of the target, although only very rare amount of points (1/200) are affected by that noise in reality. Training with other loss functions although is a valid approach. Comparing the models trained with different loss functions usually is not straightforward since it is often the case that the model trained with loss function A is better in terms of metric A, and model trained with Loss function B is better in terms of metric B, however our loss function agnostic framework enables us to make such a comparison by determining to what extent the **reliable information** is learned by the model. Please see Section 4.2, e.g., in particular "Another conclusion that we draw from this experiment ...".
> With our framework, we detect if the trained models are distracted by unreliable patterns such as the noise in the data (robust to any complex noise/unpredictable pattern)
>
> To emphasize more this aspect of limitations that any mutual information estimator might have, we extended the Limitations and Assumptions section in the Discussion, saying that we should not rely on a single estimator “Due to the fact that any approximation or estimation of a measure, like mutual information, may not work sufficiently accurate ...”.
>
> Our method, however, suggests considering some other optimal parameters of neural networks, which could lead to a better performance according to our metrics.
> In this sense, our framework doesn't exclude the traditional optimal points but identifies some other optimum points too, which can be a better solution where traditional loss functions may not capture them. Even if there is a significant discrepancy or high error in estimating MI, then traditional optimal points remain observable in the training curves.
> Moreover, our method does not rely on a specific method of estimating MI, leaving doors open for recently developed diffusion-based MI estimation methods **franzese2023** to play a role in our problem.
> In summary our approach does not preclude traditional methods but may uncover additional optimal points, potentially offering improved solutions.
> As shown also in MI graphs, learning information, and minimizing loss functions have a good compatibility in many cases, the former being more interpretable and decisive in model selection offering new insights and opportunity to select the best model.
>
> ------
>
> ### Writing and Presentation Improvements:
>
>  **Reduce excessive cross-referencing and repetitive statements to improve readability.**
> We deleted some of the cross references that were like an enforcing feedback loop, like
> *"see the Related Work section about the application of mutual information for further details"* in the Discussion.
>
> **Correct typos and improve the overall organization and formatting of the manuscript.**
> Thank you for the hint. After a thorough review for typos, we have corrected several ones. If you find more, please provide the exact site. Regarding the formatting of the manuscript, we used the TMLR template. Furthermore, we removed the large spaces. If you think of other formatting issues, please tell us.
>
> ----
>
> ### Continued on the next page

---

> ### Author Response · Authors · 2024-12-03
> **Answer to reviewer oj2b (5th part)**
>
> **Provide clearer, more concise explanations of key concepts and findings.**
>
> We would like to emphasize that the main difference between our work and others is considering input in evaluation of the model performance. More formally, we consider the **triple of input, prediction and ground truth** in contrast to other methods that consider **tuple of prediction and the ground truth** to evaluate the performance of a model. Therefore, our framework comments on the performance based on the provided input as well as the prediction, not solely based on the prediction. We have added *"In contrast, our approach consistently evaluates ..."* and *"All these measures are supposed to consider the triple of input ..."*.
>
> Although there are plenty of works such as **xu2019** considering information theoretic measures, they are not using it for the purpose we use, i.e., predictability of target based on the provided input/context. In light of this consideration, we apply the information theoretic measures/statistical tests between **input** and **residual** while **xu2019** utilizes it to reduce the distance between **prediction** and **ground truth**. There are some other works such as **Hjelm, 2018**, **Chen, 2016**, **Oord, 2018**, **Brakel, 2017** that utilize information theoretic measures for representation learning and thus have a different focus.
>
> To the best of our knowledge, none of the existing works try to comment on *"to what extent a data is predictable"* or *"for a given ML task the provided information is sufficient"*. Instead, they mainly try to improve the performance using information theoretic measures or find a better representation of the data.
>
> In industrial use cases, through our framework, stakeholders/customers can be notified if their provided data can be used to solve their AI use case or if this data has a low information shared with the tasks such as prediction or classification.
>
> These are highlighted in the introduction as we say:
> - *We utilize measures to evaluate the stochastic dependence and information content between random variables modeling the data to evaluate to what extent the data is interconnected and to quantify extractable information based on defined input and output (target) variables assembled from the dataset.*
> - *After fitting a model, we use these measures to estimate if there is still information left to extract. This evaluation is done by testing the stochastic independence and information content between the input and the deviations of the model output and the ground truth. For this purpose, we investigate the relation of random variables that model corresponding quantities.*
>
> Please also see Figure 1 that illustrates these bullet points. Apart from the introduction, we provide in the Methods section a concise explanation *"2.1 Basic concept for unpredictability in a nutshell"*. Furthermore, we structure the Methods section with subheadings to provide a clear overview of where to find explanations for the basic concepts. Furthermore, we provide a short overview about the purpose of each subsection of the results section in its beginning. Our findings of the work are the evidence that our proposed framework provides a useful tool for model evaluation. Section 4 is intended to show the usefulness with examples, we wrote *"In this section, we showcase our framework ..."*. With headings summarizing what we would like to show in each subsection, we intend to give the reader a quick overview over what we show. Please keep in mind that the concept is the finding in the sense of the main result.
>
> In case the presentation still needs more clearness and conciseness, please provide the sites where we need to improve.
>
> ------
>
> ### Broader Impact Concerns
>
> We agree that these questions the reviewers raise are very valid questions when decisions are automated to ensure they are aligned with ethics. However, in line with the other two reviewers, we think that we do not need such a statement as our framework is intended as a pure data analysis tool, meaning to evaluate well the extraction of the information from the data works. Our framework has no ethical assessment of these information that are extracted and thus is not able to prefer or reject some kind of information regarding ethical aspects.
>
> As an example, if a bias is in the data, it might only be considered as a successful extraction if the model has extracted it. Consequently, the data set needs to be collected and generated accordingly. On the other hand, our framework does not counteract any present safeguards that intend to protect ML algorithms taking unethical decisions. An analogous reasoning holds for data privacy issues. The measures are interpretable as we quantify extracted information and with a statistical consideration can make a clear decision on a level of significance if it cannot be rejected that the model has extracted all information available in the data.

---

> ### Author Response · Authors · 2024-12-03
> **Answer to reviewer oj2b (6th part)**
>
> ### **References**
>
> - **belghazi2018**: Belghazi, M. I., Baratin, A., Rajeshwar, S., Ozair, S., Bengio, Y., Courville, A., & Hjelm, D. (2018). Mutual information neural estimation. International conference on machine learning, 531--540.
>
> - **franzese2023**: Franzese, G., Bounoua, M., & Michiardi, P. (2023). MINDE: Mutual Information Neural Diffusion Estimation. arXiv preprint arXiv:2310.09031.
>
> - **brakel2017**: Brakel, P., & Bengio, Y. (2017). Learning independent features with adversarial nets for non-linear ica. arXiv preprint arXiv:1710.05050.
>
> - **oord2018**: Oord, A. van den, Li, Y., & Vinyals, O. (2018). Representation learning with contrastive predictive coding. arXiv preprint arXiv:1807.03748.
>
> - **chen2016**: Chen, X., Duan, Y., Houthooft, R., Schulman, J., Sutskever, I., & Abbeel, P. (2016). Infogan: Interpretable representation learning by information maximizing generative adversarial nets. Advances in neural information processing systems, 29.
>
> - **hjelm2018**: Hjelm, R. D., Fedorov, A., Lavoie-Marchildon, S., Grewal, K., Bachman, P., Trischler, A., & Bengio, Y. (2019). Learning deep representations by mutual information estimation and maximization. International Conference on Learning Representations.
>
> - **gromov2024**: Gromov, A., Tirumala, K., Shapourian, H., Glorioso, P., & Roberts, D. A. (2024). The unreasonable ineffectiveness of the deeper layers. arXiv preprint arXiv:2403.17887.
>
> - **choi2022**: Choi, K., & Lee, S. (2022). Combating the instability of mutual information-based losses via regularization. Uncertainty in Artificial Intelligence, 411--421.
>
> - **darbellay1999**: Darbellay, G. A., & Vajda, I. (1999). Estimation of the information by an adaptive partitioning of the observation space. IEEE Transactions on Information Theory, 45(4), 1315--1321.
>
> - **czyz2023**: Czy{.z}, P., Grabowski, F., Vogt, J. E., Beerenwinkel, N., & Marx, A. (2023). Beyond Normal: On the Evaluation of Mutual Information Estimators. Thirty-seventh Conference on Neural Information Processing Systems.
>
> - **moon1995**: Moon, Y.-I., Rajagopalan, B., & Lall, U. (1995). Estimation of mutual information using kernel density estimators. Physical Review E, 52(3), 2318.
>
> - **pizer1987**: Pizer, S. M., Amburn, E. P., Austin, J. D., Cromartie, R., Geselowitz, A., Greer, T., ... & Zuiderveld, K. (1987). Adaptive histogram equalization and its variations. Computer vision, graphics, and image processing, 39(3), 355--368.
> - **Song2020**: Song, J., & Ermon, S. (2020). Understanding the Limitations of Variational Mutual Information Estimators. International Conference on Learning Representations.
>
> - **gao2017**: Gao, W., Oh, S., & Viswanath, P. (2017). Density functional estimators with k-nearest neighbor bandwidths. 2017 IEEE International Symposium on Information Theory (ISIT), 1351--1355.
>
> - **xu2019**: Xu, Y., Cao, P., Kong, Y., & Wang, Y. (2019). L_dmi: A novel information-theoretic loss function for training deep nets robust to label noise. Advances in neural information processing systems, 32.
>
> - **bode1950**: Bode, H. W., & Shannon, C. E. (1950). A simplified derivation of linear least square smoothing and prediction theory. *Proceedings of the IRE, 38*(4), 417–425.
>
> - **houts2013**: Houts, S. E., Dektor, S. G., & Rock, S. M. (2013). A robust framework for failure detection and recovery for terrain-relative navigation. *Unmanned Untethered Submersible Technology*.
>
> - **kraskov2004**: Kraskov, A., Stögbauer, H., & Grassberger, P. (2004). Estimating mutual information. *Physical Review E, 69*(6), 066138.
>
> - **mitter2005**: Mitter, S. K. (2005). Nonlinear filtering of diffusion processes: A guided tour. In *Advances in Filtering and Optimal Stochastic Control: Proceedings of the IFIP-WG 7/1 Working Conference, Cocoyoc, Mexico, February 1–6, 1982* (pp. 256–266). Springer.
>
> - **reid2001**: Reid, I., & Term, H. (2001). Estimation II. *University of Oxford, Lecture Notes*.

---

### Review · Reviewer_HzAU · 2024-11-22

**Summary Of Contributions:**

The paper introduces a statistical framework designed to assess the predictability of a target variable given input data and to evaluate how well a machine learning model captures deterministic relationships within the data. The framework distinguishes between model errors that are due to unpredictable stochastic components and those that result from a failure to learn deterministic patterns. The proposed approach extends traditional measures of model success by focusing not just on prediction error magnitude but also on the model's ability to extract all available deterministic information. The framework provides tools to evaluate the extent to which a target variable is predictable from input data, using measures like mutual information, and chi-square tests. After training a model, the framework assesses whether there is remaining information to be extracted by testing the independence between input data and the residuals (deviations between model predictions and ground truth). Overall, the paper provides an interesting perspective on model evaluation, emphasizing the importance of understanding the limits of predictability and the potential for further model improvement.

**Audience:**

Yes

**Claims And Evidence:**

Yes

**Requested Changes:**

Mentioned in the weaknesses section

**Strengths And Weaknesses:**

### Strengths
- The paper introduces a novel framework that goes beyond traditional loss function minimization by focusing on the predictability of deterministic relationships, which is a valuable addition to model evaluation techniques.
- The paper has a clear practical application - by providing a method to assess when a model has extracted all available information, the framework can help prevent unnecessary computational effort and guide efficient model training and hyperparameter tuning.
- The framework is not limited to specific machine learning models, making it versatile and applicable across various domains and model architectures.


### Weaknesses
- The paper is not well drafted, particularly the experiment section is hard to follow with last of empty spaces and unclear plots. I would ask the authors to concisely present the important results in the main paper and move the rest to Appendix.
- I would like to see more experiments on real world datasets and comparisons with SOTA time series approaches. With the claims mentioned in contributions, it would be possible to improve the SOTA on time series forecasting and provide analysis. Can the authors also report the common metrics such as MSE/MAE for time series forecasting.

---

> ### Author Response · Authors · 2024-12-02
> **Answer to HzAU's review report**
>
> We thank the reviewer for the  time dedicated for providing feedback. Further, we thank for clearly outlining strengths of our work, in particular mentioning the practical application.
>
> Below we explain in detail how we
> improved our manuscript upon the received feedback. In the manuscript, changes are marked in blue.
>
> ## The paper is not well drafted, particularly the experiment section is hard to follow with last of empty spaces and unclear plots. I would ask the authors to concisely present the important results in the main paper and move the rest to Appendix.
>
> We apologize for the effort for following the result section. We now reduced the empty spaces and moved some part of the stacking procedure part for 1-step ahead prediction to the appendix as well as the multi-step ahead prediction for the stacking as the results are in line with the first experiment. Furthermore, we present now an overview in the beginning of the result section that very shortly says what the intention of each subsection is and what we would like to demonstrate as each section is selected to showcase an application scenario.
> We agree that the plots are not intended as a stand alone item, and might thus be unclear without further explanation. The plots are intended to provide a visualization/evidence to what is described and explained in the text. If the reviewer meant different things to be unclear, we kindly ask for more information what exactly is unclear and we need to improve.
>
>
> ## I would like to see more experiments on real world datasets and comparisons with SOTA time series approaches. With the claims mentioned in contributions, it would be possible to improve the SOTA on time series forecasting and provide analysis. Can the authors also report the common metrics such as MSE/MAE for time series forecasting.
>
> We would like to remark that our framework is intended to extend the current term of "accuracy" and model success as we outline in the Introduction "Our mathematical framework provides an additional insight for model evaluation
> and extends the current performance measures of ML models.".
> We have extended this site with more literature that also see the need for further extension. We write: "This extension aligns  with the perception in the area of generative models ...".
>
> According to the best of our knowledge there is no evaluation framework that has the same intention to evaluate the model with regard to extracting the deterministic relation regarding input and output/model deviations "We extend the term for model success by its ability
> to extract or learn, respectively, all the available deterministic relations."
> In the Related methods section, we outline how we can extend other methods that evaluate different notions of model success.
>
> Consequently, when we ask for improvement of SOTA methods, we need to define according to what metric. We do not claim in our work that applying our framework provides better MSE/MAE values. In Section 4.3 we write "Furthermore,
> the increase of the chi-square and the mutual information on the test set for later epochs might indicate an
> overfitting since the minimum based on the $L^1$-loss function does not necessarily coincide with the maximum
> of extracted deterministic relations defined by a stochastic measure, ...". We have extended more here to provide an illustration why this could be the case: " A reason for the difference of the loss functions in taking optima might be that ..."
> We summarize that the main contribution of our paper is exactly that MSE/MAE are not a sufficient measure as they do not differentiate what data points are predictable given the input and what are not. We have explained that point more in the Introduction with "The ability to differentiate what data points are predictable and what are not given the input data is not provided by accuracy measures or loss functions that only consider the model output and ground truth, like mean squared or absolute errors, instead we additionally consider the information the model deviations share with the input data."
>
>
> Further research to have a loss function based on stochastic independence that optimizes for extracting these deterministic relations is ongoing. We outline this future research in the Conclusion "According to the presented framework the mutual information between...".
> With such a framework we can then systematically optimize parameters to extract as much of the deterministic relations as the used model architecture can. To measure the "amount of extracted relations/information", a measure for stochastic independence, like mutual information, can be used to measure which is more successful in that.

---

### Author Response · Authors · 2024-12-03
**Response to Comments and Updated Manuscript**

Dear Action Editor and Reviewers,

We have addressed all the questions and incorporated the requested changes. Additionally, we have uploaded an updated manuscript in which the newly added text is highlighted in blue for clarity. We have also slightly revised some figures.

We hope that our responses and updates adequately address the concerns raised. We remain at your disposal for any further questions, suggestions, or points for discussion.

Thank you for your time and consideration.

Authors

---

### Decision · Action_Editor_mii3 · 2025-01-11

**Recommendation:** Accept with minor revision

**Comment:**

First, the reviewers were split in their recommendation. Some of their suggested improvements had been addressed and some not. The remaining concerns include extended evaluation on real-world problems and further theoretical analysis of the success and failure of the proposed approach (this question is only currently addressed in the discussion, and only informally). Beyond this, several reviewers had requested that the authors further edit the manuscript to improve readability. This is my main concern with the current version. I therefore recommend that the paper is revised before publication.

Second, the "model stacking" procedure proposed in 2.3 is closely related to the long-established idea of boosting [1] (e.g., AdaBoost, gradient boosting, etc), in the sense of sequentially growing models through repeated prediction of residuals. The similarity is high enough to warrant comment at the very least. Section 3 comments on the relationship to ensemble methods but does not compare to boosting.

Third, the results section should be further curated and edited to help the reader. For example, Tables 2 and 3 present results for multiple trials that could be presented as aggregates. Several of the subplots in Figures 4 and 5 are completely flat lines and several of them are not commented on. The font in the legend in Figure 8 is small enough to be barely legible.

Finally, the paper is in its current form *substantially* longer than the average TMLR paper, which, while permitted by the journal format, should be "justified by its content", as stated in the author guidelines. It is my view, and mentioned by reviewers, that this criteria is not met: two of the reviewers asked for a more concise presentation of key results. Still, the revised submission is longer than the original submission, despite 2/3 reviewers asking for concisenesss.

I recommend that the paper is revised with the primary goals to:
* Clearly expose the claimed contributions of the work by editing the introduction.
* Make it easier for a reader to assess whether those claims are supported by experiments by editing the results section, highlighting key results.
* Reduce repetition and discussion of tangential results and potential applications of the proposed idea beyond what is tested in the paper

Additionally, I would like the authors to include a comment on the relationship between their stacking procedure and boosting methods [1].

[1] Shapire & Freund. Boosting: Foundations and Algorithms, 2012

**Audience:**

Yes

**Claims And Evidence:**

The claimed contributions are not clearly highlighted in the introduction, which is long and meandering in its structure. This was noted also by reviewers. In part, this is because the paper aims for a very high degree of generality, claiming that the approach is applicable for many analyses, learning problems, and model classes. However, this makes it difficult to assess whether the evidence provided is sufficient to determine whether the goals of the work have been achieved.

---

> ### Author Response · Authors · 2025-01-30
> **Summary of changes**
>
> Dear AE,
>
> Thank you for your decision and comments. We revised the manuscript accordingly and changed the following main points:
>
> We now focused on the core of the paper which is to provide a method with which we can measure the amount of deterministic relations between input and output or model deviations, respectively, and show experiments that demonstrate some basic applications for such measures.
>
> For this reason, we put everything related to stacking to the appendix making up a new section there as it is just another application of the core idea of our paper.
>
> In the corresponding section, one can also find the relation of the stacking to gradient boosting. Furthermore, how our core of the paper can extend the concept of gradient boosting is described in the Related methods section. In detail, our framework can provide a criterion when to stop adding more learners to the ensemble model, in particular, when all deterministic relations have been extracted.
> We stressed the core clearly by bullet points in the Introduction and we link these bullet points to the experiments in the beginning of the Results section to show the readers where to find the corresponding evidence of the single bullet points and thus the successful demonstration of potential applications of our proposed framework.
>
> Furthermore, we streamlined and polished the introduction section and the results section with regard to the repetition focusing on what is shown in the experiments. In this course, we changed figures according to your suggestion (in particular reducing the ones with flat lines; increasing font size) and replaced Table 2 and 3 with a graphic illustrating the text.
> Due to the narrower focus of the paper, we pruned the Discussion accordingly. Summarizing, the main part of the paper is reduced by 6 pages.
>
> Moreover, we further optimized the code with regard to computation time and updated the corresponding git repository.